



# Research on the unusual spring 2020 Arctic stratospheric ozone depletion above Ny-Ålesund, Norway

Qidi Li[1,2], Yuhan Luo[1*], Yuanyuan Qian[1,2], Chen Pan[3,4], Ke Dou[1], Xuewei Hou[5], Fuqi Si[1], Wenqing Liu[1]

[1]Key Laboratory of Environmental Optics and Technology, Anhui Institute of Optics and Fine Mechanics, Hefei Institutes of Physical Science, Chinese Academy of Sciences, Hefei, 230031, China
[2]University of Science and Technology of China, Hefei, 230026, China
[3]Jiangsu Meteorological Observatory, Jiangsu Meteorological Bureau, Nanjing, 210008, China
[4]Key Laboratory of Transportation Meteorology, China Meteorological Administration, Nanjing, 210009, China
[5]Collaborative Innovation Center on Forecast and Evaluation of Meteorological Disasters, Nanjing University of Information Science and Technology, Nanjing, 210044, China

*Correspondence to*: Yuhan Luo (yhluo@aiofm.ac.cn)

**Abstract.** Of the severe stratospheric ozone depletion events (ODEs) reported over the Arctic, the third and most severe occurred during the spring of 2020; we analyzed the reasons for this event herein. We retrieved the critical indicator ozone vertical column density (VCD) using zenith scattered light differential optical absorption spectroscopy (ZSL-DOAS) located in Ny-Ålesund, Svalbard, Norway. The average ozone VCDs over Ny-Ålesund between March 18 and April 18, 2020, were approximately 274.8 Dobson units (DU), which was only 64.7% of that in normal years. The retrieved daily averages of ozone VCDs were compared with satellite observations from Global Ozone Monitoring Experiment 2 (GOME-2), a Brewer spectrophotometer, and a Système d'Analyze par Observation Zénithale (SAOZ) spectrometer at Ny-Ålesund; the resulting Pearson correlation coefficients were relatively high at 0.94, 0.86, and 0.91, with relative deviations of 2.3%, 3.1%, and 3.5%, respectively. Compared with normal years, the 2020 daily peak relative ozone loss was 44.3%. During the 2020 Arctic spring ODE, the ozone VCDs and potential vorticity (PV) had a negative correlation with their fluctuations, suggesting a clear effect of the polar vortex on stratospheric ozone depletion. To better understand what caused the ozone depletion, we also considered the chemical components of this process in the Arctic winter of 2019/2020 with the specified dynamics version of the Whole Atmosphere Community Climate Model (SD-WACCM). The SD-WACCM results indicated that both ClO and BrO concentrations peaked in late March, which was a critical factor during the ozone depletion observed in Ny-Ålesund. Chlorine activation was clearly apparent during the Arctic spring of 2020, whereas the partitioning of bromine compounds was different from that of chlorine. By combining observations with modeling, we provide a reliable basis for

further research on global climate change due to polar ozone concentrations and the prediction of severe Arctic ozone

depletion in the future.

**Key Words:** Arctic ozone depletion, DOAS, ozone VCD, polar vortex, SD-WACCM, halogen species

## 1 Introduction

Stratospheric ozone is essential for human health, surface ecosystems, and the climate in general (McKenzie et al., 2011) because it absorbs ultraviolet (UV) solar radiation and converts it into thermal energy. The characteristic absorption bands of

stratospheric ozone are mainly located in the Hartley and Huggins zones of the UV region and in the Chappuis zone of the visible spectrum, thereby absorbing almost all UV-C (i.e., wavelengths < 280 nm) and some UV-B (i.e., wavelengths ranging between 280 and 315 nm) radiation. Since the late 1970s, Antarctic stratospheric ozone during the austral spring has decreased sharply, mainly because of the heterogeneous catalytic reactions between ozone and active halogen radicals generated by the conversion of chlorofluorocarbons derived from anthropogenic emissions (Farman et al., 1985). As

anthropogenic emissions of ozone-depleting substances since the Montreal Protocol was enforced, the concentrations of ozone in the stratosphere were predicted to recover to pre-1980 values in 2060 (Solomon et al., 2016).

The severe ozone depletion over the Arctic is relatively uncommon compared with that in the Antarctic. During normal Arctic winters, the polar vortex usually fractures and disperses early due to huge planetary wave activities and Brewer–Dobson circulation dynamics (Manney et al., 2003; Dameris, 2010; Harris et al., 2010). Thus, in the Arctic, the duration of

the vortex is shorter and relative ozone loss is also lower (Solomon et al., 2007). However, irregular changes in Arctic ozone in recent years have attracted worldwide attention and challenged the existing model. The most severe Arctic ozone depletion lasted for nearly a month, from March to April 2020 (Dameris et al., 2021). Between mid-February and late March 2020, the persistence of anomalously faint wave activities in the Arctic led to an abnormally persistent and cold vortex, which caused significant ozone loss (Hu, 2020). This event was the third reported low Arctic ozone event, following those

that occurred in the springs of 1997 and 2011 (Hansen and Chipperfield, 1999; Manney et al., 2011).

The powerful and persistent vortex during the winter and spring is considered a main cause of significant ozone depletion in the Arctic (Bognar et al., 2021). Extremely low air temperatures  (< −195 K) are essential to produce polar stratospheric

clouds (PSC). The PSC formed from water-ice and nitric acid trihydrate can be used as a surface for heterogeneous

interactions, leading to the conversion of reactive halogens from the halogen reservoirs, which can cause serious ozone loss

(Frieβ et al., 2005; Marsing et al., 2019). PSCs are classified into three types: nitric acid trihydrate (NAT), ice PSCs, and

supercooled ternary solution (STS), and their threshold temperatures are $T_{nat}$ (−195 K), $T_{ice}$, and $T_{sts}$, respectively (Toohey et

al., 1993). PSC might also grow large enough to precipitate and remove $HNO_3$ in the stratosphere, which is the reservoir of

$NO_2$. The resulting denitrification from the polar vortex hinders chlorine deactivation by $NO_2$ (Salawitch et al., 1989;

Arblaster et al. 2014). Active chlorine is rapidly photolyzed because of the recovery of spring sunlight when ozone loss

occurs via the self-reaction of ClO (Molina and Molina, 1987), as well as the cross-reaction of ClO and BrO (McElroy et al.,

1986). It is essential that the vortex retains low temperatures and carries on as a transport impediment so that ozone can

remain depleted without $NO_2$ to inactivate chlorine.

The observed Arctic ozone depletion is invaluable for validating stratospheric ozone simulations and for understanding the

processes that cause Arctic stratospheric ozone depletion. Currently, total column ozone (TCO) detection utilizes the

characteristic ozone absorption in the UV and visible spectra, which provides accurate ozone identification and quantitative

measurements. Ozone vertical column density (VCD) is primarily achieved by satellite observation, Pandora

spectrophotometer, Fourier-transform infrared spectrometer, Brewer spectrophotometer, balloon-borne ozone sonde, and

ground-based differential optical absorption spectroscopy (DOAS) observation (Kuttippurath et al., 2012; Manney et al.,

2020; Bognar et al., 2021; Grooß and Müller, 2021). Among these, ground-based observations are crucial to calibrate

remotely sensed observations and optimizing inversion results (Lu et al., 2006). The impacts of the aberrantly powerful and

persistent vortex on ozone in the Arctic were investigated using satellite observations, ozonosonde measurements, and data

from the European Centre for Medium-Range Weather Forecasts (ECMWF) (Wohltmann et al., 2020; Lawrence et al., 2020).

The major stratospheric halogen species, chlorine, and bromine were investigated in this ozone depletion event (ODE)

(Wohltmann et al., 2017, 2021). In addition, modeling plays an essential role in the investigation of ozone depletion

(Simpson et al., 2007). Recently, stratospheric chemical patterns, consisting of a group of heterogeneous reactions, have

been developed in various models according to investigations and experiments conducted in the polar area (McKenna et al.,

2002; Grooß et al., 2011, 2018). Global and area models using different stratospheric chemical patterns have been applied to



simulate ozone columns, which usually compare well with satellite observations and ozonosonde data (Pan et al., 2018; Grooß and Müller, 2021).

Accurate ground-based observations can make a significant impact on improving the accuracy and reliability of models as well as enhancing our understanding of the reasons for ozone depletion. We have developed a ground-based DOAS system that can conduct TCO observations in the Arctic. The zenith scattered light observation mode was applied to measure TCO using the Langley Plot method (Frieß et al., 2005).

We analyze the reasons for this ODE in the unusual spring of 2020 above Ny-Ålesund, Norway. The methods and data are

given in Sect. 2, which covers the presentation of the experimental location and DOAS instrument, the DOAS method, the specified dynamics version of the Whole Atmosphere Community Climate Model (SD-WACCM), Global Ozone Monitoring Experiment 2 (GOME-2) observations, Brewer measurements, Système d'Analyze par Observation Zénithale (SAOZ) measurements, ECMWF data, and ozonesonde data. Section 3 presents the results, where Sect. 3.1 describes the results of ozone VCDs from February 2017 to October 2021 and ozone loss in spring 2020. The zenith scattered light DOAS (ZSL-

DOAS) retrieved the daily variations in ozone VCDs, which were in comparison with GOME-2 observations, Brewer, and SAOZ measurements. A detailed characterization of this ODE is presented for establishing the basis of the subsequent analysis. The relationship between Arctic ozone depletion and meteorological conditions in terms to temperature and potential vorticity (PV) is described in Sect. 3.2. The effect of PV on ozone depletion was investigated using ozone VCD and stratospheric PV data. In Sect. 3.3, this ODE was analyzed using the SD-WACCM to further illustrate the ozone depletion

process, and to explore the effects of chemical depletion and dynamic transport on this ODE. The influence of the halogen species is discussed in Sect. 3.4. The comprehensive summary is provided in Sect. 4.

## 2 Methods

### 2.1 Experimental location and DOAS instrument

The DOAS instrument was placed at the Yellow River Station (78.92° N, 11.93° E) in the Arctic. Figure 1 shows the

experimental location and DOAS instrument, in Ny-Ålesund, Svalbard, Norway. The DOAS instrument mainly includes the prism, telescope, computer, and spectrometer. This spectrometer was designed for wavelengths between 290 and 420 nm,





and had the spectral resolution (FWHM) of 0.5 nm. The ozone slant column density (SCD) was retrieved, with the raw data obtained in the zenith direction.

## 2.2 Calculation of ozone VCD

Radiation intensity decreases when it passes through absorbing media (mainly trace gases). Because of the different absorption bands, characteristic peaks, and intensities of various gases, we can retrieve the content of each trace gas, according to Lambert–Beer's law as follows:

$$ln\frac{I^*(\lambda)}{I_0(\lambda)} = \sum[\sigma_j^*(\lambda)c_jL] = \sum[\sigma_j^*(\lambda)SCD_j]. \tag{1}$$

Here, $I_0(\lambda)$ represents the raw intensity of solar scattered spectral radiation received by the ground-based detector, $I^*(\lambda)$
denotes the incident intensity of the solar radiation spectrum, $L$ represents the distance travelled by the incident light in the absorbing gas, $\sigma_j^*(\lambda)$ represents the absorption cross section for the $j$th gas, $c_j$ denotes concentration of the $j$th gas, $SCD_j = \int c_jL$ represents the SCD of the $j$th gas, and $D = \ln\frac{I^*(\lambda)}{I_0(\lambda)}$ denotes the differential optical density.

We calculated the SCD for ozone with the QDOAS program (Platt and Stutz, 2008). In the experiment, ozone was retrieved in the 320–340 nm band, and the gases involved in the retrieval include $O_3$ (223K, 243K), $NO_2$ (298K), $O_4$ (293K), and ring
structure. Table 1 lists the parameters for the gases involved in the retrieval. Figure 2 shows a spectrum obtained during monitoring on June 13, 2021. The measured spectrum was fitted to give an ozone SCD of $4.09 \times 10^{17}$ molec cm$^{-2}$, and the root mean square of the spectral fitting residual was $5.28 \times 10^{-4}$.

As SCD is dependent on the instrument's observation mode and the prevailing meteorological conditions, it is necessary to shift to VCD, which is independent of the mode of observation:

$$AMF = \frac{SCD}{VCD}. \tag{2}$$

Here, the Air Mass Factor (AMF) can be obtained from the SCIATRAN model and is influenced by trace gas profiles, pressure, temperature, ozone, aerosol profiles, clouds, and surface albedo. Since the "Ring effect" in the measurement caused by the Fraunhofer reference spectra can lead to lower trace gas levels in the retrieval than in actual atmospheric levels, this is corrected in the calculation:

$$dSCD(\alpha,\beta) = SCD(\alpha,\beta) - SCD_{FRS} = AMF(\alpha,\beta)VCD - SCD_{FRS}. \tag{3}$$

**Atmospheric Chemistry and Physics Discussions**

Here, $SCD_{FRS}$ denotes Fraunhofer absorption. Figure 3 presents the results of a linear fit of the dSCD and AMF on June 13, 2021. The ozone VCD for this date was $8.799 \times 10^{18}$ molec cm$^{-2}$ and produced a fitting error of $3.361 \times 10^{16}$ molec cm$^{-2}$. The uncertainties in ozone VCD retrieval originate from uncertainties in the retrieval of SCD and AMF. The error in retrieving ozone SCD was calculated as 3.01% within the 95% confidence interval. The solar zenith angle calculated in this

research ranged between 35° and 80°, with surface albedos between 0.08 and 0.6. Based on the average monthly climate, *a priori* ozone profile can be achieved. Table 2 provides the parameters used to calculate the AMF effect on wavelength. The uncertainties of the AMF due to wavelength selection were calculated as $(AMF_\lambda - AMF_{328})/AMF_\lambda$, where λ denotes the wavelength. According to Table 2, the uncertainties of the AMF in the wavelength ranged from −4.257% to 4.630%, and the average uncertainty was 2.030%. Based on evaluation of the OMI ozone products, AMF had an uncertainty of about 2% for

*a priori* ozone profile (Bhartia, 2002). The average AMF uncertainty was calculated as 2.85% using the following equation:

$$\sqrt{AMF_{wave}^2 + AMF_{profile}^2}$$ , where $AMF_{wave}$ denotes the error of AMF influenced by wavelength, and $AMF_{profile}$ denotes the AMF error affected through *a priori* ozone profile. The total error in the retrieved ozone VCD was 4.15%, calculated using the following error equation, $E_{VCD} = \sqrt{E_{SCD}^2 + E_{AMF}^2}$ , where $E_{SCD}$ and $E_{AMF}$ denote the errors of SCD and AMF, respectively.

**2.3 SD-WACCM**

The parameters employed in the Community Atmosphere Model Version 4 (CAM4) were applied to the WACCM (Neale et al., 2013). We used the SD-WACCM with meteorological parameters driven by Modern Era Retrospective-Analysis for Research and Applications version 2 (MERRA-2) data (Gelaro et al., 2017). The Model for Ozone and Related Chemical Tracers, version 3 (MOZART-3) provided the chemical parameters for the WACCM (Kinnison et al., 2007). The SD-WACCM had the horizontal resolution of 1.9° × 2.5° (lat × lon). The model was divided vertically into 88 layers, covering

an altitude of ~140 km from the ground to the bottom of the thermogenic layer. Meteorological fields were calculated using a nudging method in the model (Lamarque et al., 2012). Data from MERRA-2 guaranteed the accuracy of simulated values for meteorological fields below 50 km (Kunz et al., 2011). This can be employed for the study of specific weather events. Linear transitions were used in the 50–60 km altitude range and over 60 km, and online calculations were performed. The SD-



WACCM can be applied for research on chemical and dynamic processes in the atmosphere. (Lamarque et al., 2012; Pan et

al., 2019).

## 2.4 Auxiliary data

On October 19, 2006, Europe launched the MetOp-A satellite, which carries the GOME-2. The GOME-2 has a band between

240 and 790 nm, a spectral resolution ranging from 0.2 to 0.5 nm, and a nominal swath width spatial resolution of $80 \times 40$

$km^2$ (Koukouli et al., 2014). The GOME-2 dataset provided the daily mean VCD data (source: https://avdc.gsfc.nasa.gov/,

155  last access: 18 June 2022). Brewer spectrophotometers used holographic diffraction gratings to obtain the directly

transmitted intensity of sunlight. (Kerr, 2002). Ozone columns were calculated by averaging five consecutive measurements.

The error of the Brewer instrument was approximately 0.5% (Zhao et al., 2021). The Brewer dataset provided the daily mean

ozone data (source: https://woudc.org/, last access: 18 June 2022). The SAOZ instrument is a UV-Vis spectrometer

belonging to the worldwide analogous instrument networks (Pommereau and Goutail, 1988). The SAOZ instrument provided

160  a viewing angle of approximately 20° and measured trace gas concentrations in the stratosphere based on DOAS technology

(Platt & Stutz, 2008). Hendrick et al. calculated an error of 5.9% for the measurement of ozone by SAOZ (2011). The SAOZ

dataset provided the daily mean ozone VCD data (source: http://saoz.obs.uvsq.fr/, last access: 18 June 2022).

The ERA5 hourly pressure-levels data from 1959 to 2022 from the ECMWF website (source:

https://www.ecmwf.int/en/newsletter/147/news/era5-reanalysis-production, last access: 18 June 2022) provided the daily

165  temperature and PV data. ERA5 replaced ERA-Interim reanalysis. The ERA5 data have the spatial resolution of $0.25° \times 0.25°$

and were divided into 37 layers vertically, from 1000 hPa to 1 hPa. Since 1992, the Alfred Wegener Institute has recorded

the total ozone column and vertical profile using balloon-borne ozonesonde. The temporal resolution of the sounding data

from March 25 to April 13, 2020, is once per day, whereas the others are normally once per 3 d during the spring and once

per week during the other seasons (source: https://ndaccdemo.org/, last access: 12 January 2021).



## 3 Results and discussion

### 3.1 Results of ozone VCDs

The ozone VCDs obtained from the GOME-2 satellite, Brewer, SAOZ, and ground-based instruments from February 2017 to October 2021 are shown in Fig. 4. In normal years, ozone VCD begins to decrease in March and has a gradient of approximately 0.92 DU per day, while the lowest ozone VCDs occur in October. The average ozone VCD between March 18 and April 18, 2020, was at an abnormally low level of ~274.8 DU, with a minimum of 241.2 DU on April 5. The average ozone VCD during the same period in normal years was approximately 424.6 DU. All instruments detected relatively low levels of ozone from March 18 to April 18, 2020.

Figure 5 presents the linear fit between observed ozone VCDs and GOME-2 observations, Brewer, and SAOZ measurements. Their pearson correlation coefficients were relatively high at 0.94, 0.86, and 0.91, and the relative deviations were 2.3%, 3.1%, and 3.5%, respectively. The ground-based DOAS measurements correlated well with ozone VCDs observed using GOME-2 onboard the MetOp satellite and Brewer and SAOZ instruments. Thus, the method of observing the VCDs of Arctic ozone using a ground-based DOAS instrument is reliable and valid.

The ozone data for 2020 and the average ozone data for the other years (2017, 2018, 2019, and 2021) from the ZSL-DOAS instrument, satellite observations from GOME-2, and measurements from the Brewer and SAOZ instruments are shown in Fig. 6a. The diurnal means of absolute and relative ozone loss between data in 2020 and the mean of the other four years are displayed in Fig. 6b and Fig. 6c, respectively. In 2020, the daily peak absolute losses from the GOME-2, ZSL-DOAS, Brewer, and SAOZ datasets were 189.8, 195.7, 181.4, and 177.7 DU, respectively. The 2020 daily peak relative losses from the GOME-2, ZSL-DOAS, Brewer, and SAOZ datasets were 43.6%, 44.3%, 40.3%, and 40.6%, respectively.

### 3.2 Relation of Arctic ozone depletion to meteorological conditions

Daily average temperatures of Ny-Ålesund between November 2016 and September 2021 were measured at 70 hPa in the low stratosphere, where significant ozone depletion tends to occur (Fig. 7). Furthermore, temperatures dropped below the threshold (−195 K) at which the PSCs were formed. A relatively colder stratosphere over Ny-Ålesund persisted for a longer duration during the winter of 2019/2020 than in previous years, with air temperatures as low as 190 K. The number of days with daily temperatures below 195 K during the winters of 2017/2018, 2019/2020, and 2020/2021 are shown in Table 3. In

addition, overall winter temperatures in 2019/2020 were lower than those of the same period in normal years and had a

prolonged period with cool temperature, leading to prolonged PSCs. Because of the atypically faint wave activities that

occurred between mid-February and late March 2020 over the Northern Hemisphere (Dameris et al., 2021), the trend for

abrupt warming in spring 2020 was lower than in normal years.

Ozone depletion occurs when the temperature is sufficiently low and reaches a threshold temperature. In addition, a cold and

stable polar vortex is a prerequisite for ensuring that Arctic stratospheric temperatures are sufficiently low. The sign of PV

was positive in the Arctic and negative in Antarctica. PV was a key parameter for characterizing polar vortex. The PSC

developed in the vortex may result in significant ODE by activating halogen species. To also assess changes after ozone

recovery, we evaluated the PV, temperature, and ozone VCD from ground-based observations between March 18 and April

28, 2020 (Fig. 8).

Figure 8a–c shows that the tendencies for PV and ozone VCD are inversely related, i.e., PV correlates negatively with ozone

VCD. Similarly, Arctic spring ozone depletion was closely related to PV. When ozone VCD decreased, the PV value

increased. The ozone VCDs fluctuated between 241.2–334.6 DU, with ozone recovering to 388.1 DU on April 19, and then

returning to normal values (Fig. 8a). The observed ozone VCD and temperature had similar fluctuation patterns, suggesting

that ozone was significantly depleted in the colder Arctic low stratospheric vortex. Thus, the effect of the polar vortex on

ozone depletion in the stratosphere was clear.

### 3.3 Impact of halogen species

To further research the conditions and mechanisms of this ODE, we used a chemical model to characterize chemical

components between November 1, 2019, and July 1, 2020. To validate the reliability of the WACCM simulated results, we

needed to prove its capability for recreating observations in the atmosphere. Therefore, we compared WACCM simulations

with ozonesonde measurements. Figure 9 presents the comparison of temperature and ozone profiles over Ny-Ålesund from

the WACCM simulations and the ozonesonde between January 1 and July 1, 2020. Fig. 9a–b shows a gradual depletion of

ozone from 16 to 20 km in early March, and the mixing ratio at a similar altitude was unusually low from late March to early

April, which corresponded to ground-based observations. A mixing ratio of less than 0.5 ppmv within the altitude range

suggested that ozone was nearly completely depleted. This low value was uncommon, as the ozone mixing ratio was above

0.5 ppmv over the Arctic during 2011 (Solomon et al., 2014). There was an aberrantly cold spring in 2020, with low

temperatures lasting until mid-April (Fig. 9c–d). In January and February 2020, the temperature in the 15–25 km altitude

range was lower than $T_{nat}$, providing favourable conditions for PSC formation. As shown in Fig. 9, accurate simulations of

the ozone and temperature profiles strengthened the credibility of the WACCM results. However, there are some

discrepancies that exist in the model and observations. Because of the overestimation of temperature, the catalytic cycles that

cause ozone depletion in PSCs are underestimated. Therefore, there was an overestimation of ozone by the model compared

to the observations.

Between late December 2019 and January 2020, we observed abnormally increasing, high $HNO_3$ values above Ny-Ålesund

(Fig. 10b), suggesting abundant PSC formations. In contrast, in January 2011, analogous but lower values were recorded

(Manney et al., 2011). Between late January and early February 2020, $HNO_3$ changed abruptly from abnormally high values

to normal values, which indicated the abundant PSC activities of the period.

In the PSC, chlorine and bromine compounds are activated and the activated halogen species can cause ozone depletion.

Figure 10d–i presents the average diurnal concentration changes of chlorine and bromine compounds above Ny-Ålesund

during the ODE. Between mid-February and early March 2020, the ClO level intensively increased over Ny-Ålesund,

whereas the concentrations of $ClONO_2$ and HCl were low. However, the HBr levels remained elevated, which did not occur

in the HCl pattern during the same period. The results showed apparent chlorine activation during the Arctic spring of 2020.

During polar springs, PSCs and aerosol particles are considered to be the main cause of halogen species activation in the

atmosphere (Portmann et al., 1996; Tritscher et al., 2021). Chlorine was activated by $ClONO_2$ + HCl and this reaction

improved up to 10 times when the temperature reduced by 2.3 K (Wegner et al., 2012). Therefore, the persistently low

temperatures during the Arctic spring of 2020 had a profound impact on the dominant chlorine activation reaction. In early

March, chlorine was deactivated as HCl, and the PSC that permitted chlorine activation remained. Activated chlorine

compounds were mainly deactivated as $ClONO_2$ by the ClO + $NO_2$ reaction (Müller et al., 1994; Douglass et al., 1995). The

model additionally simulated concentration changes in $ClONO_2$ from early March to mid-April 2020. In mid-April 2020,

$ClONO_2$ stopped increasing and ClO was almost depleted when the ozone concentration started to recover. Chlorine

activation began in early December 2019 as well as lasting until early April 2020. Owing to severe ozone depletion, large



amounts of HCl were produced during late March and April 2020, with an apparent HCl increase in mid-April, which was

similar to the deactivation that occurred in the Antarctic.

Figure 11 displays the simulated average diurnal mixing ratios of ozone, chlorine, and bromine compounds for heights of

17.5 km above Ny-Ålesund, where significant ozone depletion occurred. Bromine was predominantly present as HOBr and

$BrONO_2$ at night before chlorine activation, and almost all bromine was present as BrCl at night after chlorine activation in

the Arctic winter of 2004–2005 (Wohltmann et al., 2017). We also noted that the partitioning of bromine compounds

differed from that of chlorine. HCl and $ClONO_2$ were the main constituents of $Cl_t$ (the total concentration of the following

chlorine compounds: ClO, HCl, HOCl, and $ClONO_2$) from November 2019 to late January 2020, whereas HOBr and BrCl

were the main constituents of $Br_t$ (the total concentration of the following bromine compounds: BrO, HBr, HOBr, $BrONO_2$,

and BrCl) during the same period. BrCl was produced by the BrO + ClO reaction. Additionally, reactions in this period via

HOBr + hv into Br and via BrCl + hv into Br and Cl are extremely important for the contribution of both chlorine and

bromine radicals. Consequently, the heterogeneous reaction of stratospheric Br increases the concentration of the active BrO

component in the stratosphere. In February 2020, the values of HBr, HOBr and $BrONO_2$ in the simulated data set (Fig. 11b)

were extremely low, while the BrO value started to increase.

Bromine existed mainly as HOBr before chlorine activation began. When chlorine was activated, BrCl became the major

constituent of $Br_t$. Because they rapidly photolyzed to Br in the daytime, these were not true reservoir gases compared with

the lower active chlorine. Although the concentrations of these gases were quite low, there was a significant potential for

ozone depletion (Lary, 1996; Solomon, 1999). BrO increased to its peak values on April 31, 2020, when ozone dropped to

1.16 ppmv. After April 18, BrO gradually stablilized and ozone began to recover, when the $BrONO_2$ produced by the BrO +

$NO_2$ reaction became the main constituent of $Br_t$.

## 4 Conclusion

In this research, the ozone VCD was obtained from a ground-based instrument, the GOME-2 satellite, and the Brewer and

SAOZ instruments and further evaluated with a correlation analysis. The Pearson correlation coefficients were 0.94, 0.86,

and 0.91, and the relative deviations were 2.3%, 3.1%, and 3.5%, respectively. Therefore, we can conclude that the method

of observing the VCDs of Arctic ozone using a ground-based DOAS instrument is reliable and valid. In 2020, the daily peak

relative ozone losses compared to normal years from the GOME-2, ZSL-DOAS, Brewer, and SAOZ datasets were 43.6%,

44.3%, 40.3%, and 40.6%, respectively. The results indicated that all instruments recorded severe ozone depletion from

March 18 to April 18, 2020.

The effect of the polar vortex on ozone depletion in the stratosphere was clear. During the winter of 2019/2020, Arctic low

stratospheric temperatures were unusually low. The vortex was peculiarly steady before early April, enabling the PSCs to be

produced, and this matched the changes in simulated $HNO_3$. This resulted in substantial ozone depletion until mid-April. The

observed ozone VCD and temperatures had similar fluctuation patterns, whereas PV was negatively correlated with ozone

VCD over Ny-Ålesund in the spring.

The ozone and temperature profiles were simulated by SD-WACCM, and these simulations corresponded well with

ozonesonde measurements. The model results show that ozone depletion at a height range of 16–20 km is evident from late

March to early April, which corresponds to the ozone VCDs obtained from the ground-based instrument. In 2020,

exceptional meteorological conditions contributed to a significant increase in reactive chlorine, resulting in an unprecedented

ozone loss in the Arctic. An apparent HCl increase occurred in mid-April 2020, when $ClONO_2$ stopped increasing and ClO

was almost depleted as the ozone concentration started to recover. Before chlorine activation began, bromine mainly existed

as HOBr; however, after chlorine activation, bromine mainly existed in the form of BrCl. Furthermore, they rapidly

photolyzed to Br in the daytime and had high potential to cause ozone depletion.

Observations of ozone VCDs over Ny-Ålesund will continue in order to monitor future ozone changes over the area. Further

synthetic analyses based on chemistry–climatic modeling and observational data are needed to study ozone recovery and its

effect on climate change and the ecological environment.

*Data availability.* Measurements and calculation of ozone VCDs above Ny-Ålesund, Norway, from 2017 to 2021 and the

results from the SD-WACCM used in this research are available from Yuhan Luo from AIOFM, CAS (yhluo@aiofm.ac.cn).

GOME-2 data are download from https://avdc.gsfc.nasa.gov/, Brewer data from https://woudc.org/, SAOZ data from

http://saoz.obs.uvsq.fr/, ECMWF data from https://cds.climate.copernicus.eu/, and ozonesonde data from https://ndaccdemo.org/.

*Author contribution.* QL: Methodology, Investigation, Software, Formal analysis, Validation, Visualization, Writing. YL:
Funding acquisition, Methodology, Formal analysis, Writing, Reviewing, Editing, Resources. YQ: Methodology, Software, Formal analysis, Visualization. CP: Methodology, Software, Formal analysis. KD: Validation, Resources. XH: Methodology, Formal analysis. FS: Validation, Resources. WL: Supervision.

*Competing interests.* The authors declare that they have no conflict of interest.

*Acknowledgement.* This research was financially supported by the National Natural Science Foundation of China (Grant
Nos.41941011 and 41676184) and the Youth Innovation Promotion Association of CAS (Grant No.2020439). We thank the organizations of the Chinese Arctic and Antarctic Administration (CAAA), Polar Research Institute of China, and teammates of the Chinese Arctic Yellow River Station for their kind help. We gratefully thank the BIRA for providing the QDOAS software. The GOME-2 data can be available from the University of Bremen. The Brewer data can be provided by the World Ozone and Ultraviolet Radiation Data Centre. We appreciate Florence Goutail for providing the SAOZ data. We gratefully
thank the Alfred Wegener Institute for providing ozonesonde data. We also gratefully thank ECMWF for providing the ERA5 data.

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



**Table 1. Fitting parameters of spectral retrieval.**

| Parameter | References |
|---|---|
| $O_3$ | 223K, 243K (Bogumil et al., 2003) |
| $O_4$ | 293K (Hermans et al., 2003) |
| Ring | Ring.exe |
| Fitting Interval | 320–340 nm |
| Polynomial | 5 |



**Table 2. The fitting parameter nodes for spectral retrieval.**

| Parameters | Nodes |
|---|---|
| SAZ (°) | 35, 40, 45, 50, 55, 60, 65, 70, 75, 80 |
| Surface albedo | 0.05, 0.1, 0.2, 0.3, 0.4, 0.5, 0.6 |
| Wavelength (nm) | From 320 to 340 in 0.5 intervals |



**Table 3. The number of days below $T_{nat}$ and daily average temperatures (December–February).**

| Date | Days below $T_{nat}$ | Average temperature (K) |
|---|---|---|
| 2016.12–2017.2 | 0 | 203.5 |
| 2017.12–2018.2 | 26 | 203.6 |
| 2018.12–2019.2 | 0 | 211.8 |
| 2019.12–2020.2 | 32 | 196.9 |
| 2020.12–2021.2 | 6 | 205.3 |



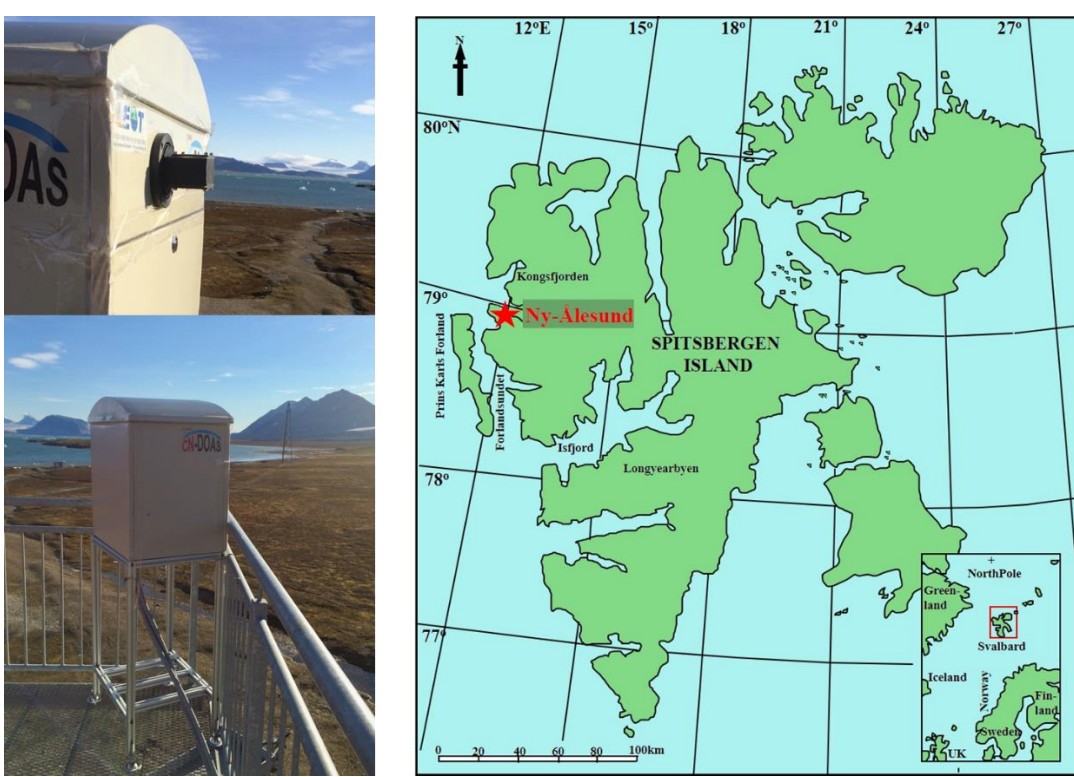

**Figure 1. The ground-based ZSL-DOAS instrument and experiment site in Ny-Ålesund.**



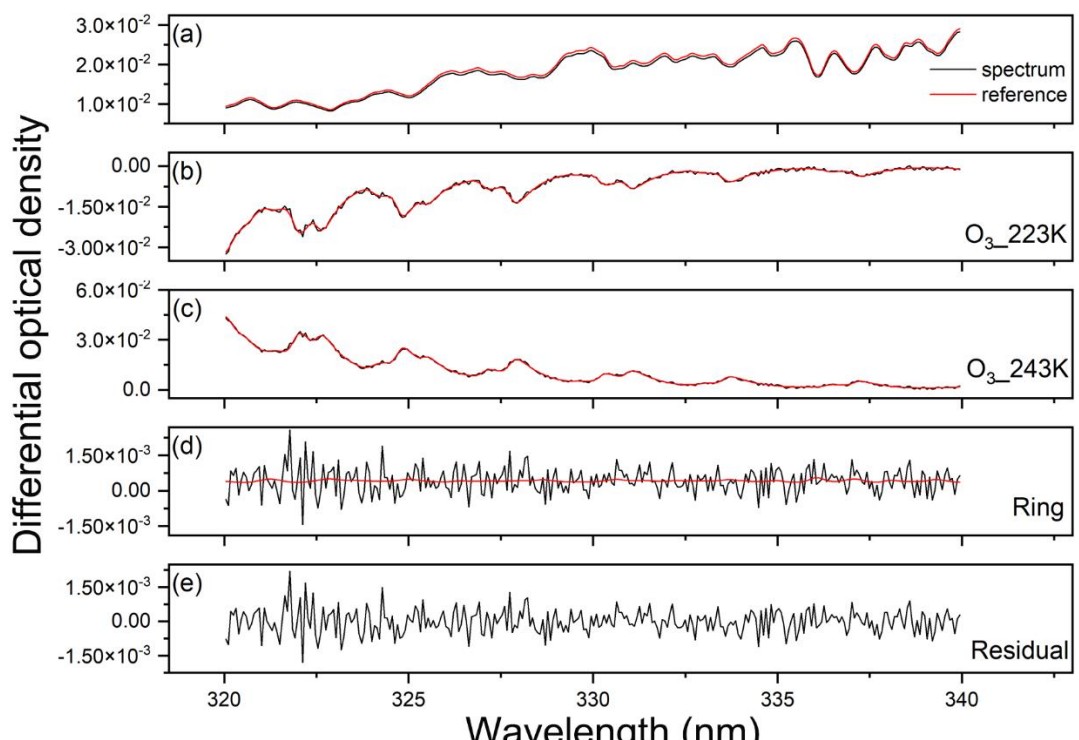

**Figure 2. Spectrum fits of ozone on June 13, 2021.**




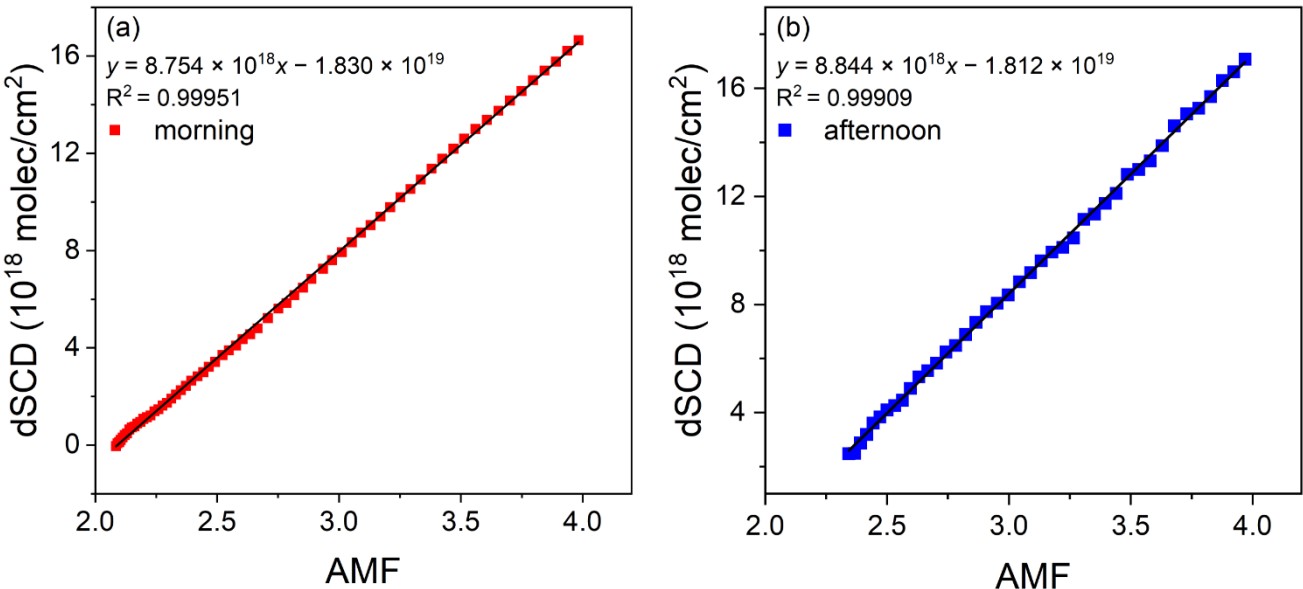

**Figure 3. Linear fit between ozone dSCDs and AMFs for the (a) morning and (b) afternoon on June 13, 2021. The correlation coefficients ($R^2$) are 0.99951 and 0.99909. The ozone VCDs for the morning and afternoon are $8.754 \times 10^{18}$ molec cm$^{-2}$ and $8.844 \times 10^{18}$ molec cm$^{-2}$. The calculated ozone VCD for June 13, 2021 is $8.799 \times 10^{18}$ molec cm$^{-2}$.**



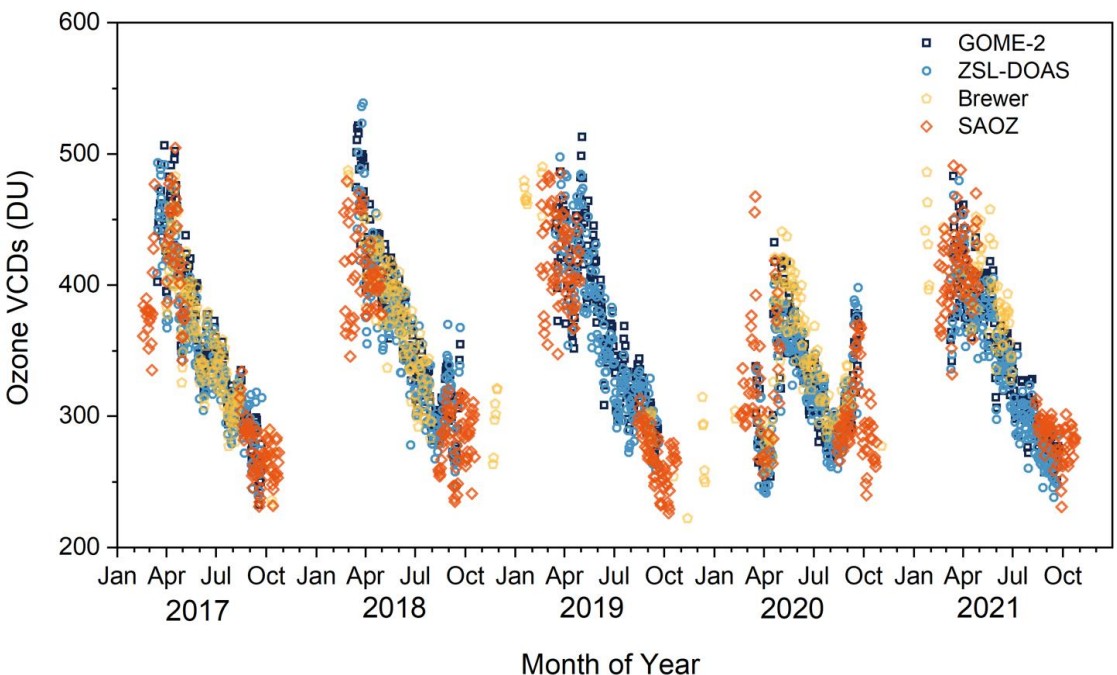

**Figure 4. The ozone VCDs from ZSL-DOAS, GOME-2, Brewer, and SAOZ.**





**Figure 5. Scatter plots and linear fits of retrieved ozone VCDs with (a) GOME-2, (b) Brewer, and (c) SAOZ.**





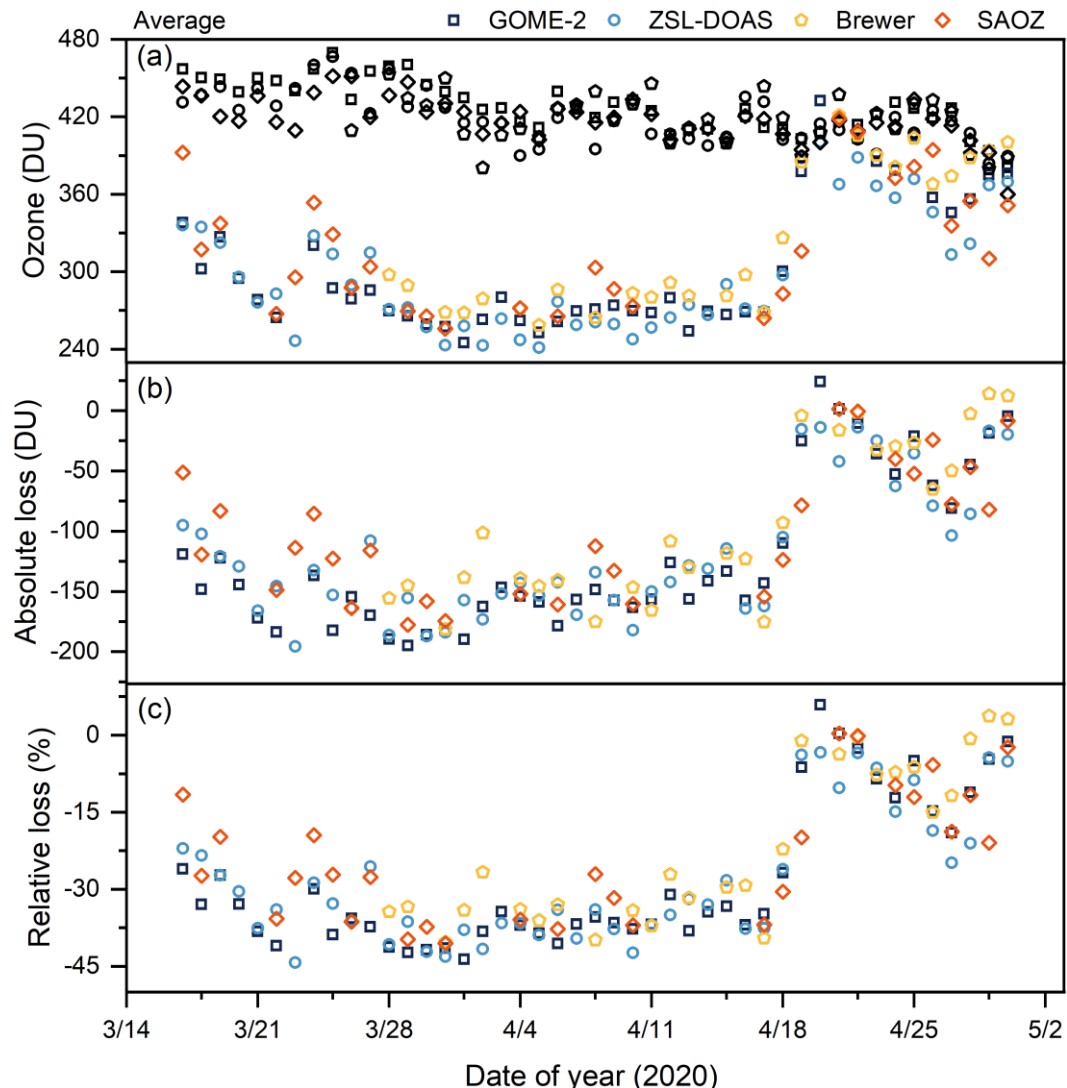

**Figure 6. (a) Ozone data for 2020 and the average ozone data (black) of 2017, 2018, 2019, and 2021. (b) Absolute and (c) relative**

**ozone loss for 2020.**





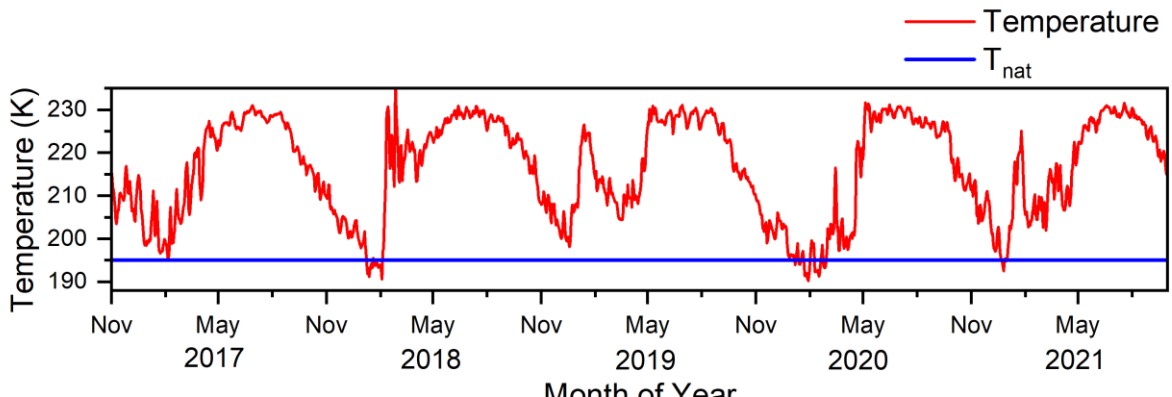

**Figure 7. Temperatures (at 70 hPa) over Ny-Ålesund from November 2016 to September 2021, where the blue line denotes the threshold temperature for the formation of PSCs.**





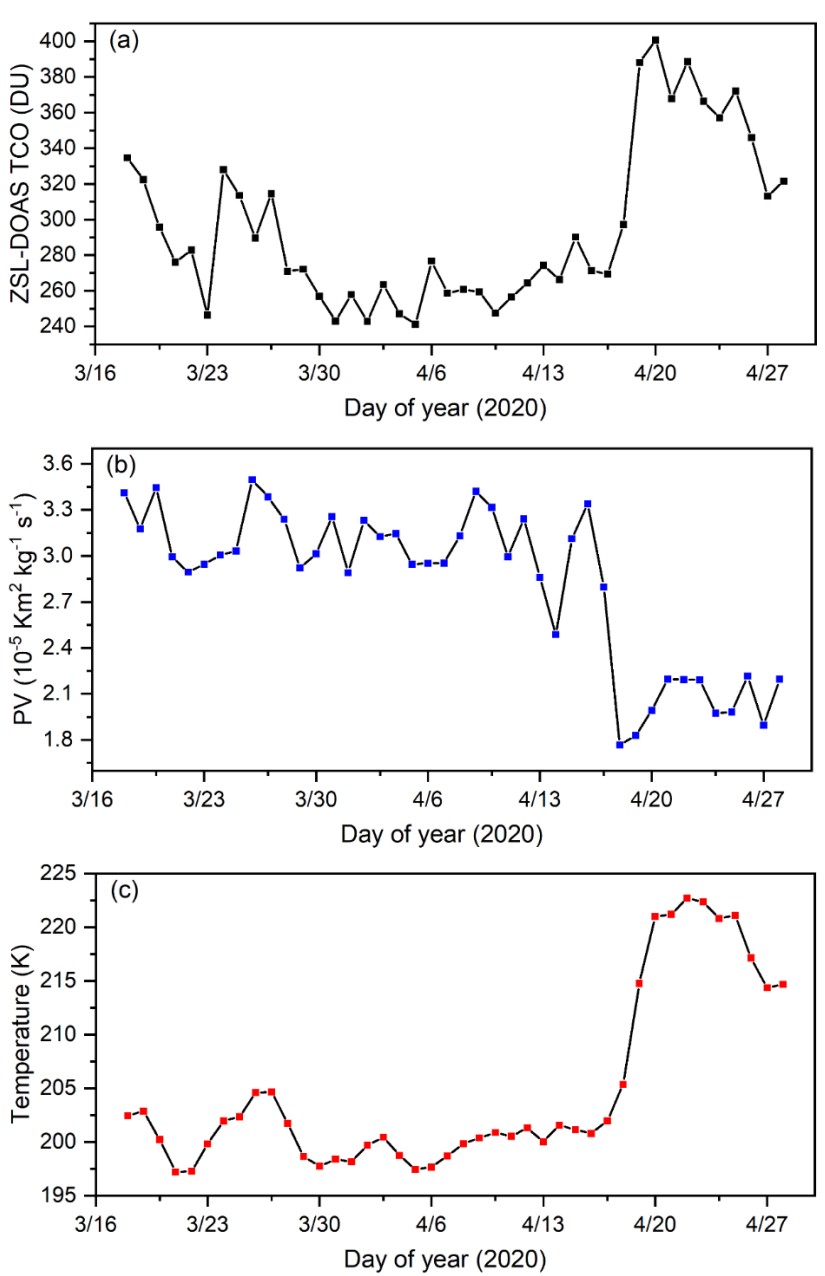

**Figure 8. From March 18 to April 28, 2020: (a) retrieved ozone VCDs, (b) PVs (at 70 hPa), and (c) temperatures (at 70 hPa).**



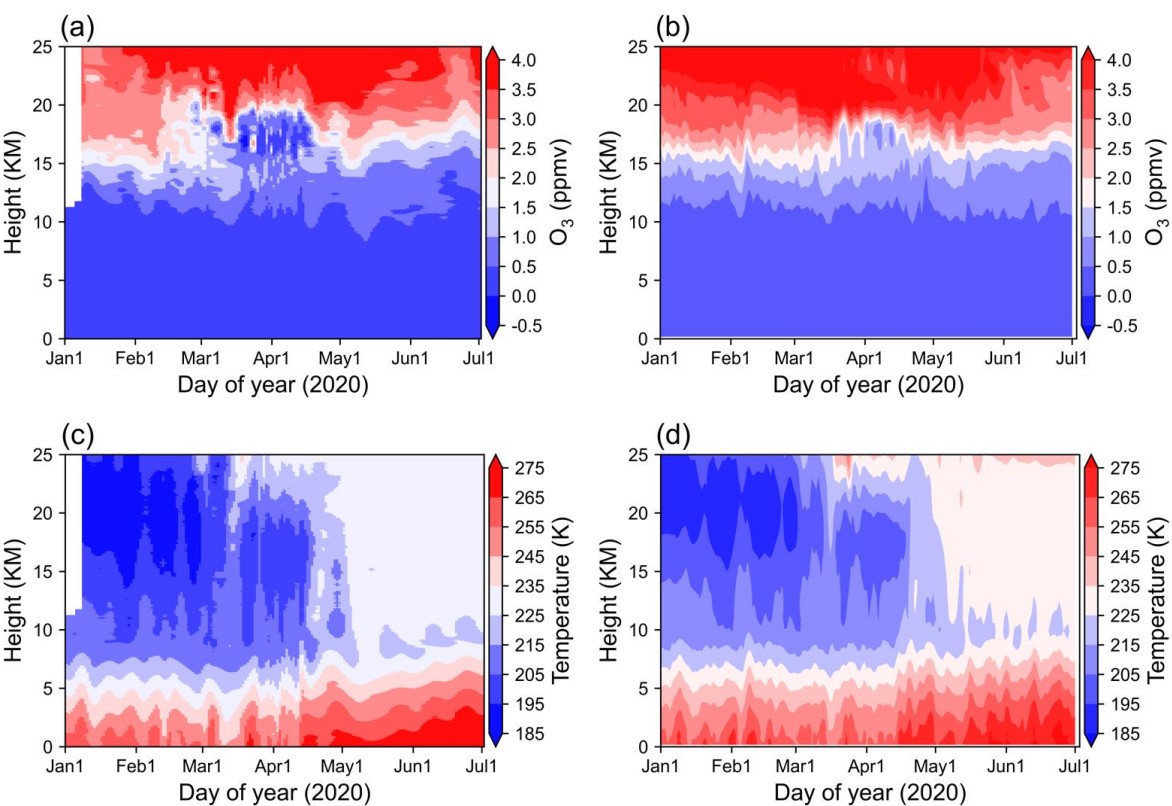

**Figure 9. Between January 1 and July 1, 2020, ozone profiles from (a) ozonesonde measurements and (b) the WACCM simulation, and temperature profiles from (c) ozonesonde measurements and (d) the WACCM simulation.**



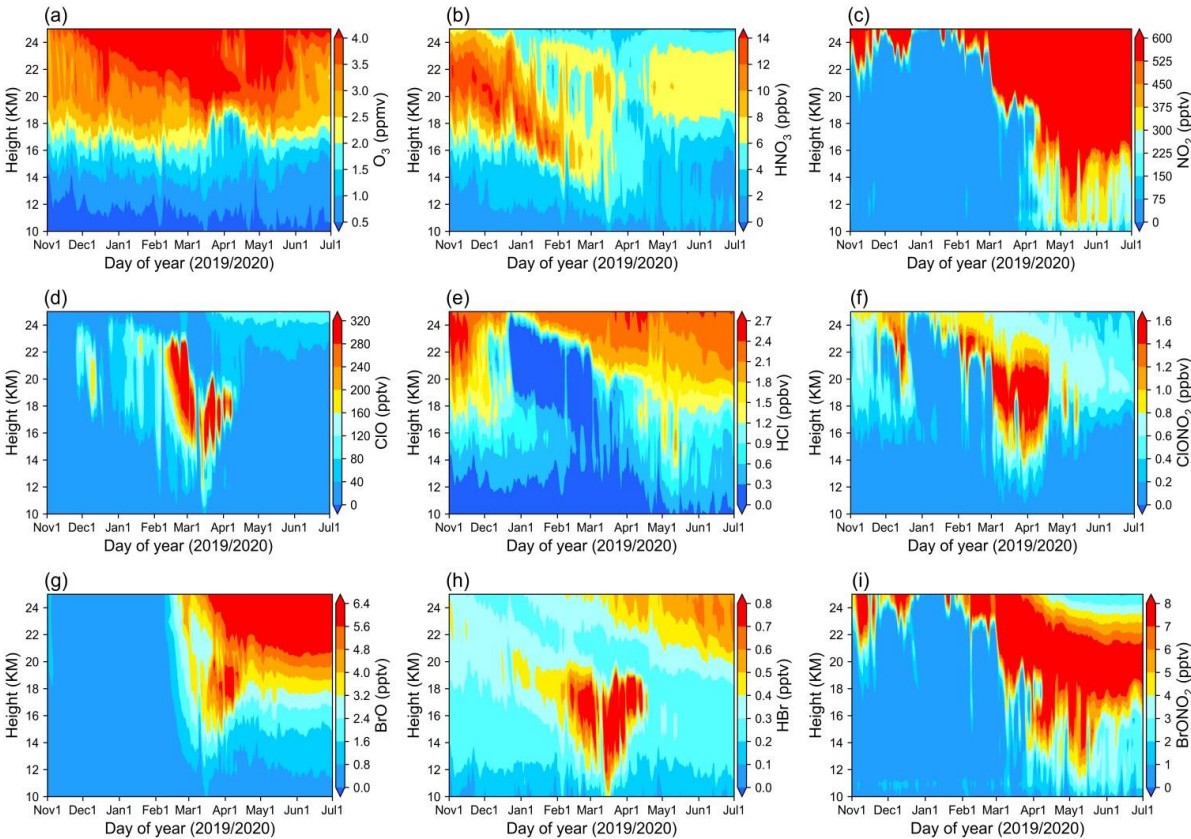

Figure 10. Simulated average diurnal profiles of the chlorine and bromine compounds between November 1, 2019, and July 1, 2020, at heights of 10–25 km above Ny-Ålesund: (a) $O_3$; (b) $HNO_3$; (c) $NO_2$; (d) ClO; (e) HCl; (f) $ClONO_2$; (g) BrO; (h) HBr; (i) $BrONO_2$.





**Figure 11. Simulated average diurnal mixing ratios of ozone, chlorine, and bromine compounds between November 1, 2019, and July 1, 2020, at a height of 17.5 km above Ny-Ålesund: (a) mixing ratios of ozone and chlorine ($Cl_t$ = ClO + HCl + HOCl + $ClONO_2$); (b) mixing ratios of ozone and bromine ($Br_t$ = BrO + HBr + HOBr + $BrONO_2$ + BrCl).**