# Peer review of "Research on the unusual spring 2020 Arctic stratospheric ozone depletion above Ny-Ålesund, Norway"

_Atmospheric Chemistry and Physics, 2022_

## Referee Comment (RC1)

**Review of "Research on the unusual spring 2020 Arctic stratospheric ozone depletion above Ny-Ålesund, Norway"**

BY QIDI LI ET AL.

**General**

I think this manuscript could make a contribution eventually, but it needs work. The reader cannot see clearly what is the main message of the paper (see also below). What is new? My understanding is the following: the first purpose is to introduce the new DOAS total ozone measurements at Ny-Ålesund at Yellow River Station. Then these measurements are used to investigate Arctic ozone loss in 2020. If the authors agree, then this point should come across much clearer in the manuscript. And the new DOAS instrument needs to be better described in the manuscript.

Given the fact that so much has been published on the Arctic winter 2019/2020 already (see also below), it might be more appropriate to present this work in ACP as a *Measurement Report*.

Further, the authors need to understand the background of the science they are reporting better. Some examples in detail and suggestions for improvement are given below. But as an obvious example: the authors report (on some occasions) the NAT temperature as $-195$ K – there are no negative values if temperature in measured in K. Overall, I think that the manuscript contains publishable material but I am afraid that restructuring and rewriting large parts of the manuscript are necessary.

**Comments**

**What are the main messages of the paper?**

First: the paper states that ozone VCD from a ground-based instrument, the GOME-2 satellite, and the Brewer and SAOZ instruments agree rather well. However, this is not a very new conclusion and had been discussed in many (mostly more technically oriented) papers before (e.g., León-Luis et al., 2018; Fioletov et al., 2002; Fioletov, 2002; Fioletov et al., 2005; Weber et al., 2005, and references therein).

Second, the paper reports that substantial ozone depletion occurred in the Arctic vortex until mid-April 2020, consistent with changes in simulated $HNO_3$. Again this is today not very new information; there is a special issue in JGR/GRL (and some of the papers on the Arctic winter 2020 in this special issue are cited/discussed in this manuscript) but there are a few more papers on Arctic ozone in 2020 in the meantime (e.g., von der Gathen et al., 2021; Kuttippurath et al., 2021; Ardra et al., 2022).

Third, ozone and temperature profiles were simulated by SD-WACCM, with these simulations corresponding well with ozonesonde measurements (but how well? – see below). The study used SD-WACCM with meteorological parameters driven by Modern Era Retrospective-Analysis for Research and Applications version 2 data; thus the simulation of temperature profiles by SD-WACCM is expected – isn't it? The fact that the ozone sonde measurements can be reproduced by the model is good but should be stated more clearly and in particular more quantitatively.

Finally the paper closes with the statement that "observations of ozone VCDs over Ny-Ålesund will continue in order to monitor future ozone changes over the area.". This is very good of course but not a conclusion from this paper.

**WACCM**

Some results of the paper rely on the model WACCM. But it is not clear how these results are obtained. I presume (although this is not stated in the paper) that openly available WACCM results have been used. If this is the case it should be clearly stated. If not, the WACCM runs conducted by the authors should be described (see also details) and then the WACCM version used should be clear. Also the way how the WACCM source code can be obtained should then be documented. Further, section 2.3 cites Kunz et al. (2011) – this is a good paper, but the paper does not deal with MERRA 2, so this sentence is confusing.

Further, which chemical scheme has been used in these simulation? I assume the most recent JPL recommendation (Burkholder et al., 2019). Müller et al. (1994, cited) emphasize the importance of $CH_3O_2$ + ClO for Arctic ozone loss – is this reaction taken into account in the WACCM simulation? More importantly, in which reference is the list of reactions described that is employed in the described chemical simulation? This information should be given in the paper. I also note that 'atmospheric simulations' are not mentioned in the author contribution. In general, it should be clear from the paper how the WACCM results were obtained.

**PSCs**

Clearly PSCs are important to polar ozone loss. However, first, one has to discriminate between PSC 'formation' and 'existence'. For crystalline particles (NAT and ice) this is not the same thing. (see e.g. Tritscher et al 2021). Also the temperature threshold for the onset of heterogeneous chemistry is not the same thing as NAT existence (Drdla and Müller, 2012, see also Tritscher 2021;Solomon1999, cited in the paper).

Further, denitrification by sedimenting NAT particles is touched upon in the paper. It is not straightforward implementing sedimentation in a model and explain the observations of large NAT particles in the atmosphere (e.g., Grooß et al., 2005; Molleker et al., 2014; Fahey et al., 2001; Tritscher et al., 2019). As simulated removal of $HNO_3$ in the paper is mentioned, the paper should give some information how NAT sedimentation is implemented in WACCM.

**Ozone from sondes and simulation**

Figure 9 (top) shows an important comparison, namely ozone sonde measurements against simulated ozone. However I suggest not showing the region below about 10 km, which is not of interest here (it also shows basically a blue area). But I think it is important to also show a plot of the differences (observations minus model) which would reveal that the model does not very well simulate to observed ozone depletion in March between 15 and 20 km. Further questions: what is the meaning of negative ozone mixing ratios (top)? WACCM seems to overestimate temperatures at about 25 km – is this a real effect?

**Formation of HCl**

The presented WACCM results suggest that the deactivation in the Arctic in 2021 is partly caused by formation of HCl, This is the classic deactivation pathway in the Antarctic, but not in the Arctic (e.g. Crutzen et al., 1992; Douglass et al., 1995; Müller et al., 2018). The authors might want to comment on this point.

**References**

Several references have been cited in this review; hopefully they are helpful. The point is not that the authors should feel obliged to cite these references. However, the paper cites WMO (2014); I suggest that a more recent ozone assessment should be used in the paper (WMO, 2018). The most recent (2022) assessment has just been released (https://ozone.unep.org/science/assessment/sap) and might be helpful when revising this paper.

**Data availability**

The data availability statement in this paper is not good. I suggest making the DOAS observations at Ny-Ålesund available for download on a server that issues a doi and where the data are permanently archived. Such links are reported for (e.g.) SAOZ but not for the DOAS measurements presented in the paper. Further, the WACCM data need to be better described (see above). Making data available through e-mail request is no longer recommended.

**Details**

- Title: I suggest avoiding "Research on" in the title; isn't this obvious? The title should rather reflect the fact that DOAS measurements from Ny-Ålesund are reported here.

- p. 1, l. 16: why this period? (I think this is the period when measurements are available, but this should be clear from the paper).

- p 1, l 21: what is a "normal year" in the Arctic?

- p. 1, l. 21: 44.3 % $\longrightarrow$ here and elsewhere in the paper: add an error estimate for the ozone loss.

- p. 1, l. 23: here ans elsewhere: PV and ozone depletion: is this only a complicated way of saying that there is no ozone loss outside the vortex? I think that Ny-Ålesund was located outside the vortex at about April 16 (see also Fig. 8.

- p. 1, l. 26: how new is the peak in ClO (chlorine activation)? Compare the papers in the JGR/GRL special issue?

- p. 2, l. 38: this is not a good description of halogen induced polar ozone loss (e.g., Müller et al., 2018, and Solomon 1999, Tritscher 2021, cited in the paper).

- p 2., l. 42: 'recovery' is an important issue, it is different in the polar regions and in mid.latitudes (WMO, 2018). See also further papers on the recovery of both the Antarctic ozone hole and global ozone levels (e.g., Kuttippurath and Nair, 2017; Strahan and Douglass, 2018; WMO, 2018; Bodeker and Kremser, 2021; Stone et al., 2021; Weber et al., 2022).

- p. 2, l. 49: there should be more citations here than just Hu 2020.

- p. 3, l. 75: Simpson is on boundary layer issues: this reference needs to be changed. There are several alternative citations, already cited in the paper and there are further modelling papers cited in this review.

- p. 3, l. 75: These citations focus on one particular model (CLaMS), which is okay. But I think you should have citations to other models here as well (e.g., Chipperfield, 1999; Khosrawi et al., 2009; Bekki et al., 2013; Chipperfield et al., 1994; Kinnison et al., 2007; Wohltmann and Rex, 2009; Wohltmann et al., 2010, []).

- p. 4, l. 99: You cannot start the Methods section with "the DOAS instrument". Which instrument? I think it is a new instrument that is described below – correct? This should be much clearer from the paper and the instrument needs to be described first before it can be "placed" somewhere. Further, given the fact that the DOAS technique is so prominent here (or should be) a bit more background on DOAS and citations (see perhaps, Hüneke et al., 2017) might be appropriate.

- p. 6, p. 140: this sentence starts with 'parameters' but the paper should state what was actually done regarding WACCM.

- p. 7, l. 166: ERA5 has 137 layers – is there a typo here?

- p. 7, l. 167: where have these measurements been done?

- p. 8, l. 173: what is a 'normal year'?

- p. 9, l. 199: what is the 'threshold temperature'? This is an important point that should be discussed in the paper.

- p. 9, l. 201: by definition the PV in the southern hemisphere is negative and positive in the northern hemisphere. This simple fact should be taken into account when making such statements.

- p. 10., l. 235: apparent $\longrightarrow$ obvious?

- p. 10., l. 243: 'recover' is problematic here, it is not the right word to use when taking about chlorine deactivation putting a halt to ozone loss.

- p. 10., l. 237: it is not only the reaction $HCl + ClONO_2$

- Figure 5: cold the errors of the individual measurements be used for weighting the data when calculating regression etc?

- Figure 6: Show error bars?

- Figure 7: the blue line shows 195 K, which is an approximation for the onset temperature for heterogeneous chemistry.

- Figure 8: show error bars?

**References**

Ardra, D., Kuttippurath, J., Roy, R., Kumar, P., Raj, S., Müller, R., and Feng, W.: The unprecedented ozone loss in the Arctic winter and spring of 2010/2011 and 2019/2020, ACS Earth and Space Chemistry, sp-2021-003338, https://doi.org/10.1021/acsearthspacechem.1c00333, 2022.

Bekki, S., Rap, A., Poulain, V., Dhomse, S., Marchand, M., Lefevre, F., Forster, P., Szopa, S., and Chipperfield, M.: Climate impact of stratospheric ozone recovery, Geophys. Res. Lett., 40, 2796–2800, https://doi.org/10.1002/grl.50358, 2013.

Bodeker, G. E. and Kremser, S.: Indicators of Antarctic ozone depletion: 1979 to 2019, Atmos. Chem. Phys., 21, 5289–5300, URL https://doi.org/10.5194/acp-21-5289-2021, 2021.

Burkholder, J. B., Sander, S. P., Abbatt, J. P. D., Barker, J. R., Cappa, C., Crounse, J. D., Dibble, T. S., Huie, R. E., Kolb, C. E., Kurylo, M. J., Orkin, V. L., Percical, C. J., Wilmouth, D. M., and Wine, P. H.: Chemical kinetics and photochemical data for use in atmospheric studies, Evaluation Number 19, JPL Publication 19-5, URL http://jpldataeval.jpl.nasa.gov, 2019.

Chipperfield, M. P.: Multiannual simulations with a three-dimensional chemical transport model, J. Geophys. Res., 104, 1781–1805, 1999.

Chipperfield, M. P., Cariolle, D., and Simon, P.: A 3D transport model study of chlorine activation during EASOE, Geophys. Res. Lett., 21, 1467–1470, https://doi.org/10.1029/93GL01679, 1994.

Crutzen, P. J., Müller, R., Brühl, C., and Peter, T.: On the potential importance of the gas phase reaction $CH_3O_2 + ClO \rightarrow ClOO + CH_3O$ and the heterogeneous reaction $HOCl + HCl \rightarrow H_2O + Cl_2$ in "ozone hole" chemistry, Geophys. Res. Lett., 19, 1113–1116, https://doi.org/10.1029/92GL01172, 1992.

Douglass, A. R., Schoeberl, M. R., Stolarski, R. S., Waters, J. W., Russell III, J. M., Roche, A. E., and Massie, S. T.: Interhemispheric differences in springtime production of HCl and $ClONO_2$ in the polar vortices, J. Geophys. Res., 100, 13 967–13 978, https://doi.org/10.1029/95JD00698, 1995.

Drdla, K. and Müller, R.: Temperature thresholds for chlorine activation and ozone loss in the polar stratosphere, Ann. Geophys., 30, 1055–1073, https://doi.org/10.5194/angeo-30-1055-2012, 2012.

Fahey, D. W., Gao, R. S., Carslaw, K. S., Kettleborough, J., Popp, P. J., Northway, M. J., Holecek, J. C., Ciciora, S. C., McLaughlin, R. J., Thompson, T. L., Winkler, R. H., Baumgardner, D. G., Gandrud, B., Wennberg, P. O., Dhaniyala, S., McKinley, K., Peter, T., Salawitch, R. J., Bui, T. P., Elkins, J. W., Webster, C. R., Atlas, E. L., Jost, H., Wilson, J. C., Herman, R. L., Kleinböhl, A., and von König, M.: The detection of large $HNO_3$-containing particles in the winter Arctic stratosphere, Science, 291, 1026–1031, 2001.

Fioletov, V. E.: Comparison of Brewer ultraviolet irradiance measurements with total ozone mapping spectrometer satellite retrieval, Opt. Eng., 41, 3051, URL `ttps://doi.org/10.1117/1.1516818`, 2002.

Fioletov, V. E., Bodeker, G. E., Miller, A. J., McPeters, R. D., and Stolarski, R.: Global and zonal total ozone variations estimated from ground-based and satellite measurements: 1964–2000, J. Geophys. Res., 107, 4647, https://doi.org/10.1029/2001JD001350, 2002.

Fioletov, V. E., Kerr, J., McElroy, C., Wardle, D., Savastiouk, V., and Grajnar, T.: The Brewer reference triad, Geophys. Res. Lett., 32, L20805, URL `https://doi.org/10.1029/2005GL024244`, 2005.

Grooß, J.-U., Günther, G., Müller, R., Konopka, P., Bausch, S., Schlager, H., Voigt, C., Volk, C. M., and Toon, G. C.: Simulation of denitrification and ozone loss for the Arctic winter 2002/2003, Atmos. Chem. Phys., 5, 1437–1448, 2005.

Hüneke, T., Aderhold, O.-A., Bounin, J., Dorf, M., Gentry, E., Grossmann, K., Grooß, J.-U., Hoor, P., Jöckel, P., Kenntner, M., Knapp, M., Knecht, M., Lörks, D., Ludmann, S., Matthes, S., Raecke, R., Reichert, M., Weimar, J., Werner, B., Zahn, A., Ziereis, H., and Pfeilsticker, K.: The novel HALO mini-DOAS instrument: inferring trace gas concentrations from airborne UV/visible limb spectroscopy under all skies using the scaling method, Atmos. Meas. Tech., 10, 4209–4234, https://doi.org/10.5194/amt-10-4209-2017, 2017.

Khosrawi, F., Müller, R., Proffitt, M. H., Ruhnke, R., Kirner, O., Jöckel, P., Grooß, J.-U., Urban, J., Murtagh, D., and Nakajima, H.: Evaluation of CLaMS, KASIMA and ECHAM5/MESSy1 simulations in the lower stratosphere using observations of Odin/SMR and ILAS/ILAS-II, Atmos. Chem. Phys., 9, 5759–5783, 2009.

Kinnison, D. E., Brasseur, G. P., Walters, S., Garcia, R. R., Sassi, D. R. M. F., Harvey, V. L., Randall, C. E., Emmons, L., Lamarque, J. F., Hess, P., Orlando, J. J., Tie, X. X., Randel, W., Pan, L. L., Gettelman, A., Granier, C., Diehl, T., Niemeier, U., and Simmons, A. J.: Sensitivity of chemical tracers to meteorological parameters in the MOZART-3 chemical transport model, J. Geophys. Res., 112, https://doi.org/10.1029/2006JD007879, 2007.

Kuttippurath, J. and Nair, P. J.: The signs of Antarctic ozone hole recovery, Sci. Rep., 7, 585, https://doi.org/10.1038/s41598-017-00722-7, 2017.

Kuttippurath, J., Feng, W., Müller, R., Kumar, P., Raj, S., Gopikrishnan, G. P., and Roy, R.: Exceptional loss in ozone in the Arctic winter/spring 2020, Atmos. Chem. Phys., accepted, 2021.

León-Luis, S. F., Redondas, A., Carreño, V., López-Solano, J., Berjón, A., Hernández-Cruz, B., and Santana-Díaz, D.: Internal consistency of the Regional Brewer Calibration Centre for Europe triad during the period 2005–2016, Atmos. Meas. Tech., 11, 4059–4072, https://doi.org/10.5194/amt-11-4059-2018, 2018.

Molleker, S., Borrmann, S., Schlager, H., Luo, B., Frey, W., Klingebiel, M., Weigel, R., Ebert, M., Mitev, V., Matthey, R., Woiwode, W., Oelhaf, H., Dörnbrack, A., Stratmann, G., Grooß, J.-U., Günther, G., Vogel, B., Müller, R., Krämer, M., Meyer, J., and Cairo, F.: Microphysical properties of synoptic-scale polar stratospheric clouds: in situ measurements of unexpectedly large $HNO_3$-containing particles in the Arctic vortex, Atmospheric Chemistry and Physics, 14, 10 785–10 801, https://doi.org/10.5194/acp-14-10785-2014, 2014.

Müller, R., Grooß, J.-U., Zafar, A. M., Robrecht, S., and Lehmann, R.: The maintenance of elevated active chlorine levels in the Antarctic lower stratosphere through HCl null cycles, Atmos. Chem. Phys., 18, 2985–2997, https://doi.org/10.5194/acp-18-2985-2018, 2018.

Stone, K. A., Solomon, S., Kinnison, D. E., and Mills, M. J.: On Recent Large Antarctic Ozone Holes and Ozone Recovery Metrics, Geophys. Res. Lett., 48, https://doi.org/10.1029/2021GL095232, 2021.

Strahan, S. E. and Douglass, A. R.: Decline in Antarctic Ozone Depletion and Lower Stratospheric Chlorine Determined From Aura Microwave Limb Sounder Observations, Geophys. Res. Lett., 45, 382–390, https://doi.org/https://doi.org/10.1002/2017GL074830, 2018.

Tritscher, I., Grooß, J.-U., Spang, R., Pitts, M. P., Poole, L. R., Müller, R., and Riese, M.: Lagrangian simulation of ice particles and resulting dehydration in the polar winter stratosphere, Atmos. Chem. Phys., 19, 543–563, https://doi.org/10.5194/acp-19-543-2019, 2019.

von der Gathen, P., Kivi, R., Wohltmann, I., Salawitch, R. J., and Rex, M.: Climate change favours large seasonal loss of Arctic ozone, Nat. Commun., 12, 3886, https://doi.org/10.1038/s41467-021-24089-6, 2021.

Weber, M., Lamsal, L. N., Coldewey-Egbers, M., Bramstedt, K., and Burrows, J. P.: Pole-to-pole validation of GOME WFDOAS total ozone with groundbased data, Atmos. Chem. Phys., 5, 1341–1355, 2005.

Weber, M., Arosio, C., Coldewey-Egbers, M., Fioletov, V. E., Frith, S. M., Wild, J. D., Tourpali, K., Burrows, J. P., and Loyola, D.: Global total ozone recovery trends attributed to ozone-depleting substance (ODS) changes derived from five merged ozone datasets, Atmos. Chem. Phys., 22, 6843–6859, https://doi.org/10.5194/acp-22-6843-2022, 2022.

WMO: Scientific assessment of ozone depletion: 2014, Global Ozone Research and Monitoring Project–Report No. 55, Geneva, Switzerland, 2014.

WMO: Scientific assessment of ozone depletion: 2018, Global Ozone Research and Monitoring Project–Report No. 58, Geneva, Switzerland, 2018.

Wohltmann, I. and Rex, M.: The Lagrangian chemistry and transport model ATLAS: validation of advective transport and mixing, Geosci. Model Dev., 2, 153–173, https://doi.org/10.5194/gmd-2-153-2009, 2009.

Wohltmann, I., Lehmann, R., and Rex, M.: The Lagrangian chemistry and transport model ATLAS: simulation and validation of stratospheric chemistry and ozone loss in the winter 1999/2000, Geosci. Model Dev., 3, 585–601, https://doi.org/10.5194/gmd-3-585-2010, 2010.

---

## Author Comment (AC1)

**Response to Reviewer #1**

**General**

I think this manuscript could make a contribution eventually, but it needs work. The reader cannot see clearly what the main message of the paper is (see also below). What is new? My understanding is the following: the first purpose is to introduce the new DOAS total ozone measurements at Ny-Ålesund at Yellow River Station. Then these measurements are used to investigate Arctic ozone loss in 2020. If the authors agree, then this point should come across much clearer in the manuscript. And the new DOAS instrument needs to be better described in the manuscript. Given the fact that so much has been published on the Arctic winter 2019/2020 already (see also below), it might be more appropriate to present this work in ACP as a Measurement Report. Further, the authors need to understand the background of the science they are reporting better. Some examples in detail and suggestions for improvement are given below. But as an obvious example: the authors report (on some occasions) the NAT temperature as −195 K – there are no negative values if temperature in measured in K. Overall, I think that the manuscript contains publishable material but I am afraid that restructuring and rewriting large parts of the manuscript are necessary.

**Author's Response:**

We would like to thank the reviewer #1 for the careful and valuable comments, which enable us to improve our study and the manuscript remarkably. Please kindly find our point-to-point response to the problems/comments below in blue and the change of the manuscript in orange.

We agreed to present this work as a Measurement Report, in which the measurements are reported and the consistency with other studies and measurements are shown. We focused on introducing the new DOAS total ozone measurements at Ny-Ålesund at Yellow River Station and then used these measurements to study the Arctic ozone loss in 2020. In addition, the new DOAS instrument was further described in the revised manuscript. Please see P6 lines 135–145. The temperature threshold for the existence of NAT as 195K has been revised.

"The ZSL-DOAS instrument mainly includes the prism, telescope, computer, filter, motor, and CCD spectrometer. The motor controlled the telescope that can change the angle of elevation between the horizon and the zenith. As the angle of elevation changes, the telescope can acquire scattered sunlight at different angles (2°, 3°, 4°, 6°, 8°, 10°, 15°, 30°, and 90°). The quartz fibre can transform the incident light and its numerical aperture is 0.22. The light is received by the spectrometer (Ocean Optics MAYA pro) and measured by a 2048 pixels CCD. This spectrometer was designed for wavelengths between 290 and 429 nm, and had the spectral resolution (FWHM) of 0.5 nm. The integration time varied between 100 and 2000 ms due to the light intensity. The detector operates normally at approximately 20°C with a thermal controller. The mercury lamp spectra, offsets and dark currents were calibrated ahead of the experiments. The ZSL-DOAS instrument can detect $O_3$, $NO_2$, $OClO$, $BrO$, and $O_4$. The ozone slant column density (SCD) was retrieved, with the raw data obtained in the zenith direction (90°). The ZSL-DOAS instrument was placed at the Yellow River Station (78.92° N, 11.93° E)

in the Arctic. Figure 1 shows the ZSL-DOAS instrument and experimental location, in Ny-Ålesund, Svalbard, Norway."

**Comments**

**A) What are the main messages of the paper?**
First: the paper states that ozone VCD from a ground-based instrument, the GOME-2 satellite, and the Brewer and SAOZ instruments agree rather well. However, this is not a very new conclusion and had been discussed in many (mostly more technically oriented) papers before (e.g., Léon-Luis et al., 2018; Fioletov et al., 2002; Fioletov, 2002; Fioletov et al., 2005; Weber et al., 2005, and references therein). Second, the paper reports that substantial ozone depletion occurred in the Arctic vortex until mid-April 2020, consistent with changes in simulated HNO3. Again this is today not very new information; there is a special issue in JGR/GRL (and some of the papers on the Arctic winter 2020 in this special issue are cited/discussed in this manuscript) but there are a few more papers on Arctic ozone in 2020 in the meantime (e.g., von der Gathen et al., 2021; Kuttippurath et al., 2021; Ardra et al., 2022). Third, ozone and temperature profiles were simulated by SD-WACCM, with these simulations corresponding well with ozonesonde measurements (but how well? – see below). The study used SD-WACCM with meteorological parameters driven by Modern Era Retrospective-Analysis for Research and Applications version 2 data; thus the simulation of temperature profiles by SD-WACCM is expected – isn't it? The fact that the ozone sonde measurements can be reproduced by the model is good but should be stated more clearly and in particular more quantitatively. Finally the paper closes with the statement that "observations of ozone VCDs over Ny-Ålesund will continue in order to monitor future ozone changes over the area." This is very good of course but not a conclusion from this paper.

**Author's Response:**
Thanks for the reviewer's advices. Ozone VCDs from a ground-based instrument, the GOME-2 satellite, and the Brewer and SAOZ instruments agree rather well and substantial ozone depletion occurred in the Arctic vortex until mid-April 2020, consistent with changes in simulated HNO$_3$. The reviewer is correctly saying that these are not very new information today. Thus, we have presented this work as a Measurement Report, in which the measurements are reported and the consistency with other studies and measurements are shown.

The simulation of temperature profiles by SD-WACCM indeed corresponded well with ozonesonde measurements, and this can be used to validate the simulation. The temporal resolution of the sounding data from March 25 to April 13, 2020, is once per day, whereas the others are normally once per 3 d during the spring and once per week during the other seasons. The other missing days were obtained by interpolation, so we did not show a plot of the differences (observations minus model).

The sentence has been revised. Please see P17 lines 408–414.

"In summary, by ZSL-DOAS observations, we provided another evidence for unprecedented ozone depletion during the Arctic spring of 2020. The ZSL-DOAS ozone VCD observations can also provide calibration for satellite observations and model simulations, and in the future

can provide the support for observations at more Chinese research stations or international local stations in the polar area. Additionally, although WACCM can depict the evolution of ozone during this Arctic ozone depletion event, there are some problems such as overestimation of the temperature and the $CH_3O_2+ClO$ reaction is not considered in the current chemical mechanism of the model. This could be considered in future models to improve the simulation performance.”

**B) WACCM**
**B1)** Some results of the paper rely on the model WACCM. But it is not clear how these results are obtained. I presume (although this is not stated in the paper) that openly available WACCM results have been used. If this is the case it should be clearly stated. If not, the WACCM runs conducted by the authors should be described (see also details) and then the WACCM version used should be clear.

**Author's Response:**
    Thanks for the reviewer's suggestion. We have rewritten the description of model setting in section 2.2 of the revised manuscript. Please see P8 and P9, lines 186–222.

“The physical parameterizations employed in the Community Atmosphere Model Version 4 (CAM4) were applied to the WACCM (Neale et al., 2013). At present, the WACCM model is incorporated into a component set of the Community Earth System Model, whose source code is available online (https://svn-ccsm-release.cgd.ucar.edu/model versions/). The Model for Ozone and Related Chemical Tracers, version 3 (MOZART-3) provided the chemical parameters for the WACCM (Kinnison et al., 2007). This mechanism contains 52 neutral species, one invariant ($N_2$), 127 neutral gas-phase reactions, 48 neutral photolytic reactions, and 17 heterogeneous reactions [see Tables 5.1-5.5 in Neale et al. (2013)]. The chemical mechanism of WACCM4 also contains 4 aerosol types heterogeneous reactions: liquid binary sulfate (LBS), supercooled ternary solution (STS), nitric acid trihydrate (NAT), and water-ice. When model temperatures above 200K, only the LBS exists. The surface area density (SAD) of LBS is from SAGE, SAGE-II and SAMS observations (Thomason et al., 1997) and Considine update it (World Meteorological Organization, 2003). With the model atmosphere cooling, the LBS aerosol expands and absorbs both $HNO_3$ and $H_2O$ to obtain the STS aerosol. Tabazadeh et al. (1994) derived the composition of STS by the Aerosol Physical Chemistry Model (ACPM). The STS aerosol median radius and SAD is derived following the approach of Considine et al. (2000). When model temperatures reach a specified supersaturation ratio of $HNO_3$ for NAT, $HNO_3$ containing aerosols are allowed to form. In WACCM4, Peter et al. (1991) set this ratio to 10. NAT median radius and SAD are derived in the same way with STS aerosol. If the derived atmospheric temperature does not exceed the saturation temperature of water vapour on ice ($T_{sat}$), then this results in the formation of water-ice aerosols. In WACCM4, the CAM's prognostic water routines gives the condensed phase $H_2O$, which is conveyed to the chemistry module. According to the method of Considine et al. (2000), the median radius and SAD of water-ice can be derived by this condensed phase $H_2O$. The polar stratospheric cloud module used in this study followed Wegner et al. (2013) rather than the standard module of Kinnison et al. (2007), improving the capabilities of WACCM in modelling ozone and its

associated components (Brakebusch et al., 2013). The sedimentation of HNO$_3$ in NAT aerosol follows the approach in Considine et al. (2000). The flux (F) of HNO$_3$ can be derived as follows:

$$F = V \cdot C \cdot \exp(8\ln^2\sigma). \tag{4}$$

here $V$ represents the terminal velocity of NAT aerosol, $C$ denotes the condensed-phase concentration of HNO$_3$, $\sigma$=1.6 (Dye et al., 1992) represents the width of the lognormal size distribution for NAT.

We used the SD-WACCM with meteorological parameters driven by Modern Era Retrospective-Analysis for Research and Applications version 2 (MERRA-2) data (Gelaro et al., 2017). The SD-WACCM had the horizontal resolution of 1.9° × 2.5° (lat × lon). The model was divided vertically into 88 layers, covering an altitude of ~140 km from the ground to the bottom of the lower thermosphere layer. Meteorological fields were calculated using a nudging method in the model (Lamarque et al., 2012). Data for the horizontal winds, temperature, and surface pressure from MERRA-2 were used to drive the physical parameterization from the surface to 50 km (Kunz et al., 2011), which allowed for more accurate comparisons between the measurements of atmospheric composition and the model output (Lamarque et al., 2012). This can be employed for the study of specific weather events. Linear transitions were used in the 50–60 km altitude range and over 60 km, and online calculations were performed. In this study, the MERRA-2 dataset has the same resolution with the SD-WACCM, which can be accessed on the Earth System Grid (https://www.earthsystemgrid.org/home.html) and are obtained from the original resolution (1/2°×2/3°) by a conservative re-gridding procedure (Lamarque et al., 2012; Pan et al., 2019). In this study, the simulation is initiated between November 1, 2019, and July 1, 2020."

**B2)** Also the way how the WACCM source code can be obtained should then be documented. Further, section 2.3 cites Kunz et al. (2011) – this is a good paper, but the paper does not deal with MERRA 2, so this sentence is confusing.

**Author's Response:**

The WACCM is a component set of CESM. And the CESM code is available online (https://svn-ccsm-release.cgd.ucar.edu/model_versions/). Similar sentences have been mentioned in P8 lines 187–188 of the revised manuscript.

We have rewritten the description of the nudging method used in SD-WACCM. Please see P9 lines 215–219.

" Data for the horizontal winds, temperature, and surface pressure from MERRA-2 were used to drive the physical parameterization from the surface to 50 km (Kunz et al., 2011), which allowed for more accurate comparisons between the measurements of atmospheric composition and the model output (Lamarque et al., 2012). This can be employed for the study of specific weather events. Linear transitions were used in the 50–60 km altitude range and over 60 km, and online calculations were performed."

**B3)** Further, which chemical scheme has been used in these simulations? I assume the most recent JPL recommendation (Burkholder et al., 2019).

**Author's Response:**

    The basic chemistry mechanism in the WACCM is taken from the MOZART-3. Please see P8 lines 188–190.

"The Model for Ozone and Related Chemical Tracers, version 3 (MOZART-3) provided the chemical parameters for the WACCM (Kinnison et al., 2007)."

**B4)** Müller et al. (1994, cited) emphasize the importance of $CH_3O_2 + ClO$ for Arctic ozone loss – is this reaction taken into account in the WACCM simulation?

**Author's Response:**

    This reaction is not included in the WACCM model. Müller et al. (1994) found that the $CH_3O_2+ClO$ reaction is also important but is not included in the current chemical mechanism of the model and could be taken into account in future models to improve simulation performance. Please see P14, lines 325–327.

"On the other hand, Müller et al. (1994) found that the $CH_3O_2+ClO$ reaction is also important but is not included in the current chemical mechanism of the model and could be taken into account in future models to improve simulation performance."

**B5)** More importantly, in which reference is the list of reactions described that is employed in the described chemical simulation? This information should be given in the paper.

**Author's Response:**

    Thank for your suggestion. Please see P8, lines 190–191.

"This mechanism contains 52 neutral species, one invariant ($N_2$), 127 neutral gas-phase reactions, 48 neutral photolytic reactions, and 17 heterogeneous reactions [see Tables 5.1-5.5 in Neale et al. (2013)]."

**B6)** I also note that 'atmospheric simulations' are not mentioned in the author contribution. In general, it should be clear from the paper how the WACCM results were obtained.

**Author's Response:**

    The WACCM simulation was conducted by Chen Pan. We have added statements in the author contribution. We also rewritten the description of the model settings. Please see section 2.2 in the revised manuscript.

**C) PSCs**

**C1)** Clearly PSCs are important to polar ozone loss. However, first, one has to discriminate between PSC 'formation' and 'existence'. For crystalline particles (NAT and ice) this is not the same thing. (see e.g. Tritscher et al 2021). Also the temperature threshold for the onset of heterogeneous chemistry is not the same thing as NAT existence (Drdla and Müller, 2012, see also Tritscher 2021;Solomon1999, cited in the paper).

**Author's Response:**

Thanks for the reviewer's suggestions. Please see P3, lines 68–76.

"Polar stratospheric clouds (PSCs) are classified into three types: nitric acid trihydrate (NAT), ice PSCs, and supercooled ternary solution (STS), and their threshold temperatures for existence are $T_{nat}$ (195 K), $T_{ice}$ (188 K), and $T_{sts}$ (195–197 K), respectively (Toohey et al., 1993; Poole and McCormick, 1988; Solomon, 1999). Extremely low air temperatures are essential to produce PSC. The PSC can be used as a surface for heterogeneous interactions, leading to the conversion of reactive halogens from the halogen reservoirs, which can cause serious ozone loss (Frieβ et al., 2005; Marsing et al., 2019). Although the PSC is not only composed of NAT (Pitts et al. 2009; Spang et al. 2018), the temperature threshold for the existence of NAT provides a good estimate on the occurrence of heterogeneous chemistry (Drdla and Müller 2012; Kirner et al. 2015; Grooß and Müller 2021; von der Gathen et al. 2021)."

**C2)** Further, denitrification by sedimenting NAT particles is touched upon in the paper. It is not straightforward implementing sedimentation in a model and explain the observations of large NAT particles in the atmosphere (e.g., Grooß et al., 2005; Molleker et al., 2014; Fahey et al., 2001; Tritscher et al., 2019). As simulated removal of HNO3 in the paper is mentioned, the paper should give some information how NAT sedimentation is implemented in WACCM.

**Author's Response:**

Thanks for the reviewer's suggestions. The sedimentation of $HNO_3$ in NAT aerosol follows the approach in Considine et al. (2000). The flux (F) of HNO3 can be derived as follows:
$$F = V \cdot C \cdot \exp(8\ln^2\sigma).$$
here $V$ represents the terminal velocity of NAT aerosol, $C$ denotes the condensed-phase concentration of $HNO_3$, $\sigma$=1.6 (Dye et al., 1992) represents the width of the lognormal size distribution for NAT. Similar sentences have been mentioned in P9 lines 206–210 of the revised manuscript.

**D) Ozone from sondes and simulation**

**D1)** Figure 9 (top) shows an important comparison, namely ozone sonde measurements against simulated ozone. However I suggest not showing the region below about 10 km, which is not of interest here (it also shows basically a blue area). But I think it is important to also show a plot of the differences (observations minus model) which would reveal that the model does not very well simulate to observed ozone depletion in March between 15 and 20 km. Further questions: what is the meaning of negative ozone mixing ratios (top)?

**Author's Response:**

Thanks for the reviewer's advices. The region below about 10 km has been deleted. The temporal resolution of the sounding data from March 25 to April 13, 2020, is once per day, whereas the others are normally once per 3 d during the spring and once per week during the other seasons. The other missing days were obtained by interpolation, so we did not show a plot of the differences (observations minus model). The negative ozone mixing ratio occurs

because of the range of colour scale settings and it has been revised.

**D2)** WACCM seems to overestimate temperatures at about 25 km – is this a real effect?

**Author's Response:**

    The SD-WACCM simulated temperatures were generally 0.6–3 K higher than the MLS temperatures between 100 and 1 hPa. For SD-WACCM, because heterogeneous chemistry is temperature-dependent, the model generally overestimated HCl and underestimated ClO in the lower stratosphere during winter, implying insufficient chlorine activation. Similar conclusions have also been reported by previous studies (Brakebusch et al., 2013; Solomon et al., 2015; Pan et al., 2018).

**E) Formation of HCl**

The presented WACCM results suggest that the deactivation in the Arctic in 2021 is partly caused by formation of HCl, This is the classic deactivation pathway in the Antarctic, but not in the Arctic (e.g. Crutzen et al., 1992; Douglass et al., 1995; Müller et al., 2018). The authors might want to comment on this point.

**Author's Response:**

    Thanks for the reviewer's advices. We indeed want to comment this point. Please see P14, lines 348–352.

"Formation of HCl is considered to be the main chlorine deactivation mechanism in Antarctica (Müller et al., 2018), but not in the Arctic. HCl increases more rapidly in the Antarctic vortex in spring than in the Arctic vortex (Douglass et al., 1995). In early March 2020, in the Arctic, chlorine was deactivated as HCl and $ClONO_2$, and the PSC that permitted chlorine activation remained. Furthermore, activated chlorine compounds were mainly deactivated as $ClONO_2$ by the $ClO + NO_2$ reaction (Müller et al., 1994; Douglass et al., 1995)."

**F) References**

Several references have been cited in this review; hopefully they are helpful. The point is not that the authors should feel obliged to cite these references. However, the paper cites WMO (2014); I suggest that a more recent ozone assessment should be used in the paper (WMO, 2018). The most recent (2022) assessment has just been released (https://ozone.unep.org/science/assessment/sap) and might be helpful when revising this paper.

**Author's Response:**

    Thanks for the reviewer's suggestion. More recent references have been cited in the revised manuscript.

**G) Data availability**

The data availability statement in this paper is not good. I suggest making the DOAS observations at Ny-Ålesund available for download on a server that issues a doi and where the data are permanently archived. Such links are reported for (e.g.) SAOZ but not for the DOAS

measurements presented in the paper. Further, the WACCM data need to be better described (see above). Making data available through e-mail request is no longer recommended.

**Author's Response:**
Thanks for the reviewer's advice. The DOAS observations at Ny-Ålesund available from https://doi.org/10.17632/jx7nkspkg7.1 and where the data are permanently archived. SAOZ data from http://saoz.obs.uvsq.fr/. The WACCM data have been further described in section 2.2 of the revised manuscript.

**Details**
• Title: I suggest avoiding "Research on" in the title; isn't this obvious? The title should rather reflect the fact that DOAS measurements from Ny- Ålesund are reported here.

**Author's Response:**
Thanks for the reviewer's suggestion. The title has been revised as "The unusual spring 2020 Arctic stratospheric ozone depletion above Ny-Ålesund by ground-based ZSL-DOAS".

• p. 1, l. 16: why this period? (I think this is the period when measurements are available, but this should be clear from the paper).

**Author's Response:**
The light intensities were strong enough during this period and the measurements were available.

• p 1, l. 21: what is a "normal year" in the Arctic?

**Author's Response:**
In this manuscript, the data for 2020 were compared with that for the other years (2017, 2018, 2019, and 2021) in the Arctic. The sentence has been revised. Please see P1, lines 19–20.

"which was about 64.7±0.1% of that in the other years (2017, 2018, 2019, and 2021)"

• p. 1, l. 21: 44.3 % → here and elsewhere in the paper: add an error estimate for the ozone loss.

**Author's Response:**
Compared with the other years, the 2020 daily peak relative ozone difference was −44.3±0.1%. Error estimates have been added elsewhere in the revised manuscript. Please see P11, lines 268–270.

"Compared to the other four years, the 2020 daily average relative differences from March 18 to April 18 from the GOME-2, ZSL-DOAS, Brewer, and SAOZ datasets were −36.5%, −35.3±0.4%, −33.1±0.7%, and −32.0±0.1%, respectively."

• p. 1, l. 23: here ans elsewhere: PV and ozone depletion: is this only a complicated way of

saying that there is no ozone loss outside the vortex? I think that Ny-Ålesund was located outside the vortex at about April 16 (see also Fig. 8.

**Author's Response:**
Thanks for the reviewer's suggestion. The Fig. 8 has been deleted. We have rewritten this part. Please see P12, lines 293–300. In addition, we reviewed the literature and found that a large and strong Arctic vortex lasted from early December anomalously into the final week of April (Kuttippurath et al., 2021). As can be seen from the figure below, Ny-Ålesund was located inside the vortex on April 17.

[Figure]

**Figure 2.** Polar vortex evolution in the Arctic winter/spring of 2019/2020. The evolution of polar vortex in the Arctic winter of 2020. The vortex situation in the lower-stratospheric altitude of about 460 K (∼ 17 km) is illustrated. The vortex edge is calculated with respect to the Nash et al. (1996) criterion at each altitude.

Figure cited from Kuttippurath et al. (2021).

"A cold and stable polar vortex is a prerequisite for ensuring that Arctic stratospheric temperatures are sufficiently low. The 2019/2020 winter was unique and the polar vortex was unusually stable, prolonged, and cold (Lawrence et al., 2020; Wohltmann et al., 2020; Rao and Garfinkel, 2020). A large and strong Arctic vortex lasted from early December anomalously into the final week of April (Kuttippurath et al., 2021). The faint planetary wave activity in the Northern Hemisphere also contributed to the formation of a cold and strong vortex (Feng et al., 2021). Unusually low temperature and strong and prolonged vortex in the 2019/2020 winter provided favourable meteorological conditions for ozone depletion in the Arctic."

• p. 1, l. 26: how new is the peak in ClO (chlorine activation)? Compare the papers in the JGR/GRL special issue?

**Author's Response:**
The peak in ClO (chlorine activation) has been discussed in the JGR/GRL special issue. However, in addition to emphasizing the reliability of our observations, we also analyzed the influence of halogen chemistry processes, particularly bromine chemistry.

• p. 2, l. 38: this is not a good description of halogen induced polar ozone loss (e.g., Müller et al., 2018, and Solomon 1999, Tritscher 2021, cited in the paper).

**Author's Response:**
The description has been revised. Please see P2, lines 48–53.

"Since the late 1970s, Antarctic stratospheric ozone during the austral spring has decreased sharply, mainly because of elevated concentrations of active chlorine (Farman et al., 1985). When the weather is cold and there is sufficient sunlight, chlorofluorocarbons derived from anthropogenic emissions can be converted to produce active chlorine, and then to maintain the chlorine activation process, which causes ozone depletion (Müller et al., 2018; Solomon, 1999; Tritscher et al., 2021).

• p 2., l. 42: 'recovery' is an important issue, it is different in the polar regions and in midlatitudes (WMO, 2018). See also further papers on the recovery of both the Antarctic ozone hole and global ozone levels (e.g., Kuttippurath and Nair, 2017; Strahan and Douglass, 2018; WMO, 2018; Bodeker and Kremser, 2021; Stone et al., 2021; Weber et al., 2022).

**Author's Response:**
Thanks for the reviewer's advice. The sentence has been revised. Please see P3, lines 53–56.

"As anthropogenic emissions of ozone-depleting substances have decreased since the Montreal Protocol was enforced, the concentrations of ozone in the Antarctic stratosphere were predicted to recover to pre-1980 values in 2060 (Solomon et al., 2016; Stone et al., 2021; Dhomse et al., 2018; Kuttippurath and Nair, 2017; Strahan and Douglass, 2018)".

• p. 2, l. 49: there should be more citations here than just Hu 2020.

**Author's Response:**
More references have been cited. Please see P3, lines 62–64.

"Between mid-February and late March 2020, the persistence of anomalously faint wave activities in the Arctic led to an abnormally persistent and cold vortex, which caused significant ozone loss (Hu, 2020; Kuttippurath et al., 2021; Ardra et al., 2022)".

• p. 3, l. 75: Simpson is on boundary layer issues: this reference needs to be changed. There are several alternative citations, already cited in the paper and there are further modelling papers cited in this review.

**Author's Response:**
This reference has been changed. Please see P4, lines 101–104.

"In addition, compared to ground-based observation, modelling provides a wider coverage and favours the investigation of ozone depletion. (Müller et al., 1994; Wohltmann et al., 2010; Griffin et al., 2019; Grooß and Müller, 2021).".

• p. 3, l. 75: These citations focus on one particular model (CLaMS), which is okay. But I think you should have citations to other models here as well (e.g., Chipperfield, 1999; Khosrawi et al., 2009; Bekki et al., 2013; Chipperfield et al., 1994; Kinnison et al., 2007; Wohltmann and

Rex, 2009; Wohltmann et al., 2010).

**Author's Response:**
Thanks for the reviewer's suggestion. More references have been cited. Please see P5, lines 104–107.

"Recently, stratospheric chemical patterns, consisting of a group of heterogeneous reactions, have been developed in various models according to investigations and experiments conducted in the polar area (McKenna et al., 2002; Grooß et al., 2011, 2018; Chipperfield, 1999; Khosrawi et al., 2009; Bekki et al., 2013; Chipperfield et al., 1994; Kinnison et al., 2007; Wohltmann and Rex, 2009)".

• p. 4, l. 99: You cannot start the Methods section with "the DOAS instrument". Which instrument? I think it is a new instrument that is described below – correct? This should be much clearer from the paper and the instrument needs to be described first before it can be "placed" somewhere. Further, given the fact that the DOAS technique is so prominent here (or should be) a bit more background on DOAS and citations (see perhaps, Huneker et al., 2017) might be appropriate.

**Author's Response:**
      Thanks for the reviewer's advice. The ZSL-DOAS instrument has been further described in the revised manuscript. Please see P6 lines 135–145. We have also added a bit more background on DOAS technique. Please see P4 lines 95–97.

"The ZSL-DOAS instrument mainly includes the prism, telescope, computer, filter, motor, and CCD spectrometer. The motor controlled the telescope that can change the angle of elevation between the horizon and the zenith. As the angle of elevation changes, the telescope can acquire scattered sunlight at different angles (2°, 3°, 4°, 6°, 8°, 10°, 15°, 30°, and 90°). The quartz fibre can transform the incident light and its numerical aperture is 0.22. The light is received by the spectrometer (Ocean Optics MAYA pro) and measured by a 2048 pixels CCD. This spectrometer was designed for wavelengths between 290 and 429 nm, and had the spectral resolution (FWHM) of 0.5 nm. The integration time varied between 100 and 2000 ms due to the light intensity. The detector operates normally at approximately 20°C with a thermal controller. The mercury lamp spectra, offsets and dark currents were calibrated ahead of the experiments. The ZSL-DOAS instrument can detect $O_3$, $NO_2$, OClO, BrO, and $O_4$. The ozone slant column density (SCD) was retrieved, with the raw data obtained in the zenith direction (90°). The ZSL-DOAS instrument was placed at the Yellow River Station (78.92° N, 11.93° E) in the Arctic. Figure 1 shows the ZSL-DOAS instrument and experimental location, in Ny-Ålesund, Svalbard, Norway."

"In the 1970s, differential optical absorption spectrometry (DOAS) was developed by Platt and Stutz (2008) and has been widely used to measure several trace gases of ozone, nitrogen dioxide, bromine monoxide, and sulfur dioxide (Hüneke et al., 2017)."

• p. 6, p. 140: this sentence starts with 'parameters' but the paper should state what was actually done regarding WACCM.

**Author's Response:**
We have rewritten section 2.2. Please see response to **Comments B1**.

• p. 7, l. 166: ERA5 has 137 layers – is there a typo here?

**Author's Response:**
There is not a typo here. The ERA5 data had the spatial resolution of 0.25° × 0.25° and were divided into 37 layers vertically, from 1000 hPa to 1 hPa.

• p. 7, l. 167: where have these measurements been done?

**Author's Response:**
These measurements were carried out at Ny-Ålesund.

• p. 8, l. 173: what is a 'normal year'?

**Author's Response:**
This response is similar to **Details p 1, l. 21**.

• p. 9, l. 199: what is the 'threshold temperature'? This is an important point that should be discussed in the paper.

**Author's Response:**
Please see response to **Comments C1**.

• p. 9, l. 201: by definition the PV in the southern hemisphere is negative and positive in the northern hemisphere. This simple fact should be taken into account when making such statements.

**Author's Response:**
Thanks for the reviewer's advice. The sentence has been deleted.

• p. 10., l. 235: apparent → obvious?

**Author's Response:**
It has been revised.

• p. 10, l. 243: 'recover' is problematic here, it is not the right word to use when taking about chlorine deactivation putting a halt to ozone loss.

**Author's Response:**

It has been revised. Please see P15 lines 353–354.

"In mid-April 2020, ClONO$_2$ stopped increasing and ClO was almost depleted when the ozone concentration started to increase."

• p. 10., l. 237: it is not only the reaction HCl + ClONO$_2$

**Author's Response:**
The heterogeneous reactions HCl + ClONO$_2$ and HOCl + HCl and the gas-phase reaction CH$_3$O$_2$ + ClO contributed to the conversion of HCl to active chlorine (Müller et al., 1994; Müller et al., 2018). And the sentence has been revised. Please see P14 lines 345–346.

"Chlorine was dominantly activated by ClONO$_2$ + HCl and this reaction improved up to 10 times when the temperature reduced by 2.3 K (Wegner et al., 2012)."

• Figure 5: cold the errors of the individual measurements be used for weighting the data when calculating regression etc.?

**Author's Response:**
Thanks for the reviewer's suggestion. We added the errors of the individual measurements be used for weighting the data when calculating regression etc. No errors were provided from the GOME-2 dataset, so we did not consider measurement errors of GOME-2. Please see P35.

[Figure]

Figure 6. Scatter plots and linear fits of retrieved ozone VCDs with (a) GOME-2, (b) Brewer, and (c) SAOZ.

• Figure 6: Show error bars?

**Author's Response:**
The figure has been revised. Please see P34.

[Figure]

Figure 5. (a) Ozone data for 2020 and the average ozone data (black) of 2017, 2018, 2019, and 2021. (b) relative ozone difference for 2020.

• Figure 7: the blue line shows 195 K, which is an approximation for the onset temperature for heterogeneous chemistry.

**Author's Response:**
Thanks for the reviewer's suggestions. Please see response to **Comments C1**.

• Figure 8: show error bars?

**Author's Response:**
The figure has been deleted. Please see response to **Details p 1, l. 23**.

---

## Author Comment (AC3)

**Response to Reviewer #2**

**General**

In the current manuscript, Li et al has introduced the retrieval method (ZSL-DOAS) by the ground-based instrument installed at Yellow River Station in the Arctic. Then they have validated the retrieved ozone VCD with some widely used measurements including GOME2, Bremer spectrophotometers and SAOZ, the latter uses DOAS with precise measurements of stratospheric constituents during twilight. All these show the observed ozone value is much lower in late winter/early spring in 2020 comparing with the recent five years. Then the authors have investigated the daily variability in ozone changes by calculating the 4-year mean ozone value (2017-2021 excluding the extreme year 2020) and the absolute (and the relative percentage) difference with respace to the mean (?) and have tried to link it to the dynamics (for example by looking at the temperature, PV evolutions). They have also done a model simulation (SD-WACCM) to look at the model performance and the chemical species changes. However, the paper is not well written though the message is still clear for me. There are many confusions (for example, definitions like ozone loss etc. that are quite different from the community has used). It is no doubt that the current ZSL-DOAS observations give another evidence for the unusual Arctic 2020 spring ozone and the unique dataset will be of interest to the atmospheric community. There are many studies over many years show that chlorine and bromine compounds are responsible for the polar ozone depletion in winter and spring. However, the current manuscript has not provided the firm conceptual advance in our understanding of Arctic ozone depletion and there is no new insight into the underlying mechanism responsible for the Arctic ozone depletion. Therefore, the paper has to be rejected or rewritten to find something new.

**Author's Response:**

We would like to thank the reviewer #2 for the careful and valuable comments, which enable us to improve our study and the manuscript remarkably. Please kindly find our point-to-point response to the problems/comments below in blue and the change of the manuscript in orange.

Figure 6 shows indeed not the ozone loss, but the ozone difference between 2020 and the 4-year mean. It has been revised. Please see P34. There are many studies on the Arctic winter 2019/2020 already, so we have presented this work as a Measurement Report, in which the measurements are reported and the consistency with other studies and measurements are shown.

[Figure]

Figure 5. (a) Ozone data for 2020 and the average ozone data (black) of 2017, 2018, 2019, and 2021. (b) relative ozone difference for 2020.

**Specific Comments:**

1. The title is too general. "Research" is quite broad.

**Author's Response:**
Thanks for the reviewer's suggestion. The title has been revised as "The unusual spring 2020 Arctic stratospheric ozone depletion above Ny-Ålesund by ground-based ZSL-DOAS".

2. Line 13 in the Abstract: "Severe", the "third and most severe" is vague. There are still larger Arctic ozone loss for other years. Do you mean the long-lasting cold polar vortex years?

**Author's Response:**
The sentence has been revised. Please see P1 lines 15–16.

"Of the severe stratospheric ozone depletion events (ODEs) reported over the Arctic, the most severe occurred during the spring of 2020."

3. Actually, I find "event" is confusing.

**Author's Response:**
Many studies have reported this event about unprecedented Arctic ozone depletion in the year 2020 (Dameris et al., 2021; Feng et al., 2021; Manney et al., 2020; Wohltmann et al., 2020).

4. There are many "normal" in the whole text. What is the definition for the normal year? Need to be clear with it.

**Author's Response:**
In this manuscript, the data for 2020 were compared with that for the other years (2017, 2018, 2019, and 2021) in the Arctic. The sentence has been revised. Please see P1 lines 18–20.

"The average ozone VCD over Ny-Ålesund between March 18 and April 18, 2020, was approximately 274.8 Dobson units (DU), which was about 64.7±0.1% of that in the other years (2017, 2018, 2019, and 2021)"

5. Some results are obvious: "effect of the polar vortex on stratospheric ozone depletion", "Chlorine activation" and "bromine compounds"

**Author's Response:**
Thanks for the reviewer's suggestion. We have presented this work as a Measurement Report, in which the measurements are reported and the consistency with other studies and measurements are shown.

6. The last sentence in the abstract. What is the main point here? Is this relevant to this work?

**Author's Response:**
The sentence has been deleted. We have rewritten the part in the revised manuscript. Please see P2 lines 38–41.

"By ZSL-DOAS observations, we provided another evidence for unprecedented ozone depletion during the Arctic spring of 2020. The ZSL-DOAS ozone VCD observations can also provide calibration for satellite observations and model simulations, and in the future can provide the support for observations at more Chinese research stations or international local stations in the polar area."

7. Introduction is not well written. Most of them are too general and the background is well known. If you focus on Arctic winter/spring 2020, then you need to brief summary the available publications and what are the unique research questions you need to address or the methods you have applied.

**Author's Response:**
Thanks for the reviewer's suggestion. The introduction has been added. Please see P2–5 lines 44–126.

"Stratospheric ozone is essential for human health, surface ecosystems, and the climate in general (McKenzie et al., 2011) because it absorbs ultraviolet (UV) solar radiation and converts it into thermal energy. The characteristic absorption bands of stratospheric ozone are mainly located in the Hartley and Huggins zones of the UV region and in the Chappuis zone of the visible spectrum, thereby absorbing almost all UV-C (i.e., wavelengths < 280 nm) and some UV-B (i.e., wavelengths ranging between 280 and 315 nm) radiation. Since the late 1970s, Antarctic stratospheric ozone during the austral spring has decreased sharply, mainly because of elevated concentrations of active chlorine (Farman et al., 1985). When the weather is cold and there is sufficient sunlight, chlorofluorocarbons derived from anthropogenic emissions can be converted to produce active chlorine, and then to maintain the chlorine activation process, which causes ozone depletion (Müller et al., 2018; Solomon, 1999; Tritscher et al., 2021).. As anthropogenic emissions of ozone-depleting substances since the Montreal Protocol was enforced, the concentrations of ozone in the Antarctic stratosphere were predicted to recover to pre-1980 values in 2060 (Solomon et al., 2016; Stone et al., 2021; WMO, 2018; Kuttippurath and Nair, 2017; Strahan and Douglass, 2018).

The severe ozone depletion over the Arctic is relatively uncommon compared with that in the Antarctic. During normal Arctic winters, the polar vortex usually fractures and disperses early due to huge planetary wave activities and Brewer–Dobson circulation dynamics (Manney et al., 2003; Dameris, 2010; Harris et al., 2010). Thus, in the Arctic, the duration of the vortex is shorter and relative ozone loss is also lower (Solomon et al., 2007). However, irregular changes in Arctic ozone in recent years have attracted worldwide attention and challenged the existing model. The most severe Arctic ozone depletion lasted for nearly a month, from March to April 2020 (Dameris et al., 2021). Between mid-February and late March 2020, the persistence of anomalously faint wave activities in the Arctic led to an abnormally persistent and cold vortex, which caused significant ozone loss (Hu, 2020; Kuttippurath et al., 2021; Ardra et al., 2022). This event was the most severe reported low Arctic ozone event, following those that occurred in the springs of 1997 and 2011 (Hansen and Chipperfield, 1999; Manney et al., 2011).

The powerful and persistent vortex during the winter and spring is considered as a main cause of significant ozone depletion in the Arctic (Bognar et al., 2021). Polar stratospheric clouds (PSCs) are classified into three types: nitric acid trihydrate (NAT), ice PSCs, and supercooled ternary solution (STS), and their threshold temperatures for existence are $T_{nat}$ (195 K), $T_{ice}$ (188 K), and $T_{sts}$ (195–197 K), respectively (Toohey et al., 1993; Poole and McCormick, 1988; Solomon, 1999). Extremely low air temperatures are essential to produce PSC. The PSC can be used as a surface for heterogeneous interactions, leading to the conversion of reactive halogens from the halogen reservoirs, which can cause serious ozone loss (Frieβ et al., 2005; Marsing et al., 2019). Although the PSC is not only composed of NAT (Pitts et al. 2009; Spang et al. 2018), the temperature threshold for the existence of NAT provides a good estimate on the occurrence of heterogeneous chemistry (Drdla and Müller 2012; Kirner et al. 2015; Grooß and Müller 2021; von der Gathen et al. 2021). PSC might also grow large enough to precipitate and remove $HNO_3$ in the stratosphere, which is the reservoir of $NO_2$. The resulting denitrification from the polar vortex hinders chlorine deactivation by $NO_2$ (Salawitch et al., 1989; Arblaster et al. 2014). Active chlorine is rapidly photolyzed because of the recovery of spring sunlight when ozone loss occurs via the self-reaction of ClO (Molina and Molina, 1987), as well as the cross-reaction of ClO and BrO (McElroy et al., 1986). It is essential that the

vortex retains low temperatures and carries on as a transport impediment so that ozone can remain depleted without $NO_2$ to inactivate chlorine.

The observed Arctic ozone depletion is invaluable for validating stratospheric ozone simulations and for understanding the processes that cause Arctic stratospheric ozone depletion. Currently, ozone vertical column density (VCD) detection utilizes the characteristic ozone absorption in the UV and visible spectra, which provides accurate ozone identification and quantitative measurements. Ground-based observation of ozone VCD started in the first decades of the twentieth century (Dobson, 1968; Brewer, 1973; Solomon et al., 1987; Bognar et al., 2021). From the 1960s, ozonesondes began to acquire atmospheric ozone data (Logan, 1994; Thomason et al., 2011; Wohltmann et al., 2020; Grooß and Müller, 2021).Since 1978, satellite observations have provided essential data for atmospheric ozone related studies (Kuttippurath et al., 2012; Manney et al., 2020). Among these, ground-based observations are crucial to calibrate remotely sensed observations and optimizing inversion results (Lu et al., 2006). In the 1970s, differential optical absorption spectrometry (DOAS) was developed by Platt and Stutz (2008) and has been widely used to measure several trace gases of ozone, nitrogen dioxide, bromine monoxide, and sulfur dioxide (Hüneke et al., 2017).

The impacts of the aberrantly powerful and persistent vortex on ozone in the Arctic were investigated using satellite observations, ozonosonde measurements, and data from the European Centre for Medium-Range Weather Forecasts (ECMWF) (Wohltmann et al., 2020; Lawrence et al., 2020). The major stratospheric halogen species, chlorine, and bromine were investigated in this ozone depletion event (ODE) (Wohltmann et al., 2017, 2021). In addition, compared to ground-based observation, modelling provides a wider coverage and favours the investigation of ozone depletion. (Müller et al., 1994; Wohltmann et al., 2010; Griffin et al., 2019; Grooß and Müller, 2021). Recently, stratospheric chemical patterns, consisting of a group of heterogeneous reactions, have been developed in various models according to investigations and experiments conducted in the polar area (McKenna et al., 2002; Grooß et al., 2011, 2018; Chipperfield, 1999; Khosrawi et al., 2009; Bekki et al., 2013; Chipperfield et al., 1994; Kinnison et al., 2007; Wohltmann and Rex, 2009). Global and area models using different stratospheric chemical patterns have been applied to simulate ozone columns, which usually compare well with satellite observations and ozonosonde data (Pan et al., 2018; Grooß and Müller, 2021).

Accurate ground-based observations can improve the accuracy and reliability of models as well as enhancing our understanding of the reasons for ozone depletion. In this study, we have developed a ground-based DOAS system that can conduct ozone VCD observations in the Arctic. The zenith scattered light observation mode was applied to measure ozone VCD using the Langley Plot method (Frieß et al., 2005).

We analyze the reasons for this ODE in the unusual spring of 2020 above Ny-Ålesund, Norway. The methods and data are given in Sect. 2, which covers the presentation of the experimental location and DOAS instrument, the DOAS method, the specified dynamics version of the Whole Atmosphere Community Climate Model (SD-WACCM), Global Ozone Monitoring Experiment 2 (GOME-2) observations, Brewer measurements, Système d'Analyze par Observation Zénithale (SAOZ) measurements, ECMWF data, and ozonesonde data. Section 3 presents the results, where Sect. 3.1 describes the results of ozone VCDs from February 2017 to October 2021 and ozone difference in spring 2020. The zenith scattered light DOAS (ZSL-

DOAS) retrieved the daily variations in ozone VCDs, which were in comparison with GOME-2 observations, Brewer, and SAOZ measurements. A detailed characterization of this ODE is presented for establishing the basis of the subsequent analysis. The relationship between Arctic ozone depletion and meteorological conditions in terms to temperature and potential vorticity (PV) is described in Sect. 3.2. In Sect. 3.3, this ODE was analyzed using the SD-WACCM to further illustrate the ozone depletion process, and to explore the effects of chemical depletion and dynamic transport on this ODE. The influence of the halogen species is discussed in Sect. 3.4. The comprehensive summary is provided in Sect. 4."

8. I also find it is difficult to see the purpose of this work. Are you aiming to validate your ZSL-DOAS observations using other measurements? I am not an expert in the retrieval method so it is hard for me to judge your method. What is the difference in the retrieval of ozone from ZSAL-DOAS and SAOZ because SAOZ also uses DOAS method? For the Air Mass Factor (AMF) in section 2.2, the authors have mentioned the related parameters in Table 2 and AMF will be quite different for different assumed profiles (for example Lines 121-123). So I am curious what are these profiles from. It is vague just to mention SCIATRAN without any reference there.

**Author's Response:**
Thanks for the reviewer's advice. We have presented this work as a Measurement Report, in which the measurements are reported and the consistency with other studies and measurements are shown. Both ZSL-DOAS and SAOZ use the DOAS method, but ZSL-DOAS is an instrument self-developed by our group. A priori ozone profile is obtained from the monthly mean climatology. The sentence has been revised. Please see P7 lines 164–167.

"Here, the Air Mass Factor (AMF) can be obtained from the SCIATRAN model and is influenced by a priori ozone profile, SZA (solar zenith angle), wavelength, and surface albedo. Based on the average monthly climate, a priori ozone profile can be achieved. The SZA calculated in this research ranged between 35° and 80°, with surface albedos between 0.08 and 0.6. Table 2 lists these parameters."

9. SD-WACCM. This section needs some more details since the authors have carried out the simulation. It should come from CESM1 (but which version) but not sure what other changes have been made by the author. I assume this is a released version but ported and run on a different HPCx in China.

**Author's Response:**
Thank for your suggestion. We have rewritten this section and added description on the WACCM model. In this study, we replace the standard polar stratospheric cloud module with that from Wegner et al. (2013). Similar sentences have been written in P9 lines 204–206.

"The polar stratospheric cloud module used in this study followed Wegner et al. (2013) rather than the standard module of Kinnison et al. (2007), improving the capabilities of WACCM in modelling ozone and its associated components (Brakebusch et al., 2013)."

10. However, some of the emissions and other input data have not been updated/available for year 2020. One simple example is that this version uses the prescribed stratospheric sulphur aerosol density (SAD) which is important for the ozone depletion. How the SAD used in SD-WACCM4 for this? Do you use the previous year's values or fixed values? Which component you are using? It seems that the authors only mentioned MOZART3, so I am not sure if this one "pp_waccm_mozart" used or you also have MAM model in the model simulation etc..

**Author's Response:**
We have rewritten this section and added description on the WACCM model. Please see P8 and P9, lines 191–206.

"The chemical mechanism of WACCM4 also contains 4 aerosol types heterogeneous reactions: liquid binary sulfate (LBS), supercooled ternary solution (STS), nitric acid trihydrate (NAT), and water-ice. When model temperatures above 200K, only the LBS exists. The surface area density (SAD) of LBS is from SAGE, SAGE-II and SAMS observations (Thomason et al., 1997) and Considine update it (World Meteorological Organization, 2003). With the model atmosphere cooling, the LBS aerosol expands and absorbs both $HNO_3$ and $H_2O$ to obtain the STS aerosol. Tabazadeh et al. (1994) derived the composition of STS by the Aerosol Physical Chemistry Model (ACPM). The STS aerosol median radius and SAD is derived following the approach of Considine et al. (2000). When model temperatures reach a specified supersaturation ratio of $HNO_3$ for NAT, $HNO_3$ containing aerosols are allowed to form. In WACCM4, Peter et al. (1991) set this ratio to 10. NAT median radius and SAD are derived in the same way with STS aerosol. If the derived atmospheric temperature does not exceed the saturation temperature of water vapour on ice (Tsat), then this results in the formation of water-ice aerosols. In WACCM4, the CAM's prognostic water routines gives the condensed phase $H_2O$, which is conveyed to the chemistry module. According to the method of Considine et al. (2000), the median radius and SAD of water-ice can be derived by this condensed phase $H_2O$. The polar stratospheric cloud module used in this study followed Wegner et al. (2013) rather than the standard module of Kinnison et al. (2007), improving the capabilities of WACCM in modelling ozone and its associated components (Brakebusch et al., 2013)."

11. For the SD, the author need to realize that this is not a fully nudged version, which all depends on the relaxation time etc. This needs to make clear.

**Author's Response:**
Thank for your suggestion. We have added some description of the calculation of the meteorological fields. Please see P9, lines 214–221.

"Meteorological fields were calculated using a nudging method in the model (Lamarque et al., 2012). Data for the horizontal winds, temperature, and surface pressure from MERRA-2 were used to drive the physical parameterization from the surface to 50 km (Kunz et al., 2011), which allowed for more accurate comparisons between the measurements of atmospheric composition and the model output (Lamarque et al., 2012). This can be employed for the study of specific

weather events. Linear transitions were used in the 50–60 km altitude range and over 60 km, and online calculations were performed. In this study, the MERRA-2 dataset has the same resolution with the SD-WACCM, which can be accessed on the Earth System Grid (https://www.earthsystemgrid.org/home.html) and are obtained from the original resolution (1/2°×2/3°) by a conservative re-gridding procedure (Lamarque et al., 2012; Pan et al., 2019).”

12. I have not heard "thermogenic" layer, should be use the correct term like "lower thermosphere". It is strange to say the "parameters" in the Line 140. Please note it uses CAM4 physics.

**Author's Response:**
Thank for your suggestion. The word has been revised and we have rewritten the section 2.2. The WACCM4 model is based on the physical parameterizations used in the Community Atmosphere Model version 4 (CAM4) (Neale et al., 2013).

13. It is improperiate to say "the SD-WACCM with meteorological parameters driven by …", note even SD-WACCM most of the temperature etc. are still from WACCM itself (driven by SST, solar etc.…). How can you say "Data from MERRA-2 guaranteed the accuracy of simulated values"? Note that other processes play roles.

**Author's Response:**
we have rewritten the section 2.2. These sentences have been corrected. Please see P9, lines 215–217.

"Data for the horizontal winds, temperature, and surface pressure from MERRA-2 were used to drive the physical parameterization from the surface to 50 km (Kunz et al., 2011), which allowed for more accurate comparisons between the measurements of atmospheric composition and the model output (Lamarque et al., 2012)."

14. Auxiliary data: Can be concise and have proper references. For example ERA5. Some sentences are not necessary at all. For example. Lines 152, 165.

**Author's Response:**
Thanks for the reviewer's suggestion. Some sentences have been deleted.

15. It reads to me that the authors just SHOW the results itself, rather than describe the results in a correct/proper way. For example, for Figure4, we can see large daily variability that has never mentioned. We also see the differences among these measurements but have never explained. For example, why your data ZSL-DOAS is much lower than other observations?    I don't think the gradient of "~0.92DU per day" is similar for all the "normal" years claimed. It is also not correct to say "Ozone VCD begins to decrease in March."

**Author's Response:**
Thanks for the reviewer's advice. During the observation period, the average ozone VCD from

ZSL-DOAS was 2.25% lower than that from GOME-2, 4.92% lower than that from Brewer, and 2.26% higher than that from SAOZ. These differences of ZSL-DOAS observations compared to other observations are due to systematic errors of the instrument. These descriptions have been revised. Please see P11, lines 257–261.

"In the other years (2017, 2018, 2019, and 2021), ozone VCD showed a fluctuating downward trend between March and September, with a small upward trend around March and August. In 2020, however, severe ozone depletion occurred between March 18 and April 18, after which ozone VCD gradually increased. Ozone VCD decreased further in mid-May. In around September, the ozone VCD increased obviously again, probably due to clear warming of the polar stratosphere."

16. For Figure 5, I understand the VCD and TCO can be used but the authors need to be consistent in the whole text.

**Author's Response:**
It has been revised. We have used VCD in the whole text.

17. The presentation for Figure 6 is not good at all. Why the authors term this "ozone loss"? How do you estimate "ozone loss"? It looks that this is just ozone difference between 2020 and the 4-year mean.

**Author's Response:**
Figure 6 shows indeed not the ozone loss, but the ozone difference between 2020 and the 4-year mean. It has been revised. Please see response to **General** above.

18. For Figure 7, it seems that temperature is from ERA5, why use "measured" in Line 190? It is a reanalysis product, which is from ECMWF model simulation using the data assimilation from the measurements.

**Author's Response:**
It has been revised. Please see P12, lines 283–284.

"Daily average temperatures of Ny-Ålesund between November 2016 and September 2021 were showed at 70 hPa in the low stratosphere, where significant ozone depletion tends to occur (Fig. 7)."

19. Figure 8, it seems that the authors just look at one time period of ozone and PV evolution, then comes the conclusion of "PV correlates negatively with ozone VCD" etc. The authors seems not have a deep understanding their figures even they made it (for example, stratospheric warming after mid-April that made polar vortex weaker and TOC higher) etc..

**Author's Response:**
Thanks for the reviewer's suggestion. The Fig. 8 has been deleted. We have rewritten this part.

"A cold and stable polar vortex is a prerequisite for ensuring that Arctic stratospheric temperatures are sufficiently low. The 2019/2020 winter was unique and the polar vortex was unusually stable, prolonged, and cold (Lawrence et al., 2020; Wohltmann et al., 2020; Rao and Garfinkel, 2020). A large and strong Arctic vortex lasted from early December anomalously into the final week of April (Kuttippurath et al., 2021). The faint planetary wave activity in the Northern Hemisphere also contributed to the formation of a cold and strong vortex (Feng et al., 2021). Unusually low temperature and strong and prolonged vortex in the 2019/2020 winter provided favourable meteorological conditions for ozone depletion in the Arctic."

20. Figure9, the authors said "unusual low". This is only for 2020, have you made O3 volume mixing ration comparison with others.

**Author's Response:**

I made O$_3$ volume mixing ration comparison with that of 2017,2018, and 2019 from the ozonesonde. Besides, I have reviewed relevant literatures that reported unprecedented Arctic ozone depletion in the year 2020 (Dameris et al., 2021; Feng et al., 2021; Manney et al., 2020; Wohltmann et al., 2020).

[Figure]

Figure A1. Between January 1 and July 1, ozone profiles of 2017 (a), 2018 (b), 2019 (c), and 2020 (d) from ozonesonde measurements.

21. Why "< 0.5ppmv suggested the ozone was nearly completed depleted. "?.

**Author's Response:**
By reviewing the literature, "Mixing ratios were consistently below 0.5 ppmv in a wide altitude range (with minima below 0.2 ppmv), indicating near-complete depletion of ozone (Bognar et al., 2021)".

22. For the Tnat, it also depends on $H_2O$, $HNO_3$ and $H_2SO_4$? What is their values used for the Tnat?

**Author's Response:**
Please see P3, lines 68–76.

"Polar stratospheric clouds (PSCs) are classified into three types: nitric acid trihydrate (NAT), ice PSCs, and supercooled ternary solution (STS), and their threshold temperatures for existence are $T_{nat}$ (195 K), $T_{ice}$ (188 K), and $T_{sts}$ (195–197 K), respectively (Toohey et al., 1993; Poole and McCormick, 1988; Solomon, 1999). Extremely low air temperatures are essential to produce PSC. The PSC can be used as a surface for heterogeneous interactions, leading to the conversion of reactive halogens from the halogen reservoirs, which can cause serious ozone loss (Frieß et al., 2005; Marsing et al., 2019). Although the PSC is not only composed of NAT (Pitts et al. 2009; Spang et al. 2018), the temperature threshold for the existence of NAT provides a good estimate on the occurrence of heterogeneous chemistry (Drdla and Müller 2012; Kirner et al. 2015; Grooß and Müller 2021; von der Gathen et al. 2021)."

23. Figure 10, why "$HNO_3$ changes abruptly from abnormally high values to normal values, which indicated the abundant PSC activities of the period" only applies for "Between late January and early February"? What caused the low value patches around 20-22km?

**Author's Response:**
It also applies other period and the sentence has been revised. The low value of HNO3 at 20-22 km is probably due to low temperatures (Fig. 9c–d) leading to PSC activity and severe denitrification (Ardra et al., 2022). Please see P14, lines 335–337.

"$HNO_3$ changed abruptly from abnormally high values to normal values, which indicated the abundant PSC activities of the period (Bognar et al., 2021). The low value of $HNO_3$ at 20-22 km is probably due to low temperatures (Fig. 8c–d) leading to PSC activity and severe denitrification (Ardra et al., 2022)."

24. Figure 11, I saw the model showed a complete HCl depletion on 21 Feb. How the modelled HCl etc. chemical species compared with ACE observation for example?

**Author's Response:**
The figure displays the simulated average diurnal mixing ratios of ozone, chlorine, and bromine compounds for heights of 17.5 km above Ny-Ålesund, but ACE observations do not have the corresponding altitude. Then, we reviewed the literature and also found that a complete HCl depletion around 20 Feb (Grooß and Müller 2021). Furthermore, the reliability of the model

can be validated by comparing it with ozone sonde measurements.

[Figure]

**Figure 10.** Fifty day development of one example air parcel trajectory from the CLaMS simulation that is not affected by mixing over the time period shown. The top panel shows the temperature along the air parcel trajectory (black). The periods when heterogeneous HCl loss rates ($k_{HCl + ClONO2}$ [ClONO$_2$] + $k_{HCl + HOCl}$ [HOCl]) are larger than 1 day$^{-1}$ are marked as gray shaded areas. The middle panel shows the mixing ratio of the chlorine compounds HCl, ClONO$_2$, ClO$_x$ (= ClO + 2 × Cl$_2$O$_2$ + 2 × Cl$_2$) and HOCl. Ozone is shown in the lowest panel on a logarithmic ordinate. The minimum ozone mixing ratio on 24 March (84°N, 131°E, $\theta$ = 439 K) is 38 ppbv.

Figure cited from Grooß and Müller. (2021).

25. This reads like a summary from the main text. I am not sure why the last sentence is matter based on this study.

**Author's Response:**
Thanks for the reviewer's suggestion. The conclusion part has been revised. Please see P16 and P17, lines 376–414.

"In this research, the ozone VCD was obtained from a ground-based instrument, the GOME-2 satellite, and the Brewer and SAOZ instruments and further evaluated with a correlation analysis. The Pearson correlation coefficients were 0.97, 0.87, and 0.91, and the relative deviations were 2.3%, 3.1%, and 3.5%, respectively. Therefore, we can conclude that the method of observing the VCDs of Arctic ozone using a ground-based DOAS instrument is reliable and valid. Compared to the other four years, the 2020 daily average relative differences from March 18 to April 18 from the GOME-2, ZSL-DOAS, Brewer, and SAOZ datasets were −36.5%, −35.3±0.4%, −33.1±0.7%, and −32.0±0.1%, respectively. The results indicated that all instruments recorded severe ozone depletion from March 18 to April 18, 2020.
Unusually low temperature and strong and prolonged vortex in the 2019/2020 winter provided favourable meteorological conditions for ozone depletion in the Arctic. The ozone and temperature profiles were simulated by SD-WACCM, and these simulations corresponded well with ozonesonde measurements. The model results show that ozone depletion at a height range

of 16–20 km is evident from late March to early April, which corresponds to the ozone VCDs obtained from the ground-based instrument. Chlorine and bromine activation were clearly obvious during the Arctic spring of 2020, whereas the partitioning of bromine compounds was different from that of chlorine. Chlorine was predominantly present as HCl and $ClONO_2$ before activation, whereas bromine was predominantly present as HOBr and BrCl before activation. Particularly, bromine existed mainly as HOBr before chlorine activation began. When chlorine was activated, bromine existed mainly as BrCl. In addition, formation of HCl is considered to be the main chlorine deactivation mechanism in Antarctica (Müller et al., 2018). However, in the Arctic, due to HCl increased more slowly than in the Antarctic, chlorine was mainly deactivated as $ClONO_2$.

In summary, by ZSL-DOAS observations, we provided another evidence for unprecedented ozone depletion during the Arctic spring of 2020. The ZSL-DOAS ozone VCD observations can also provide calibration for satellite observations and model simulations, and in the future can provide the support for observations at more Chinese research stations or international local stations in the polar area. Additionally, although WACCM can depict the evolution of ozone during this Arctic ozone depletion event, there are some problems such as overestimation of the temperature and the $CH_3O_2$+ClO reaction is not considered in the current chemical mechanism of the model. This could be considered in future models to improve the simulation performance."

---

## Author Comment (AC4)

**Response to Reviewer #1**

**General**

I think this manuscript could make a contribution eventually, but it needs work. The reader cannot see clearly what the main message of the paper is (see also below). What is new? My understanding is the following: the first purpose is to introduce the new DOAS total ozone measurements at Ny-Ålesund at Yellow River Station. Then these measurements are used to investigate Arctic ozone loss in 2020. If the authors agree, then this point should come across much clearer in the manuscript. And the new DOAS instrument needs to be better described in the manuscript. Given the fact that so much has been published on the Arctic winter 2019/2020 already (see also below), it might be more appropriate to present this work in ACP as a Measurement Report. Further, the authors need to understand the background of the science they are reporting better. Some examples in detail and suggestions for improvement are given below. But as an obvious example: the authors report (on some occasions) the NAT temperature as −195 K – there are no negative values if temperature in measured in K. Overall, I think that the manuscript contains publishable material but I am afraid that restructuring and rewriting large parts of the manuscript are necessary.

**Author's Response:**

We would like to thank the reviewer #1 for the careful and valuable comments, which enable us to improve our study and the manuscript remarkably. Please kindly find our point-to-point response to the problems/comments below in blue and the change of the manuscript in orange.

We agreed to present this work as a Measurement Report, in which the measurements are reported and the consistency with other studies and measurements are shown. We focused on introducing the new DOAS total ozone measurements at Ny-Ålesund at Yellow River Station and then used these measurements to study the Arctic ozone loss in 2020. In addition, the new DOAS instrument was further described in the revised manuscript. Please see P6 lines 135–145. The temperature threshold for the existence of NAT as 195K has been revised.

"The ZSL-DOAS instrument mainly includes the prism, telescope, computer, filter, motor, and CCD spectrometer. The motor controlled the telescope that can change the angle of elevation between the horizon and the zenith. As the angle of elevation changes, the telescope can acquire scattered sunlight at different angles (2°, 3°, 4°, 6°, 8°, 10°, 15°, 30°, and 90°). The quartz fibre can transform the incident light and its numerical aperture is 0.22. The light is received by the spectrometer (Ocean Optics MAYA pro) and measured by a 2048 pixels CCD. This spectrometer was designed for wavelengths between 290 and 429 nm, and had the spectral resolution (FWHM) of 0.5 nm. The integration time varied between 100 and 2000 ms due to the light intensity. The detector operates normally at approximately 20°C with a thermal controller. The mercury lamp spectra, offsets and dark currents were calibrated ahead of the experiments. The ZSL-DOAS instrument can detect $O_3$, $NO_2$, $OClO$, $BrO$, and $O_4$. The ozone slant column density (SCD) was retrieved, with the raw data obtained in the zenith direction (90°). The ZSL-DOAS instrument was placed at the Yellow River Station (78.92° N, 11.93° E)

in the Arctic. Figure 1 shows the ZSL-DOAS instrument and experimental location, in Ny-Ålesund, Svalbard, Norway."

**Comments**

**A) What are the main messages of the paper?**
First: the paper states that ozone VCD from a ground-based instrument, the GOME-2 satellite, and the Brewer and SAOZ instruments agree rather well. However, this is not a very new conclusion and had been discussed in many (mostly more technically oriented) papers before (e.g., Léon-Luis et al., 2018; Fioletov et al., 2002; Fioletov, 2002; Fioletov et al., 2005; Weber et al., 2005, and references therein). Second, the paper reports that substantial ozone depletion occurred in the Arctic vortex until mid-April 2020, consistent with changes in simulated HNO3. Again this is today not very new information; there is a special issue in JGR/GRL (and some of the papers on the Arctic winter 2020 in this special issue are cited/discussed in this manuscript) but there are a few more papers on Arctic ozone in 2020 in the meantime (e.g., von der Gathen et al., 2021; Kuttippurath et al., 2021; Ardra et al., 2022). Third, ozone and temperature profiles were simulated by SD-WACCM, with these simulations corresponding well with ozonesonde measurements (but how well? – see below). The study used SD-WACCM with meteorological parameters driven by Modern Era Retrospective-Analysis for Research and Applications version 2 data; thus the simulation of temperature profiles by SD-WACCM is expected – isn't it? The fact that the ozone sonde measurements can be reproduced by the model is good but should be stated more clearly and in particular more quantitatively. Finally the paper closes with the statement that "observations of ozone VCDs over Ny-Ålesund will continue in order to monitor future ozone changes over the area." This is very good of course but not a conclusion from this paper.

**Author's Response:**
    Thanks for the reviewer's advices. Ozone VCDs from a ground-based instrument, the GOME-2 satellite, and the Brewer and SAOZ instruments agree rather well and substantial ozone depletion occurred in the Arctic vortex until mid-April 2020, consistent with changes in simulated $HNO_3$. The reviewer is correctly saying that these are not very new information today. Thus, we have presented this work as a Measurement Report, in which the measurements are reported and the consistency with other studies and measurements are shown.
    The simulation of temperature profiles by SD-WACCM indeed corresponded well with ozonesonde measurements, and this can be used to validate the simulation. The temporal resolution of the sounding data from March 25 to April 13, 2020, is once per day, whereas the others are normally once per 3 d during the spring and once per week during the other seasons. The other missing days were obtained by interpolation, so we did not show a plot of the differences (observations minus model).
    The sentence has been revised. Please see P17 lines 408–414.

"In summary, by ZSL-DOAS observations, we provided another evidence for unprecedented ozone depletion during the Arctic spring of 2020. The ZSL-DOAS ozone VCD observations can also provide calibration for satellite observations and model simulations, and in the future

can provide the support for observations at more Chinese research stations or international local stations in the polar area. Additionally, although WACCM can depict the evolution of ozone during this Arctic ozone depletion event, there are some problems such as overestimation of the temperature and the $CH_3O_2+ClO$ reaction is not considered in the current chemical mechanism of the model. This could be considered in future models to improve the simulation performance."

**B) WACCM**
**B1)** Some results of the paper rely on the model WACCM. But it is not clear how these results are obtained. I presume (although this is not stated in the paper) that openly available WACCM results have been used. If this is the case it should be clearly stated. If not, the WACCM runs conducted by the authors should be described (see also details) and then the WACCM version used should be clear.

**Author's Response:**
     Thanks for the reviewer's suggestion. We have rewritten the description of model setting in section 2.2 of the revised manuscript. Please see P8 and P9, lines 186–222.

"The physical parameterizations employed in the Community Atmosphere Model Version 4 (CAM4) were applied to the WACCM (Neale et al., 2013). At present, the WACCM model is incorporated into a component set of the Community Earth System Model, whose source code is available online (https://svn-ccsm-release.cgd.ucar.edu/model versions/). The Model for Ozone and Related Chemical Tracers, version 3 (MOZART-3) provided the chemical parameters for the WACCM (Kinnison et al., 2007). This mechanism contains 52 neutral species, one invariant ($N_2$), 127 neutral gas-phase reactions, 48 neutral photolytic reactions, and 17 heterogeneous reactions [see Tables 5.1-5.5 in Neale et al. (2013)]. The chemical mechanism of WACCM4 also contains 4 aerosol types heterogeneous reactions: liquid binary sulfate (LBS), supercooled ternary solution (STS), nitric acid trihydrate (NAT), and water-ice. When model temperatures above 200K, only the LBS exists. The surface area density (SAD) of LBS is from SAGE, SAGE-II and SAMS observations (Thomason et al., 1997) and Considine update it (World Meteorological Organization, 2003). With the model atmosphere cooling, the LBS aerosol expands and absorbs both $HNO_3$ and $H_2O$ to obtain the STS aerosol. Tabazadeh et al. (1994) derived the composition of STS by the Aerosol Physical Chemistry Model (ACPM). The STS aerosol median radius and SAD is derived following the approach of Considine et al. (2000). When model temperatures reach a specified supersaturation ratio of $HNO_3$ for NAT, $HNO_3$ containing aerosols are allowed to form. In WACCM4, Peter et al. (1991) set this ratio to 10. NAT median radius and SAD are derived in the same way with STS aerosol. If the derived atmospheric temperature does not exceed the saturation temperature of water vapour on ice ($T_{sat}$), then this results in the formation of water-ice aerosols. In WACCM4, the CAM's prognostic water routines gives the condensed phase $H_2O$, which is conveyed to the chemistry module. According to the method of Considine et al. (2000), the median radius and SAD of water-ice can be derived by this condensed phase $H_2O$. The polar stratospheric cloud module used in this study followed Wegner et al. (2013) rather than the standard module of Kinnison et al. (2007), improving the capabilities of WACCM in modelling ozone and its

associated components (Brakebusch et al., 2013). The sedimentation of $HNO_3$ in NAT aerosol follows the approach in Considine et al. (2000). The flux (F) of $HNO_3$ can be derived as follows:

$$F = V \cdot C \cdot \exp(8\ln^2 \sigma). \tag{4}$$

here $V$ represents the terminal velocity of NAT aerosol, $C$ denotes the condensed-phase concentration of $HNO_3$, $\sigma$=1.6 (Dye et al., 1992) represents the width of the lognormal size distribution for NAT.

We used the SD-WACCM with meteorological parameters driven by Modern Era Retrospective-Analysis for Research and Applications version 2 (MERRA-2) data (Gelaro et al., 2017). The SD-WACCM had the horizontal resolution of 1.9° × 2.5° (lat × lon). The model was divided vertically into 88 layers, covering an altitude of ~140 km from the ground to the bottom of the lower thermosphere layer. Meteorological fields were calculated using a nudging method in the model (Lamarque et al., 2012). Data for the horizontal winds, temperature, and surface pressure from MERRA-2 were used to drive the physical parameterization from the surface to 50 km (Kunz et al., 2011), which allowed for more accurate comparisons between the measurements of atmospheric composition and the model output (Lamarque et al., 2012). This can be employed for the study of specific weather events. Linear transitions were used in the 50–60 km altitude range and over 60 km, and online calculations were performed. In this study, the MERRA-2 dataset has the same resolution with the SD-WACCM, which can be accessed on the Earth System Grid (https://www.earthsystemgrid.org/home.html) and are obtained from the original resolution (1/2°×2/3°) by a conservative re-gridding procedure (Lamarque et al., 2012; Pan et al., 2019). In this study, the simulation is initiated between November 1, 2019, and July 1, 2020."

**B2)** Also the way how the WACCM source code can be obtained should then be documented. Further, section 2.3 cites Kunz et al. (2011) – this is a good paper, but the paper does not deal with MERRA 2, so this sentence is confusing.

**Author's Response:**

The WACCM is a component set of CESM. And the CESM code is available online (https://svn-ccsm-release.cgd.ucar.edu/model_versions/). Similar sentences have been mentioned in P8 lines 187–188 of the revised manuscript.

We have rewritten the description of the nudging method used in SD-WACCM. Please see P9 lines 215–219.

" Data for the horizontal winds, temperature, and surface pressure from MERRA-2 were used to drive the physical parameterization from the surface to 50 km (Kunz et al., 2011), which allowed for more accurate comparisons between the measurements of atmospheric composition and the model output (Lamarque et al., 2012). This can be employed for the study of specific weather events. Linear transitions were used in the 50–60 km altitude range and over 60 km, and online calculations were performed."

**B3)** Further, which chemical scheme has been used in these simulations? I assume the most recent JPL recommendation (Burkholder et al., 2019).

**Author's Response:**

The basic chemistry mechanism in the WACCM is taken from the MOZART-3. Please see P8 lines 188–190.

"The Model for Ozone and Related Chemical Tracers, version 3 (MOZART-3) provided the chemical parameters for the WACCM (Kinnison et al., 2007)."

**B4)** Müller et al. (1994, cited) emphasize the importance of $CH_3O_2 + ClO$ for Arctic ozone loss – is this reaction taken into account in the WACCM simulation?

**Author's Response:**

This reaction is not included in the WACCM model. Müller et al. (1994) found that the $CH_3O_2+ClO$ reaction is also important but is not included in the current chemical mechanism of the model and could be taken into account in future models to improve simulation performance. Please see P14, lines 325–327.

"On the other hand, Müller et al. (1994) found that the $CH_3O_2+ClO$ reaction is also important but is not included in the current chemical mechanism of the model and could be taken into account in future models to improve simulation performance."

**B5)** More importantly, in which reference is the list of reactions described that is employed in the described chemical simulation? This information should be given in the paper.

**Author's Response:**

Thank for your suggestion. Please see P8, lines 190–191.

"This mechanism contains 52 neutral species, one invariant ($N_2$), 127 neutral gas-phase reactions, 48 neutral photolytic reactions, and 17 heterogeneous reactions [see Tables 5.1-5.5 in Neale et al. (2013)]."

**B6)** I also note that 'atmospheric simulations' are not mentioned in the author contribution. In general, it should be clear from the paper how the WACCM results were obtained.

**Author's Response:**

The WACCM simulation was conducted by Chen Pan. We have added statements in the author contribution. We also rewritten the description of the model settings. Please see section 2.2 in the revised manuscript.

**C) PSCs**

**C1)** Clearly PSCs are important to polar ozone loss. However, first, one has to discriminate between PSC 'formation' and 'existence'. For crystalline particles (NAT and ice) this is not the same thing. (see e.g. Tritscher et al 2021). Also the temperature threshold for the onset of heterogeneous chemistry is not the same thing as NAT existence (Drdla and Müller, 2012, see also Tritscher 2021;Solomon1999, cited in the paper).

**Author's Response:**

Thanks for the reviewer's suggestions. Please see P3, lines 68–76.

"Polar stratospheric clouds (PSCs) are classified into three types: nitric acid trihydrate (NAT), ice PSCs, and supercooled ternary solution (STS), and their threshold temperatures for existence are $T_{nat}$ (195 K), $T_{ice}$ (188 K), and $T_{sts}$ (195–197 K), respectively (Toohey et al., 1993; Poole and McCormick, 1988; Solomon, 1999). Extremely low air temperatures are essential to produce PSC. The PSC can be used as a surface for heterogeneous interactions, leading to the conversion of reactive halogens from the halogen reservoirs, which can cause serious ozone loss (Frieβ et al., 2005; Marsing et al., 2019). Although the PSC is not only composed of NAT (Pitts et al. 2009; Spang et al. 2018), the temperature threshold for the existence of NAT provides a good estimate on the occurrence of heterogeneous chemistry (Drdla and Müller 2012; Kirner et al. 2015; Grooß and Müller 2021; von der Gathen et al. 2021)."

**C2)** Further, denitrification by sedimenting NAT particles is touched upon in the paper. It is not straightforward implementing sedimentation in a model and explain the observations of large NAT particles in the atmosphere (e.g., Grooß et al., 2005; Molleker et al., 2014; Fahey et al., 2001; Tritscher et al., 2019). As simulated removal of HNO3 in the paper is mentioned, the paper should give some information how NAT sedimentation is implemented in WACCM.

**Author's Response:**

Thanks for the reviewer's suggestions. The sedimentation of $HNO_3$ in NAT aerosol follows the approach in Considine et al. (2000). The flux (F) of HNO3 can be derived as follows:

$$F = V \cdot C \cdot \exp(8\ln^2\sigma).$$

here $V$ represents the terminal velocity of NAT aerosol, $C$ denotes the condensed-phase concentration of $HNO_3$, $\sigma$=1.6 (Dye et al., 1992) represents the width of the lognormal size distribution for NAT. Similar sentences have been mentioned in P9 lines 206–210 of the revised manuscript.

**D) Ozone from sondes and simulation**

**D1)** Figure 9 (top) shows an important comparison, namely ozone sonde measurements against simulated ozone. However I suggest not showing the region below about 10 km, which is not of interest here (it also shows basically a blue area). But I think it is important to also show a plot of the differences (observations minus model) which would reveal that the model does not very well simulate to observed ozone depletion in March between 15 and 20 km. Further questions: what is the meaning of negative ozone mixing ratios (top)?

**Author's Response:**

Thanks for the reviewer's advices. The region below about 10 km has been deleted. The temporal resolution of the sounding data from March 25 to April 13, 2020, is once per day, whereas the others are normally once per 3 d during the spring and once per week during the other seasons. The other missing days were obtained by interpolation, so we did not show a plot of the differences (observations minus model). The negative ozone mixing ratio occurs

because of the range of colour scale settings and it has been revised.

**D2)** WACCM seems to overestimate temperatures at about 25 km – is this a real effect?

**Author's Response:**
The SD-WACCM simulated temperatures were generally 0.6–3 K higher than the MLS temperatures between 100 and 1 hPa. For SD-WACCM, because heterogeneous chemistry is temperature-dependent, the model generally overestimated HCl and underestimated ClO in the lower stratosphere during winter, implying insufficient chlorine activation. Similar conclusions have also been reported by previous studies (Brakebusch et al., 2013; Solomon et al., 2015; Pan et al., 2018).

**E) Formation of HCl**
The presented WACCM results suggest that the deactivation in the Arctic in 2021 is partly caused by formation of HCl, This is the classic deactivation pathway in the Antarctic, but not in the Arctic (e.g. Crutzen et al., 1992; Douglass et al., 1995; Müller et al., 2018). The authors might want to comment on this point.

**Author's Response:**
Thanks for the reviewer's advices. We indeed want to comment this point. Please see P14, lines 348–352.

"Formation of HCl is considered to be the main chlorine deactivation mechanism in Antarctica (Müller et al., 2018), but not in the Arctic. HCl increases more rapidly in the Antarctic vortex in spring than in the Arctic vortex (Douglass et al., 1995). In early March 2020, in the Arctic, chlorine was deactivated as HCl and $ClONO_2$, and the PSC that permitted chlorine activation remained. Furthermore, activated chlorine compounds were mainly deactivated as $ClONO_2$ by the $ClO + NO_2$ reaction (Müller et al., 1994; Douglass et al., 1995)."

**F) References**
Several references have been cited in this review; hopefully they are helpful. The point is not that the authors should feel obliged to cite these references. However, the paper cites WMO (2014); I suggest that a more recent ozone assessment should be used in the paper (WMO, 2018). The most recent (2022) assessment has just been released (https://ozone.unep.org/science/assessment/sap) and might be helpful when revising this paper.

**Author's Response:**
Thanks for the reviewer's suggestion. More recent references have been cited in the revised manuscript.

**G) Data availability**
The data availability statement in this paper is not good. I suggest making the DOAS observations at Ny-Ålesund available for download on a server that issues a doi and where the data are permanently archived. Such links are reported for (e.g.) SAOZ but not for the DOAS

measurements presented in the paper. Further, the WACCM data need to be better described (see above). Making data available through e-mail request is no longer recommended.

**Author's Response:**
   Thanks for the reviewer's advice. The DOAS observations at Ny-Ålesund available from https://doi.org/10.17632/jx7nkspkg7.1 and where the data are permanently archived. SAOZ data from http://saoz.obs.uvsq.fr/. The WACCM data have been further described in section 2.2 of the revised manuscript.

**Details**
• Title: I suggest avoiding "Research on" in the title; isn't this obvious? The title should rather reflect the fact that DOAS measurements from Ny- Ålesund are reported here.

**Author's Response:**
Thanks for the reviewer's suggestion. The title has been revised as "The unusual spring 2020 Arctic stratospheric ozone depletion above Ny-Ålesund by ground-based ZSL-DOAS".

• p. 1, l. 16: why this period? (I think this is the period when measurements are available, but this should be clear from the paper).

**Author's Response:**
The light intensities were strong enough during this period and the measurements were available.

• p 1, l. 21: what is a "normal year" in the Arctic?

**Author's Response:**
In this manuscript, the data for 2020 were compared with that for the other years (2017, 2018, 2019, and 2021) in the Arctic. The sentence has been revised. Please see P1, lines 19–20.

"which was about 64.7±0.1% of that in the other years (2017, 2018, 2019, and 2021)"

• p. 1, l. 21: 44.3 % → here and elsewhere in the paper: add an error estimate for the ozone loss.

**Author's Response:**
Compared with the other years, the 2020 daily peak relative ozone difference was −44.3±0.1%. Error estimates have been added elsewhere in the revised manuscript. Please see P11, lines 268–270.

"Compared to the other four years, the 2020 daily average relative differences from March 18 to April 18 from the GOME-2, ZSL-DOAS, Brewer, and SAOZ datasets were −36.5%, −35.3±0.4%, −33.1±0.7%, and −32.0±0.1%, respectively."

• p. 1, l. 23: here ans elsewhere: PV and ozone depletion: is this only a complicated way of

saying that there is no ozone loss outside the vortex? I think that Ny-Ålesund was located outside the vortex at about April 16 (see also Fig. 8.

**Author's Response:**
Thanks for the reviewer's suggestion. The Fig. 8 has been deleted. We have rewritten this part. Please see P12, lines 293–300. In addition, we reviewed the literature and found that a large and strong Arctic vortex lasted from early December anomalously into the final week of April (Kuttippurath et al., 2021). As can be seen from the figure below, Ny-Ålesund was located inside the vortex on April 17.

[Figure]

**Figure 2.** Polar vortex evolution in the Arctic winter/spring of 2019/2020. The evolution of polar vortex in the Arctic winter of 2020. The vortex situation in the lower-stratospheric altitude of about 460 K (∼ 17 km) is illustrated. The vortex edge is calculated with respect to the Nash et al. (1996) criterion at each altitude.

Figure cited from Kuttippurath et al. (2021).

"A cold and stable polar vortex is a prerequisite for ensuring that Arctic stratospheric temperatures are sufficiently low. The 2019/2020 winter was unique and the polar vortex was unusually stable, prolonged, and cold (Lawrence et al., 2020; Wohltmann et al., 2020; Rao and Garfinkel, 2020). A large and strong Arctic vortex lasted from early December anomalously into the final week of April (Kuttippurath et al., 2021). The faint planetary wave activity in the Northern Hemisphere also contributed to the formation of a cold and strong vortex (Feng et al., 2021). Unusually low temperature and strong and prolonged vortex in the 2019/2020 winter provided favourable meteorological conditions for ozone depletion in the Arctic."

• p. 1, l. 26: how new is the peak in ClO (chlorine activation)? Compare the papers in the JGR/GRL special issue?

**Author's Response:**
The peak in ClO (chlorine activation) has been discussed in the JGR/GRL special issue. However, in addition to emphasizing the reliability of our observations, we also analyzed the influence of halogen chemistry processes, particularly bromine chemistry.

• p. 2, l. 38: this is not a good description of halogen induced polar ozone loss (e.g., Müller et al., 2018, and Solomon 1999, Tritscher 2021, cited in the paper).

**Author's Response:**
The description has been revised. Please see P2, lines 48–53.

"Since the late 1970s, Antarctic stratospheric ozone during the austral spring has decreased sharply, mainly because of elevated concentrations of active chlorine (Farman et al., 1985). When the weather is cold and there is sufficient sunlight, chlorofluorocarbons derived from anthropogenic emissions can be converted to produce active chlorine, and then to maintain the chlorine activation process, which causes ozone depletion (Müller et al., 2018; Solomon, 1999; Tritscher et al., 2021).

• p 2., l. 42: 'recovery' is an important issue, it is different in the polar regions and in midlatitudes (WMO, 2018). See also further papers on the recovery of both the Antarctic ozone hole and global ozone levels (e.g., Kuttippurath and Nair, 2017; Strahan and Douglass, 2018; WMO, 2018; Bodeker and Kremser, 2021; Stone et al., 2021; Weber et al., 2022).

**Author's Response:**
Thanks for the reviewer's advice. The sentence has been revised. Please see P3, lines 53–56.

"As anthropogenic emissions of ozone-depleting substances have decreased since the Montreal Protocol was enforced, the concentrations of ozone in the Antarctic stratosphere were predicted to recover to pre-1980 values in 2060 (Solomon et al., 2016; Stone et al., 2021; Dhomse et al., 2018; Kuttippurath and Nair, 2017; Strahan and Douglass, 2018)".

• p. 2, l. 49: there should be more citations here than just Hu 2020.

**Author's Response:**
More references have been cited. Please see P3, lines 62–64.

"Between mid-February and late March 2020, the persistence of anomalously faint wave activities in the Arctic led to an abnormally persistent and cold vortex, which caused significant ozone loss (Hu, 2020; Kuttippurath et al., 2021; Ardra et al., 2022)".

• p. 3, l. 75: Simpson is on boundary layer issues: this reference needs to be changed. There are several alternative citations, already cited in the paper and there are further modelling papers cited in this review.

**Author's Response:**
This reference has been changed. Please see P4, lines 101–104.

"In addition, compared to ground-based observation, modelling provides a wider coverage and favours the investigation of ozone depletion. (Müller et al., 1994; Wohltmann et al., 2010; Griffin et al., 2019; Grooß and Müller, 2021).".

• p. 3, l. 75: These citations focus on one particular model (CLaMS), which is okay. But I think you should have citations to other models here as well (e.g., Chipperfield, 1999; Khosrawi et al., 2009; Bekki et al., 2013; Chipperfield et al., 1994; Kinnison et al., 2007; Wohltmann and

Rex, 2009; Wohltmann et al., 2010).

**Author's Response:**
Thanks for the reviewer's suggestion. More references have been cited. Please see P5, lines 104–107.

"Recently, stratospheric chemical patterns, consisting of a group of heterogeneous reactions, have been developed in various models according to investigations and experiments conducted in the polar area (McKenna et al., 2002; Grooß et al., 2011, 2018; Chipperfield, 1999; Khosrawi et al., 2009; Bekki et al., 2013; Chipperfield et al., 1994; Kinnison et al., 2007; Wohltmann and Rex, 2009)".

• p. 4, l. 99: You cannot start the Methods section with "the DOAS instrument". Which instrument? I think it is a new instrument that is described below – correct? This should be much clearer from the paper and the instrument needs to be described first before it can be "placed" somewhere. Further, given the fact that the DOAS technique is so prominent here (or should be) a bit more background on DOAS and citations (see perhaps, Huneker et al., 2017) might be appropriate.

**Author's Response:**
Thanks for the reviewer's advice. The ZSL-DOAS instrument has been further described in the revised manuscript. Please see P6 lines 135–145. We have also added a bit more background on DOAS technique. Please see P4 lines 95–97.

"The ZSL-DOAS instrument mainly includes the prism, telescope, computer, filter, motor, and CCD spectrometer. The motor controlled the telescope that can change the angle of elevation between the horizon and the zenith. As the angle of elevation changes, the telescope can acquire scattered sunlight at different angles (2°, 3°, 4°, 6°, 8°, 10°, 15°, 30°, and 90°). The quartz fibre can transform the incident light and its numerical aperture is 0.22. The light is received by the spectrometer (Ocean Optics MAYA pro) and measured by a 2048 pixels CCD. This spectrometer was designed for wavelengths between 290 and 429 nm, and had the spectral resolution (FWHM) of 0.5 nm. The integration time varied between 100 and 2000 ms due to the light intensity. The detector operates normally at approximately 20°C with a thermal controller. The mercury lamp spectra, offsets and dark currents were calibrated ahead of the experiments. The ZSL-DOAS instrument can detect $O_3$, $NO_2$, $OClO$, $BrO$, and $O_4$. The ozone slant column density (SCD) was retrieved, with the raw data obtained in the zenith direction (90°). The ZSL-DOAS instrument was placed at the Yellow River Station (78.92° N, 11.93° E) in the Arctic. Figure 1 shows the ZSL-DOAS instrument and experimental location, in Ny-Ålesund, Svalbard, Norway."

"In the 1970s, differential optical absorption spectrometry (DOAS) was developed by Platt and Stutz (2008) and has been widely used to measure several trace gases of ozone, nitrogen dioxide, bromine monoxide, and sulfur dioxide (Hüneke et al., 2017)."

• p. 6, p. 140: this sentence starts with 'parameters' but the paper should state what was actually done regarding WACCM.

**Author's Response:**
We have rewritten section 2.2. Please see response to **Comments B1**.

• p. 7, l. 166: ERA5 has 137 layers – is there a typo here?

**Author's Response:**
There is not a typo here. The ERA5 data had the spatial resolution of 0.25° × 0.25° and were divided into 37 layers vertically, from 1000 hPa to 1 hPa.

• p. 7, l. 167: where have these measurements been done?

**Author's Response:**
These measurements were carried out at Ny-Ålesund.

• p. 8, l. 173: what is a 'normal year'?

**Author's Response:**
This response is similar to **Details p 1, l. 21**.

• p. 9, l. 199: what is the 'threshold temperature'? This is an important point that should be discussed in the paper.

**Author's Response:**
Please see response to **Comments C1**.

• p. 9, l. 201: by definition the PV in the southern hemisphere is negative and positive in the northern hemisphere. This simple fact should be taken into account when making such statements.

**Author's Response:**
Thanks for the reviewer's advice. The sentence has been deleted.

• p. 10., l. 235: apparent → obvious?

**Author's Response:**
It has been revised.

• p. 10, l. 243: 'recover' is problematic here, it is not the right word to use when taking about chlorine deactivation putting a halt to ozone loss.

**Author's Response:**

It has been revised. Please see P15 lines 353–354.

"In mid-April 2020, ClONO₂ stopped increasing and ClO was almost depleted when the ozone concentration started to increase."

• p. 10., l. 237: it is not only the reaction HCl + ClONO₂

**Author's Response:**
The heterogeneous reactions HCl + ClONO₂ and HOCl + HCl and the gas-phase reaction CH₃O₂ + ClO contributed to the conversion of HCl to active chlorine (Müller et al., 1994; Müller et al., 2018). And the sentence has been revised. Please see P14 lines 345–346.

"Chlorine was dominantly activated by ClONO₂ + HCl and this reaction improved up to 10 times when the temperature reduced by 2.3 K (Wegner et al., 2012)."

• Figure 5: cold the errors of the individual measurements be used for weighting the data when calculating regression etc.?

**Author's Response:**
Thanks for the reviewer's suggestion. We added the errors of the individual measurements be used for weighting the data when calculating regression etc. No errors were provided from the GOME-2 dataset, so we did not consider measurement errors of GOME-2. Please see P35.

[Figure]

Figure 6. Scatter plots and linear fits of retrieved ozone VCDs with (a) GOME-2, (b) Brewer, and (c) SAOZ.

• Figure 6: Show error bars?

**Author's Response:**
The figure has been revised. Please see P34.

[Figure]

Figure 5. (a) Ozone data for 2020 and the average ozone data (black) of 2017, 2018, 2019, and 2021. (b) relative ozone difference for 2020.

• Figure 7: the blue line shows 195 K, which is an approximation for the onset temperature for heterogeneous chemistry.

**Author's Response:**
Thanks for the reviewer's suggestions. Please see response to **Comments C1**.

• Figure 8: show error bars?

**Author's Response:**
The figure has been deleted. Please see response to **Details p 1, l. 23**.

**Response to Reviewer #2**

**General**

In the current manuscript, Li et al has introduced the retrieval method (ZSL-DOAS) by the ground-based instrument installed at Yellow River Station in the Arctic. Then they have validated the retrieved ozone VCD with some widely used measurements including GOME2, Bremer spectrophotometers and SAOZ, the latter uses DOAS with precise measurements of stratospheric constituents during twilight. All these show the observed ozone value is much lower in late winter/early spring in 2020 comparing with the recent five years. Then the authors have investigated the daily variability in ozone changes by calculating the 4-year mean ozone value (2017-2021 excluding the extreme year 2020) and the absolute (and the relative percentage) difference with respace to the mean (?) and have tried to link it to the dynamics (for example by looking at the temperature, PV evolutions). They have also done a model simulation (SD-WACCM) to look at the model performance and the chemical species changes. However, the paper is not well written though the message is still clear for me. There are many confusions (for example, definitions like ozone loss etc. that are quite different from the community has used). It is no doubt that the current ZSL-DOAS observations give another evidence for the unusual Arctic 2020 spring ozone and the unique dataset will be of interest to the atmospheric community. There are many studies over many years show that chlorine and bromine compounds are responsible for the polar ozone depletion in winter and spring. However, the current manuscript has not provided the firm conceptual advance in our understanding of Arctic ozone depletion and there is no new insight into the underlying mechanism responsible for the Arctic ozone depletion. Therefore, the paper has to be rejected or rewritten to find something new.

**Author's Response:**

We would like to thank the reviewer #2 for the careful and valuable comments, which enable us to improve our study and the manuscript remarkably. Please kindly find our point-to-point response to the problems/comments below in blue and the change of the manuscript in orange.

Figure 6 shows indeed not the ozone loss, but the ozone difference between 2020 and the 4-year mean. It has been revised. Please see P34. There are many studies on the Arctic winter 2019/2020 already, so we have presented this work as a Measurement Report, in which the measurements are reported and the consistency with other studies and measurements are shown.

[Figure]

Figure 5. (a) Ozone data for 2020 and the average ozone data (black) of 2017, 2018, 2019, and 2021. (b) relative ozone difference for 2020.

**Specific Comments:**

1. The title is too general. "Research" is quite broad.

**Author's Response:**
Thanks for the reviewer's suggestion. The title has been revised as "The unusual spring 2020 Arctic stratospheric ozone depletion above Ny-Ålesund by ground-based ZSL-DOAS".

2. Line 13 in the Abstract: "Severe", the "third and most severe" is vague. There are still larger Arctic ozone loss for other years. Do you mean the long-lasting cold polar vortex years?

**Author's Response:**
The sentence has been revised. Please see P1 lines 15–16.

"Of the severe stratospheric ozone depletion events (ODEs) reported over the Arctic, the most severe occurred during the spring of 2020."

3. Actually, I find "event" is confusing.

**Author's Response:**

Many studies have reported this event about unprecedented Arctic ozone depletion in the year 2020 (Dameris et al., 2021; Feng et al., 2021; Manney et al., 2020; Wohltmann et al., 2020).

4. There are many "normal" in the whole text. What is the definition for the normal year? Need to be clear with it.

**Author's Response:**

In this manuscript, the data for 2020 were compared with that for the other years (2017, 2018, 2019, and 2021) in the Arctic. The sentence has been revised. Please see P1 lines 18–20.

"The average ozone VCD over Ny-Ålesund between March 18 and April 18, 2020, was approximately 274.8 Dobson units (DU), which was about 64.7±0.1% of that in the other years (2017, 2018, 2019, and 2021)"

5. Some results are obvious: "effect of the polar vortex on stratospheric ozone depletion", "Chlorine activation" and "bromine compounds"

**Author's Response:**

Thanks for the reviewer's suggestion. We have presented this work as a Measurement Report, in which the measurements are reported and the consistency with other studies and measurements are shown.

6. The last sentence in the abstract. What is the main point here? Is this relevant to this work?

**Author's Response:**

The sentence has been deleted. We have rewritten the part in the revised manuscript. Please see P2 lines 38–41.

"By ZSL-DOAS observations, we provided another evidence for unprecedented ozone depletion during the Arctic spring of 2020. The ZSL-DOAS ozone VCD observations can also provide calibration for satellite observations and model simulations, and in the future can provide the support for observations at more Chinese research stations or international local stations in the polar area."

7. Introduction is not well written. Most of them are too general and the background is well known. If you focus on Arctic winter/spring 2020, then you need to brief summary the available publications and what are the unique research questions you need to address or the methods you have applied.

**Author's Response:**

Thanks for the reviewer's suggestion. The introduction has been added. Please see P2–5 lines 44–126.

"Stratospheric ozone is essential for human health, surface ecosystems, and the climate in general (McKenzie et al., 2011) because it absorbs ultraviolet (UV) solar radiation and converts it into thermal energy. The characteristic absorption bands of stratospheric ozone are mainly located in the Hartley and Huggins zones of the UV region and in the Chappuis zone of the visible spectrum, thereby absorbing almost all UV-C (i.e., wavelengths < 280 nm) and some UV-B (i.e., wavelengths ranging between 280 and 315 nm) radiation. Since the late 1970s, Antarctic stratospheric ozone during the austral spring has decreased sharply, mainly because of elevated concentrations of active chlorine (Farman et al., 1985). When the weather is cold and there is sufficient sunlight, chlorofluorocarbons derived from anthropogenic emissions can be converted to produce active chlorine, and then to maintain the chlorine activation process, which causes ozone depletion (Müller et al., 2018; Solomon, 1999; Tritscher et al., 2021).. As anthropogenic emissions of ozone-depleting substances since the Montreal Protocol was enforced, the concentrations of ozone in the Antarctic stratosphere were predicted to recover to pre-1980 values in 2060 (Solomon et al., 2016; Stone et al., 2021; WMO, 2018; Kuttippurath and Nair, 2017; Strahan and Douglass, 2018).

The severe ozone depletion over the Arctic is relatively uncommon compared with that in the Antarctic. During normal Arctic winters, the polar vortex usually fractures and disperses early due to huge planetary wave activities and Brewer–Dobson circulation dynamics (Manney et al., 2003; Dameris, 2010; Harris et al., 2010). Thus, in the Arctic, the duration of the vortex is shorter and relative ozone loss is also lower (Solomon et al., 2007). However, irregular changes in Arctic ozone in recent years have attracted worldwide attention and challenged the existing model. The most severe Arctic ozone depletion lasted for nearly a month, from March to April 2020 (Dameris et al., 2021). Between mid-February and late March 2020, the persistence of anomalously faint wave activities in the Arctic led to an abnormally persistent and cold vortex, which caused significant ozone loss (Hu, 2020; Kuttippurath et al., 2021; Ardra et al., 2022). This event was the most severe reported low Arctic ozone event, following those that occurred in the springs of 1997 and 2011 (Hansen and Chipperfield, 1999; Manney et al., 2011).

The powerful and persistent vortex during the winter and spring is considered as a main cause of significant ozone depletion in the Arctic (Bognar et al., 2021). Polar stratospheric clouds (PSCs) are classified into three types: nitric acid trihydrate (NAT), ice PSCs, and supercooled ternary solution (STS), and their threshold temperatures for existence are $T_{nat}$ (195 K), $T_{ice}$ (188 K), and $T_{sts}$ (195–197 K), respectively (Toohey et al., 1993; Poole and McCormick, 1988; Solomon, 1999). Extremely low air temperatures are essential to produce PSC. The PSC can be used as a surface for heterogeneous interactions, leading to the conversion of reactive halogens from the halogen reservoirs, which can cause serious ozone loss (Frieβ et al., 2005; Marsing et al., 2019). Although the PSC is not only composed of NAT (Pitts et al. 2009; Spang et al. 2018), the temperature threshold for the existence of NAT provides a good estimate on the occurrence of heterogeneous chemistry (Drdla and Müller 2012; Kirner et al. 2015; Grooß and Müller 2021; von der Gathen et al. 2021). PSC might also grow large enough to precipitate and remove $HNO_3$ in the stratosphere, which is the reservoir of $NO_2$. The resulting denitrification from the polar vortex hinders chlorine deactivation by $NO_2$ (Salawitch et al., 1989; Arblaster et al. 2014). Active chlorine is rapidly photolyzed because of the recovery of spring sunlight when ozone loss occurs via the self-reaction of ClO (Molina and Molina, 1987), as well as the cross-reaction of ClO and BrO (McElroy et al., 1986). It is essential that the

vortex retains low temperatures and carries on as a transport impediment so that ozone can remain depleted without $NO_2$ to inactivate chlorine.

The observed Arctic ozone depletion is invaluable for validating stratospheric ozone simulations and for understanding the processes that cause Arctic stratospheric ozone depletion. Currently, ozone vertical column density (VCD) detection utilizes the characteristic ozone absorption in the UV and visible spectra, which provides accurate ozone identification and quantitative measurements. Ground-based observation of ozone VCD started in the first decades of the twentieth century (Dobson, 1968; Brewer, 1973; Solomon et al., 1987; Bognar et al., 2021). From the 1960s, ozonesondes began to acquire atmospheric ozone data (Logan, 1994; Thomason et al., 2011; Wohltmann et al., 2020; Grooß and Müller, 2021).Since 1978, satellite observations have provided essential data for atmospheric ozone related studies (Kuttippurath et al., 2012; Manney et al., 2020). Among these, ground-based observations are crucial to calibrate remotely sensed observations and optimizing inversion results (Lu et al., 2006). In the 1970s, differential optical absorption spectrometry (DOAS) was developed by Platt and Stutz (2008) and has been widely used to measure several trace gases of ozone, nitrogen dioxide, bromine monoxide, and sulfur dioxide (Hüneke et al., 2017).

The impacts of the aberrantly powerful and persistent vortex on ozone in the Arctic were investigated using satellite observations, ozonosonde measurements, and data from the European Centre for Medium-Range Weather Forecasts (ECMWF) (Wohltmann et al., 2020; Lawrence et al., 2020). The major stratospheric halogen species, chlorine, and bromine were investigated in this ozone depletion event (ODE) (Wohltmann et al., 2017, 2021). In addition, compared to ground-based observation, modelling provides a wider coverage and favours the investigation of ozone depletion. (Müller et al., 1994; Wohltmann et al., 2010; Griffin et al., 2019; Grooß and Müller, 2021). Recently, stratospheric chemical patterns, consisting of a group of heterogeneous reactions, have been developed in various models according to investigations and experiments conducted in the polar area (McKenna et al., 2002; Grooß et al., 2011, 2018; Chipperfield, 1999; Khosrawi et al., 2009; Bekki et al., 2013; Chipperfield et al., 1994; Kinnison et al., 2007; Wohltmann and Rex, 2009). Global and area models using different stratospheric chemical patterns have been applied to simulate ozone columns, which usually compare well with satellite observations and ozonosonde data (Pan et al., 2018; Grooß and Müller, 2021).

Accurate ground-based observations can improve the accuracy and reliability of models as well as enhancing our understanding of the reasons for ozone depletion. In this study, we have developed a ground-based DOAS system that can conduct ozone VCD observations in the Arctic. The zenith scattered light observation mode was applied to measure ozone VCD using the Langley Plot method (Frieß et al., 2005).

We analyze the reasons for this ODE in the unusual spring of 2020 above Ny-Ålesund, Norway. The methods and data are given in Sect. 2, which covers the presentation of the experimental location and DOAS instrument, the DOAS method, the specified dynamics version of the Whole Atmosphere Community Climate Model (SD-WACCM), Global Ozone Monitoring Experiment 2 (GOME-2) observations, Brewer measurements, Système d'Analyze par Observation Zénithale (SAOZ) measurements, ECMWF data, and ozonesonde data. Section 3 presents the results, where Sect. 3.1 describes the results of ozone VCDs from February 2017 to October 2021 and ozone difference in spring 2020. The zenith scattered light DOAS (ZSL-

DOAS) retrieved the daily variations in ozone VCDs, which were in comparison with GOME-2 observations, Brewer, and SAOZ measurements. A detailed characterization of this ODE is presented for establishing the basis of the subsequent analysis. The relationship between Arctic ozone depletion and meteorological conditions in terms to temperature and potential vorticity (PV) is described in Sect. 3.2. In Sect. 3.3, this ODE was analyzed using the SD-WACCM to further illustrate the ozone depletion process, and to explore the effects of chemical depletion and dynamic transport on this ODE. The influence of the halogen species is discussed in Sect. 3.4. The comprehensive summary is provided in Sect. 4."

8. I also find it is difficult to see the purpose of this work. Are you aiming to validate your ZSL-DOAS observations using other measurements? I am not an expert in the retrieval method so it is hard for me to judge your method. What is the difference in the retrieval of ozone from ZSAL-DOAS and SAOZ because SAOZ also uses DOAS method? For the Air Mass Factor (AMF) in section 2.2, the authors have mentioned the related parameters in Table 2 and AMF will be quite different for different assumed profiles (for example Lines 121-123). So I am curious what are these profiles from. It is vague just to mention SCIATRAN without any reference there.

**Author's Response:**
Thanks for the reviewer's advice. We have presented this work as a Measurement Report, in which the measurements are reported and the consistency with other studies and measurements are shown. Both ZSL-DOAS and SAOZ use the DOAS method, but ZSL-DOAS is an instrument self-developed by our group. A priori ozone profile is obtained from the monthly mean climatology. The sentence has been revised. Please see P7 lines 164–167.

"Here, the Air Mass Factor (AMF) can be obtained from the SCIATRAN model and is influenced by a priori ozone profile, SZA (solar zenith angle), wavelength, and surface albedo. Based on the average monthly climate, a priori ozone profile can be achieved. The SZA calculated in this research ranged between 35° and 80°, with surface albedos between 0.08 and 0.6. Table 2 lists these parameters."

9. SD-WACCM. This section needs some more details since the authors have carried out the simulation. It should come from CESM1 (but which version) but not sure what other changes have been made by the author. I assume this is a released version but ported and run on a different HPCx in China.

**Author's Response:**
Thank for your suggestion. We have rewritten this section and added description on the WACCM model. In this study, we replace the standard polar stratospheric cloud module with that from Wegner et al. (2013). Similar sentences have been written in P9 lines 204–206.

"The polar stratospheric cloud module used in this study followed Wegner et al. (2013) rather than the standard module of Kinnison et al. (2007), improving the capabilities of WACCM in modelling ozone and its associated components (Brakebusch et al., 2013)."

10. However, some of the emissions and other input data have not been updated/available for year 2020. One simple example is that this version uses the prescribed stratospheric sulphur aerosol density (SAD) which is important for the ozone depletion. How the SAD used in SD-WACCM4 for this? Do you use the previous year's values or fixed values? Which component you are using? It seems that the authors only mentioned MOZART3, so I am not sure if this one "pp_waccm_mozart" used or you also have MAM model in the model simulation etc..

**Author's Response:**
We have rewritten this section and added description on the WACCM model. Please see P8 and P9, lines 191–206.

"The chemical mechanism of WACCM4 also contains 4 aerosol types heterogeneous reactions: liquid binary sulfate (LBS), supercooled ternary solution (STS), nitric acid trihydrate (NAT), and water-ice. When model temperatures above 200K, only the LBS exists. The surface area density (SAD) of LBS is from SAGE, SAGE-II and SAMS observations (Thomason et al., 1997) and Considine update it (World Meteorological Organization, 2003). With the model atmosphere cooling, the LBS aerosol expands and absorbs both $HNO_3$ and $H_2O$ to obtain the STS aerosol. Tabazadeh et al. (1994) derived the composition of STS by the Aerosol Physical Chemistry Model (ACPM). The STS aerosol median radius and SAD is derived following the approach of Considine et al. (2000). When model temperatures reach a specified supersaturation ratio of $HNO_3$ for NAT, $HNO_3$ containing aerosols are allowed to form. In WACCM4, Peter et al. (1991) set this ratio to 10. NAT median radius and SAD are derived in the same way with STS aerosol. If the derived atmospheric temperature does not exceed the saturation temperature of water vapour on ice (Tsat), then this results in the formation of water-ice aerosols. In WACCM4, the CAM's prognostic water routines gives the condensed phase $H_2O$, which is conveyed to the chemistry module. According to the method of Considine et al. (2000), the median radius and SAD of water-ice can be derived by this condensed phase $H_2O$. The polar stratospheric cloud module used in this study followed Wegner et al. (2013) rather than the standard module of Kinnison et al. (2007), improving the capabilities of WACCM in modelling ozone and its associated components (Brakebusch et al., 2013)."

11. For the SD, the author need to realize that this is not a fully nudged version, which all depends on the relaxation time etc. This needs to make clear.

**Author's Response:**
Thank for your suggestion. We have added some description of the calculation of the meteorological fields. Please see P9, lines 214–221.

"Meteorological fields were calculated using a nudging method in the model (Lamarque et al., 2012). Data for the horizontal winds, temperature, and surface pressure from MERRA-2 were used to drive the physical parameterization from the surface to 50 km (Kunz et al., 2011), which allowed for more accurate comparisons between the measurements of atmospheric composition and the model output (Lamarque et al., 2012). This can be employed for the study of specific

weather events. Linear transitions were used in the 50–60 km altitude range and over 60 km, and online calculations were performed. In this study, the MERRA-2 dataset has the same resolution with the SD-WACCM, which can be accessed on the Earth System Grid (https://www.earthsystemgrid.org/home.html) and are obtained from the original resolution ($1/2°×2/3°$) by a conservative re-gridding procedure (Lamarque et al., 2012; Pan et al., 2019)."

12. I have not heard "thermogenic" layer, should be use the correct term like "lower thermosphere". It is strange to say the "parameters" in the Line 140. Please note it uses CAM4 physics.

**Author's Response:**
Thank for your suggestion. The word has been revised and we have rewritten the section 2.2. The WACCM4 model is based on the physical parameterizations used in the Community Atmosphere Model version 4 (CAM4) (Neale et al., 2013).

13. It is impropriate to say "the SD-WACCM with meteorological parameters driven by …", note even SD-WACCM most of the temperature etc. are still from WACCM itself (driven by SST, solar etc.…). How can you say "Data from MERRA-2 guaranteed the accuracy of simulated values"? Note that other processes play roles.

**Author's Response:**
we have rewritten the section 2.2. These sentences have been corrected. Please see P9, lines 215–217.

"Data for the horizontal winds, temperature, and surface pressure from MERRA-2 were used to drive the physical parameterization from the surface to 50 km (Kunz et al., 2011), which allowed for more accurate comparisons between the measurements of atmospheric composition and the model output (Lamarque et al., 2012)."

14. Auxiliary data: Can be concise and have proper references. For example ERA5. Some sentences are not necessary at all. For example. Lines 152, 165.

**Author's Response:**
Thanks for the reviewer's suggestion. Some sentences have been deleted.

15. It reads to me that the authors just SHOW the results itself, rather than describe the results in a correct/proper way. For example, for Figure4, we can see large daily variability that has never mentioned. We also see the differences among these measurements but have never explained. For example, why your data ZSL-DOAS is much lower than other observations? I don't think the gradient of "~0.92DU per day" is similar for all the "normal" years claimed. It is also not correct to say "Ozone VCD begins to decrease in March."

**Author's Response:**
Thanks for the reviewer's advice. During the observation period, the average ozone VCD from

ZSL-DOAS was 2.25% lower than that from GOME-2, 4.92% lower than that from Brewer, and 2.26% higher than that from SAOZ. These differences of ZSL-DOAS observations compared to other observations are due to systematic errors of the instrument. These descriptions have been revised. Please see P11, lines 257–261.

"In the other years (2017, 2018, 2019, and 2021), ozone VCD showed a fluctuating downward trend between March and September, with a small upward trend around March and August. In 2020, however, severe ozone depletion occurred between March 18 and April 18, after which ozone VCD gradually increased. Ozone VCD decreased further in mid-May. In around September, the ozone VCD increased obviously again, probably due to clear warming of the polar stratosphere."

16. For Figure 5, I understand the VCD and TCO can be used but the authors need to be consistent in the whole text.

**Author's Response:**
It has been revised. We have used VCD in the whole text.

17. The presentation for Figure 6 is not good at all. Why the authors term this "ozone loss"? How do you estimate "ozone loss"? It looks that this is just ozone difference between 2020 and the 4-year mean.

**Author's Response:**
Figure 6 shows indeed not the ozone loss, but the ozone difference between 2020 and the 4-year mean. It has been revised. Please see response to **General** above.

18. For Figure 7, it seems that temperature is from ERA5, why use "measured" in Line 190? It is a reanalysis product, which is from ECMWF model simulation using the data assimilation from the measurements.

**Author's Response:**
It has been revised. Please see P12, lines 283–284.

"Daily average temperatures of Ny-Ålesund between November 2016 and September 2021 were showed at 70 hPa in the low stratosphere, where significant ozone depletion tends to occur (Fig. 7)."

19. Figure 8, it seems that the authors just look at one time period of ozone and PV evolution, then comes the conclusion of "PV correlates negatively with ozone VCD" etc.  The authors seems not have a deep understanding their figures even they made it (for example, stratospheric warming after mid-April that made polar vortex weaker and TOC higher) etc..

**Author's Response:**
Thanks for the reviewer's suggestion. The Fig. 8 has been deleted. We have rewritten this part.

"A cold and stable polar vortex is a prerequisite for ensuring that Arctic stratospheric temperatures are sufficiently low. The 2019/2020 winter was unique and the polar vortex was unusually stable, prolonged, and cold (Lawrence et al., 2020; Wohltmann et al., 2020; Rao and Garfinkel, 2020). A large and strong Arctic vortex lasted from early December anomalously into the final week of April (Kuttippurath et al., 2021). The faint planetary wave activity in the Northern Hemisphere also contributed to the formation of a cold and strong vortex (Feng et al., 2021). Unusually low temperature and strong and prolonged vortex in the 2019/2020 winter provided favourable meteorological conditions for ozone depletion in the Arctic."

20. Figure9, the authors said "unusual low". This is only for 2020, have you made O3 volume mixing ration comparison with others.

**Author's Response:**

I made O₃ volume mixing ration comparison with that of 2017,2018, and 2019 from the ozonesonde. Besides, I have reviewed relevant literatures that reported unprecedented Arctic ozone depletion in the year 2020 (Dameris et al., 2021; Feng et al., 2021; Manney et al., 2020; Wohltmann et al., 2020).

[Figure]

Figure A1. Between January 1 and July 1, ozone profiles of 2017 (a), 2018 (b), 2019 (c), and 2020 (d) from ozonesonde measurements.

21. Why "< 0.5ppmv suggested the ozone was nearly completed depleted. "?.

**Author's Response:**
By reviewing the literature, "Mixing ratios were consistently below 0.5 ppmv in a wide altitude range (with minima below 0.2 ppmv), indicating near-complete depletion of ozone (Bognar et al., 2021)".

22. For the Tnat, it also depends on $H_2O$, $HNO_3$ and $H_2SO_4$? What is their values used for the Tnat?

**Author's Response:**
Please see P3, lines 68–76.

"Polar stratospheric clouds (PSCs) are classified into three types: nitric acid trihydrate (NAT), ice PSCs, and supercooled ternary solution (STS), and their threshold temperatures for existence are $T_{nat}$ (195 K), $T_{ice}$ (188 K), and $T_{sts}$ (195–197 K), respectively (Toohey et al., 1993; Poole and McCormick, 1988; Solomon, 1999). Extremely low air temperatures are essential to produce PSC. The PSC can be used as a surface for heterogeneous interactions, leading to the conversion of reactive halogens from the halogen reservoirs, which can cause serious ozone loss (Frieβ et al., 2005; Marsing et al., 2019). Although the PSC is not only composed of NAT (Pitts et al. 2009; Spang et al. 2018), the temperature threshold for the existence of NAT provides a good estimate on the occurrence of heterogeneous chemistry (Drdla and Müller 2012; Kirner et al. 2015; Grooß and Müller 2021; von der Gathen et al. 2021)."

23. Figure 10, why "$HNO_3$ changes abruptly from abnormally high values to normal values, which indicated the abundant PSC activities of the period" only applies for "Between late January and early February"? What caused the low value patches around 20-22km?

**Author's Response:**
It also applies other period and the sentence has been revised. The low value of HNO3 at 20-22 km is probably due to low temperatures (Fig. 9c–d) leading to PSC activity and severe denitrification (Ardra et al., 2022). Please see P14, lines 335–337.

"$HNO_3$ changed abruptly from abnormally high values to normal values, which indicated the abundant PSC activities of the period (Bognar et al., 2021). The low value of $HNO_3$ at 20-22 km is probably due to low temperatures (Fig. 8c–d) leading to PSC activity and severe denitrification (Ardra et al., 2022)."

24. Figure 11, I saw the model showed a complete HCl depletion on 21 Feb. How the modelled HCl etc. chemical species compared with ACE observation for example?

**Author's Response:**
The figure displays the simulated average diurnal mixing ratios of ozone, chlorine, and bromine compounds for heights of 17.5 km above Ny-Ålesund, but ACE observations do not have the corresponding altitude. Then, we reviewed the literature and also found that a complete HCl depletion around 20 Feb (Grooß and Müller 2021). Furthermore, the reliability of the model

can be validated by comparing it with ozone sonde measurements.

[Figure]

**Figure 10.** Fifty day development of one example air parcel trajectory from the CLaMS simulation that is not affected by mixing over the time period shown. The top panel shows the temperature along the air parcel trajectory (black). The periods when heterogeneous HCl loss rates ($k_{HCl + ClONO2}$ [ClONO$_2$] + $k_{HCl + HOCl}$ [HOCl]) are larger than 1 day$^{-1}$ are marked as gray shaded areas. The middle panel shows the mixing ratio of the chlorine compounds HCl, ClONO$_2$, ClO$_x$ (= ClO + 2 × Cl$_2$O$_2$ + 2 × Cl$_2$) and HOCl. Ozone is shown in the lowest panel on a logarithmic ordinate. The minimum ozone mixing ratio on 24 March (84°N, 131°E, $\theta$ = 439 K) is 38 ppbv.

Figure cited from Grooß and Müller. (2021).

25. This reads like a summary from the main text. I am not sure why the last sentence is matter based on this study.

**Author's Response:**
Thanks for the reviewer's suggestion. The conclusion part has been revised. Please see P16 and P17, lines 376–414.

"In this research, the ozone VCD was obtained from a ground-based instrument, the GOME-2 satellite, and the Brewer and SAOZ instruments and further evaluated with a correlation analysis. The Pearson correlation coefficients were 0.97, 0.87, and 0.91, and the relative deviations were 2.3%, 3.1%, and 3.5%, respectively. Therefore, we can conclude that the method of observing the VCDs of Arctic ozone using a ground-based DOAS instrument is reliable and valid. Compared to the other four years, the 2020 daily average relative differences from March 18 to April 18 from the GOME-2, ZSL-DOAS, Brewer, and SAOZ datasets were −36.5%, −35.3±0.4%, −33.1±0.7%, and −32.0±0.1%, respectively. The results indicated that all instruments recorded severe ozone depletion from March 18 to April 18, 2020.
Unusually low temperature and strong and prolonged vortex in the 2019/2020 winter provided favourable meteorological conditions for ozone depletion in the Arctic. The ozone and temperature profiles were simulated by SD-WACCM, and these simulations corresponded well with ozonesonde measurements. The model results show that ozone depletion at a height range

of 16–20 km is evident from late March to early April, which corresponds to the ozone VCDs obtained from the ground-based instrument. Chlorine and bromine activation were clearly obvious during the Arctic spring of 2020, whereas the partitioning of bromine compounds was different from that of chlorine. Chlorine was predominantly present as HCl and $ClONO_2$ before activation, whereas bromine was predominantly present as HOBr and BrCl before activation. Particularly, bromine existed mainly as HOBr before chlorine activation began. When chlorine was activated, bromine existed mainly as BrCl. In addition, formation of HCl is considered to be the main chlorine deactivation mechanism in Antarctica (Müller et al., 2018). However, in the Arctic, due to HCl increased more slowly than in the Antarctic, chlorine was mainly deactivated as $ClONO_2$.

In summary, by ZSL-DOAS observations, we provided another evidence for unprecedented ozone depletion during the Arctic spring of 2020. The ZSL-DOAS ozone VCD observations can also provide calibration for satellite observations and model simulations, and in the future can provide the support for observations at more Chinese research stations or international local stations in the polar area. Additionally, although WACCM can depict the evolution of ozone during this Arctic ozone depletion event, there are some problems such as overestimation of the temperature and the $CH_3O_2$+ClO reaction is not considered in the current chemical mechanism of the model. This could be considered in future models to improve the simulation performance."

**Research on the unusual spring 2020 Arctic stratospheric ozone depletion above Ny-Ålesund, NorwayMeasurements report: The unusual spring 2020 Arctic stratospheric ozone depletion above Ny-Ålesund by ground-based ZSL-DOAS**

Qidi Li[1,2], Yuhan Luo[1*], Yuanyuan Qian[1,2], Chen Pan[3,4], Ke Dou[1], Xuewei Hou[5], Fuqi Si[1], Wenqing Liu[1]

[1]Key Laboratory of Environmental Optics and Technology, Anhui Institute of Optics and Fine Mechanics, Hefei Institutes of Physical Science, Chinese Academy of Sciences, Hefei, 230031, China
[2]University of Science and Technology of China, Hefei, 230026, China
[3]Jiangsu Meteorological Observatory, Jiangsu Meteorological Bureau, Nanjing, 210008, China
[4]Key Laboratory of Transportation Meteorology, China Meteorological Administration, Nanjing, 210009, China
[5]Collaborative Innovation Center on Forecast and Evaluation of Meteorological Disasters, Nanjing University of Information Science and Technology, Nanjing, 210044, China

*Correspondence to*: Yuhan Luo (yhluo@aiofm.ac.cn)

**Abstract.** Of the severe stratospheric ozone depletion events (ODEs) reported over the Arctic, the third and most severe occurred during the spring of 2020; we analyzed the reasons for this event herein. We retrieved the critical indicator ozone vertical column density (VCD) using zenith scattered light differential optical absorption spectroscopy (ZSL-DOAS) from March 2017 to September 2021 located in Ny-Ålesund, Svalbard, Norway. The average ozone VCDs over Ny-Ålesund between March 18 and April 18, 2020, were was approximately 274.8 Dobson units (DU), which was only about $64.7 \pm 0.1$%

of that in the other years (2017, 2018, 2019, and 2021)normal years, and the daily peak difference was 195.7 DU during this period. The retrieved daily averages of ozone VCDs were compared with satellite observations from Global Ozone Monitoring Experiment Experiment-2 (GOME-2), a Brewer spectrophotometer, and a Système d'Analyze par Observation Zénithale (SAOZ) spectrometer at Ny-Ålesund; the resulting Pearson correlation coefficients were relatively high at 0.9497, 0.8687, and 0.91, with relative deviations of 2.3%, 3.1%, and 3.5%, respectively. Polar observations are still inadequate and accurate ZSL-DOAS observations can provide reliable data for polar ozone study. Compared with normal years, the 2020 daily peak relative ozone loss was 44.3%. During the 2020 Arctic spring ODE, the ozone VCDs and potential vorticity (PV) had a negative correlation with their fluctuations, suggesting a clear effect of the polar vortex on stratospheric ozone depletion. To better understand what caused the ozone depletion, We analyzed the relationship between Arctic ozone

depletion and meteorological conditions in terms to temperature and potential vorticity (PV). We also considered the

30 chemical components of this process in the Arctic winter of 2019/2020 with the specified dynamics version of the Whole Atmosphere Community Climate Model (SD-WACCM). The SD-WACCM results indicated that both ClO and BrO concentrations peaked in late March, which was a critical factor during the ozone depletion observed in Ny-Ålesund. Chlorine and bromine activation were clearly obvious during the Arctic spring of 2020, whereas the partitioning of bromine compounds was different from that of chlorine. Before activation, chlorine was predominantly present as HCl

35 and ClONO$_2$, whereas bromine was predominantly present as HOBr and BrCl. Particularly, before chlorine activation began, bromine mainly existed as HOBr; however, after chlorine activation, bromine mainly existed in the form of BrCl. By combining observations with modeling, we provide a reliable basis for further research on global climate change due to polar ozone concentrations and the prediction of severe Arctic ozone depletion in the future. By ZSL-DOAS observations, we provided another evidence for unprecedented ozone depletion during the Arctic spring of 2020. The ZSL-DOAS ozone VCD

40 observations can also provide calibration for satellite observations and model simulations, and in the future can provide the support for observations at more Chinese research stations or international local stations in the polar area.

**Key Words:** Arctic ozone depletion, DOAS, ozone VCD, polar vortex, SD-WACCM, halogen species

**1 Introduction**

Stratospheric ozone is essential for human health, surface ecosystems, and the climate in general (McKenzie et al., 2011)

45 because it absorbs ultraviolet (UV) solar radiation and converts it into thermal energy. The characteristic absorption bands of stratospheric ozone are mainly located in the Hartley and Huggins zones of the UV region and in the Chappuis zone of the visible spectrum, thereby absorbing almost all UV-C (i.e., wavelengths < 280 nm) and some UV-B (i.e., wavelengths ranging between 280 and 315 nm) radiation. Since the late 1970s, Antarctic stratospheric ozone during the austral spring has decreased sharply, mainly because of elevated concentrations of active chlorine (Farman et al., 1985).the heterogeneous

50 catalytic reactions between ozone and active halogen radicals generated by the conversion of When the weather is cold and there is sufficient sunlight, chlorofluorocarbons derived from anthropogenic emissions can be converted to produce active chlorine, and then to maintain the chlorine activation process, which causes ozone depletion (Müller et al., 2018; Solomon,

1999; Tritscher et al., 2021).. As anthropogenic emissions of ozone-depleting substances since the Montreal Protocol was enforced, the concentrations of ozone in the Antarctic stratosphere were predicted to recover to pre-

55  1980 values in 2060 (Solomon et al., 2016; Stone et al., 2021; WMO, 2018; Kuttippurath and Nair, 2017; Strahan and Douglass, 2018).

The severe ozone depletion over the Arctic is relatively uncommon compared with that in the Antarctic. During normal Arctic winters, the polar vortex usually fractures and disperses early due to huge planetary wave activities and Brewer–Dobson circulation dynamics (Manney et al., 2003; Dameris, 2010; Harris et al., 2010). Thus, in the Arctic, the duration of

60  the vortex is shorter and relative ozone loss is also lower (Solomon et al., 2007). However, irregular changes in Arctic ozone in recent years have attracted worldwide attention and challenged the existing model. The most severe Arctic ozone depletion lasted for nearly a month, from March to April 2020 (Dameris et al., 2021). Between mid-February and late March 2020, the persistence of anomalously faint wave activities in the Arctic led to an abnormally persistent and cold vortex, which caused significant ozone loss (Hu, 2020; Kuttippurath et al., 2021; Ardra et al., 2022). This event was the most

65  severe reported low Arctic ozone event, following those that occurred in the springs of 1997 and 2011 (Hansen and Chipperfield, 1999; Manney et al., 2011).

The powerful and persistent vortex during the winter and spring is considered as a main cause of significant ozone depletion in the Arctic (Bognar et al., 2021). Polar stratospheric clouds (PSCs) are classified into three types: nitric acid trihydrate (NAT), ice PSCs, and supercooled ternary solution (STS), and their threshold temperatures for existence are $T_{nat}$ (195 K),

70  $T_{ice}$ (188 K), and $T_{sts}$ (195–197 K), respectively (Toohey et al., 1993; Poole and McCormick, 1988; Solomon, 1999). Extremely low air temperatures  are essential to produce PSC. The PSC  can be used as a surface for heterogeneous interactions, leading to the conversion of reactive halogens from the halogen reservoirs, which can cause serious ozone loss (Frieβ et al., 2005; Marsing et al., 2019). Although the PSC is not only composed of NAT (Pitts et al. 2009; Spang et al. 2018), the temperature threshold for the

75  existence of NAT provides a good estimate on the occurrence of heterogeneous chemistry (Drdla and Müller 2012; Kirner et al. 2015; Grooß and Müller 2021; von der Gathen et al. 2021).

 PSC might also grow large enough to precipitate and remove $HNO_3$ in the stratosphere, which is the reservoir of $NO_2$. The resulting denitrification from the polar vortex hinders chlorine deactivation by $NO_2$ (Salawitch et al., 1989; Arblaster et al. 2014). Active chlorine is rapidly photolyzed because of the recovery of spring sunlight when ozone loss occurs via the self-reaction of ClO (Molina and Molina, 1987), as well as the cross-reaction of ClO and BrO (McElroy et al., 1986). It is essential that the vortex retains low temperatures and carries on as a transport impediment so that ozone can remain depleted without $NO_2$ to inactivate chlorine.

The observed Arctic ozone depletion is invaluable for validating stratospheric ozone simulations and for understanding the processes that cause Arctic stratospheric ozone depletion. Currently, ozone vertical column density (VCD) detection utilizes the characteristic ozone absorption in the UV and visible spectra, which provides accurate ozone identification and quantitative measurements. Ground-based observation of ozone VCD started in the first decades of the twentieth century (Dobson, 1968; Brewer, 1973; Solomon et al., 1987; Bognar et al., 2021). From the 1960s, ozonesondes began to acquire atmospheric ozone data (Logan, 1994; Thomason et al., 2011; Wohltmann et al., 2020; Grooß and Müller, 2021). Since 1978, satellite observations have provided essential data for atmospheric ozone related studies (Kuttippurath et al., 2012; Manney et al., 2020). Among these, ground-based observations are crucial to calibrate remotely sensed observations and optimizing inversion results (Lu et al., 2006). In the 1970s, differential optical absorption spectrometry (DOAS) was developed by Platt and Stutz (2008) and has been widely used to measure several trace gases of ozone, nitrogen dioxide, bromine monoxide, and sulfur dioxide (Hüneke et al., 2017).

The impacts of the aberrantly powerful and persistent vortex on ozone in the Arctic were investigated using satellite observations, ozonosonde measurements, and data from the European Centre for Medium-Range Weather Forecasts (ECMWF) (Wohltmann et al., 2020; Lawrence et al., 2020). The major stratospheric halogen species, chlorine, and bromine were investigated in this ozone depletion event (ODE) (Wohltmann et al., 2017, 2021). In addition, compared to ground-based observation, modelling provides a wider coverage and favours the investigation of ozone depletion.

essential role in the investigation of ozone depletion (Müller et al., 1994; Wohltmann et al., 2010; Griffin et al., 2019; Grooß and Müller, 2021Simpson et al., 2007). Recently, stratospheric chemical patterns, consisting of a group of heterogeneous reactions, have been developed in various models according to investigations and experiments conducted in the polar area (McKenna et al., 2002; Grooß et al., 2011, 2018; Chipperfield, 1999; Khosrawi et al., 2009; Bekki et al., 2013; Chipperfield et al., 1994; Kinnison et al., 2007; Wohltmann and Rex, 2009). Global and area models using different stratospheric chemical patterns have been applied to simulate ozone columns, which usually compare well with satellite observations and ozonosonde data (Pan et al., 2018; Grooß and Müller, 2021).

Accurate ground-based observations can make a significant impact on improvingimprove the accuracy and reliability of models as well as enhancing our understanding of the reasons for ozone depletion. In this study, wWe have developed a ground-based DOAS system that can conduct TCO ozone VCD observations in the Arctic. The zenith scattered light observation mode was applied to measure TCO ozone VCD using the Langley Plot method (Frieß et al., 2005).

We analyze the reasons for this ODE in the unusual spring of 2020 above Ny-Ålesund, Norway. The methods and data are given in Sect. 2, which covers the presentation of the experimental location and DOAS instrument, the DOAS method, the specified dynamics version of the Whole Atmosphere Community Climate Model (SD-WACCM), Global Ozone Monitoring Experiment 2 (GOME-2) observations, Brewer measurements, Système d'Analyze par Observation Zénithale (SAOZ) measurements, ECMWF data, and ozonesonde data. Section 3 presents the results, where Sect. 3.1 describes the results of ozone VCDs from February 2017 to October 2021 and ozone loss difference in spring 2020. The zenith scattered light DOAS (ZSL-DOAS) retrieved the daily variations in ozone VCDs, which were in comparison with GOME-2 observations, Brewer, and SAOZ measurements. A detailed characterization of this ODE is presented for establishing the basis of the subsequent analysis. The relationship between Arctic ozone depletion and meteorological conditions in terms to temperature and potential vorticity (PV) is described in Sect. 3.2. The effect of PV on ozone depletion was investigated using ozone VCD and stratospheric PV data. In Sect. 3.3, this ODE was analyzed using the SD-WACCM to further illustrate the ozone depletion process, and to explore the effects of chemical depletion and dynamic transport on this ODE. The influence of the halogen species is discussed in Sect. 3.4. The comprehensive summary is provided in Sect. 4.

**2 Methods**

**2.1 Ozone VCD observation**

**2.1.1  ZSL-DOAS instrument and Experimental location**

130 ~~The DOAS instrument was placed at the Yellow River Station (78.92° N, 11.93° E) in the Arctic. Figure 1 shows the experimental location and DOAS instrument, in Ny Ålesund, Svalbard, Norway. The DOAS instrument mainly includes the prism, telescope, computer, and spectrometer. This spectrometer was designed for wavelengths between 290 and 420 nm, and had the spectral resolution (FWHM) of 0.5 nm. The ozone slant column density (SCD) was retrieved, with the raw data obtained in the zenith direction.~~

135 The ZSL-DOAS instrument mainly includes the prism, telescope, computer, filter, motor, and CCD spectrometer. The motor controlled the telescope that can change the angle of elevation between the horizon and the zenith. As the angle of elevation changes, the telescope can acquire scattered sunlight at different angles (2°, 3°, 4°, 6°, 8°, 10°, 15°, 30°, and 90°). The quartz fibre can transform the incident light and its numerical aperture is 0.22. The light is received by the spectrometer (Ocean Optics MAYA pro) and measured by a 2048 pixels CCD. This spectrometer was designed for wavelengths between 290 and

140 429 nm, and had the spectral resolution (FWHM) of 0.5 nm. The integration time varied between 100 and 2000 ms due to the light intensity. The detector operates normally at approximately 20°C with a thermal controller. The mercury lamp spectra, offsets and dark currents were calibrated ahead of the experiments. The ZSL-DOAS instrument can detect $O_3$, $NO_2$, OClO, BrO, and $O_4$. The ozone slant column density (SCD) was retrieved, with the raw data obtained in the zenith direction (90°). The ZSL-DOAS instrument was placed at the Yellow River Station (78.92° N, 11.93° E) in the Arctic. Figure 1 shows

145 the ZSL-DOAS instrument and experimental location, in Ny-Ålesund, Svalbard, Norway.

**2.1.2 Principle of the ZSL-DOAS instrument**

Radiation intensity decreases when it passes through absorbing media (mainly trace gases). Because of the different absorption bands, characteristic peaks, and intensities of various gases, we can retrieve the content of each trace gas, according to Lambert–Beer's law as follows:

150
$$ln\frac{I^*(\lambda)}{I_0(\lambda)} = \sum\left[\sigma_j^*(\lambda)c_jL\right] = \sum\left[\sigma_j^*(\lambda)SCD_j\right]. \tag{1}$$

Here, $I_0(\lambda)$ represents the raw intensity of solar scattered spectral radiation received by the ground-based detector, $I^*(\lambda)$ denotes the incident intensity of the solar radiation spectrum, $L$ represents the distance travelled by the incident light in the absorbing gas, $\sigma_j^*(\lambda)$ represents the absorption cross section for the $j$th gas, $c_j$ denotes concentration of the $j$th gas, $SCD_j = \int c_j L$ represents the SCD of the $j$th gas, and $D = \ln\frac{I^*(\lambda)}{I_0(\lambda)}$ denotes the differential optical density.

155  ### **2.1.3 Calculation of ozone VCD**

We calculated the SCD for ozone with the QDOAS program (Platt and Stutz, 2008). In the experiment, ozone was retrieved in the 320–340 nm band, and the gases involved in the retrieval include $O_3$ (223K, 243K), $NO_2$ (298K), $O_4$ (293K), and ring structure. Table 1 lists the parameters for the gases involved in the retrieval. Figure 2 shows a spectrum obtained during monitoring on June 13, 2021. The measured spectrum was fitted to give an ozone SCD of $4.09 \times 10^{17}$ molec cm$^{-2}$, and the

160  root mean square of the spectral fitting residual was $5.28 \times 10^{-4}$.

As SCD is dependent on the instrument's observation mode and the prevailing meteorological conditions, it is necessary to shift to VCD, which is independent of the mode of observation:

$$AMF = \frac{SCD}{VCD}. \tag{2}$$

Here, the Air Mass Factor (AMF) can be obtained from the SCIATRAN model and is influenced by _a priori_ ozone

165  profiletrace gas profiles, pressureSZA (solar zenith angle), temperature, ozone, aerosol profiles, cloudswavelength, and surface albedo. Based on the average monthly climate, _a priori_ ozone profile can be achieved. The SZA calculated in this research ranged between 35° and 80°, with surface albedos between 0.08 and 0.6. Table 2 lists these parameters. Since the "Ring effect" in the measurement caused by the Fraunhofer reference spectra can lead to lower trace gas levels in the retrieval than in actual atmospheric levels, this is corrected in the calculation:

170  $$dSCD(\alpha,\beta) = SCD(\alpha,\beta) - SCD_{FRS} = AMF(\alpha,\beta)VCD - SCD_{FRS}. \tag{3}$$

Here, $SCD_{FRS}$ denotes Fraunhofer absorption. Figure 3 presents the results of a linear fit of the dSCD and AMF on June 13, 2021. The ozone VCD for this date was $8.799 \times 10^{18}$ molec cm$^{-2}$ and produced a fitting error of $3.361 \times 10^{16}$ molec cm$^{-2}$.

**2.1.4 Error estimation**

The uncertainties in ozone VCD retrieval originate from uncertainties in the retrieval of SCD and AMF. The error in retrieving ozone SCD was calculated as 3.01% within the 95% confidence interval.  Table 2 provides the parameters used to calculate the AMF effect on wavelength. The uncertainties of the AMF due to wavelength selection were calculated as $(AMF_\lambda - AMF_{328})/AMF_\lambda$, where λ denotes the wavelength. According to Table 2, the uncertainties of the AMF in the wavelength ranged from −4.257% to 4.630%, and the average uncertainty was 2.030%. Based on evaluation of the OMI ozone products, AMF had an uncertainty of about 2% for *a priori* ozone profile (Bhartia, 2002). The average AMF uncertainty was calculated as 2.85% using the following equation:

$\sqrt{AMF_{wave}^2 + AMF_{profile}^2}$ , where $AMF_{wave}$ denotes the error of AMF influenced by wavelength, and $AMF_{profile}$ denotes the AMF error affected through *a priori* ozone profile. The total error in the retrieved ozone VCD was 4.15%, calculated using the following error equation, $E_{VCD} = \sqrt{E_{SCD}^2 + E_{AMF}^2}$ , where $E_{SCD}$ and $E_{AMF}$ denote the errors of SCD and AMF, respectively.

**2. 2 SD-WACCM**

The physical parameterizations employed in the Community Atmosphere Model Version 4 (CAM4) were applied to the WACCM (Neale et al., 2013). At present, the WACCM model is incorporated into a component set of the Community Earth System Model, whose source code is available online (https://svn-ccsm-release.cgd.ucar.edu/model_versions/). The Model for Ozone and Related Chemical Tracers, version 3 (MOZART-3) provided the chemical parameters for the WACCM (Kinnison et al., 2007). This mechanism contains 52 neutral species, one invariant ($N_2$), 127 neutral gas-phase reactions, 48 neutral photolytic reactions, and 17 heterogeneous reactions [see Tables 5.1-5.5 in Neale et al. (2013)]. The chemical mechanism of WACCM4 also contains 4 aerosol types heterogeneous reactions: liquid binary sulfate (LBS), supercooled ternary solution (STS), nitric acid trihydrate (NAT), and water-ice. When model temperatures above 200K, only the LBS exists. The surface area density (SAD) of LBS is from SAGE, SAGE-II and SAMS observations (Thomason et al., 1997) and Considine update it (World Meteorological Organization, 2003). With the model atmosphere cooling, the LBS aerosol expands and absorbs both $HNO_3$ and $H_2O$ to obtain the STS aerosol. Tabazadeh et al. (1994) derived the composition of STS

by the Aerosol Physical Chemistry Model (ACPM). The STS aerosol median radius and SAD is derived following the approach of Considine et al. (2000). When model temperatures reach a specified supersaturation ratio of $HNO_3$ for NAT, $HNO_3$ containing aerosols are allowed to form. In WACCM4, Peter et al. (1991) set this ratio to 10. NAT median radius and SAD are derived in the same way with STS aerosol. If the derived atmospheric temperature does not exceed the saturation temperature of water vapour on ice ($T_{sat}$), then this results in the formation of water-ice aerosols. In WACCM4, the CAM's prognostic water routines gives the condensed phase $H_2O$, which is conveyed to the chemistry module. According to the method of Considine et al. (2000), the median radius and SAD of water-ice can be derived by this condensed phase $H_2O$. The polar stratospheric cloud module used in this study followed Wegner et al. (2013) rather than the standard module of Kinnison et al. (2007), improving the capabilities of WACCM in modelling ozone and its associated components (Brakebusch et al., 2013). The sedimentation of $HNO_3$ in NAT aerosol follows the approach in Considine et al. (2000). The flux (F) of $HNO_3$ can be derived as follows:

$$F = V \cdot C \cdot \exp(8\ln^2\sigma). \tag{4}$$

here $V$ represents the terminal velocity of NAT aerosol, $C$ denotes the condensed-phase concentration of $HNO_3$, $\sigma$=1.6 (Dye et al., 1992) represents the width of the lognormal size distribution for NAT.

We used the SD-WACCM with meteorological parameters driven by Modern Era Retrospective-Analysis for Research and Applications version 2 (MERRA-2) data (Gelaro et al., 2017). The SD-WACCM had the horizontal resolution of $1.9° \times 2.5°$ (lat $\times$ lon). The model was divided vertically into 88 layers, covering an altitude of ~140 km from the ground to the bottom of the lower thermosphere layer. Meteorological fields were calculated using a nudging method in the model (Lamarque et al., 2012). Data for the horizontal winds, temperature, and surface pressure from MERRA-2 were used to drive the physical parameterization from the surface to 50 km (Kunz et al., 2011), which allowed for more accurate comparisons between the measurements of atmospheric composition and the model output (Lamarque et al., 2012). This can be employed for the study of specific weather events. Linear transitions were used in the 50–60 km altitude range and over 60 km, and online calculations were performed. In this study, the MERRA-2 dataset has the same resolution with the SD-WACCM, which can be accessed on the Earth System Grid (https://www.earthsystemgrid.org/home.html) and are obtained from the original resolution ($1/2°\times2/3°$) by a conservative re-gridding procedure (Lamarque et al., 2012; Pan et al., 2019). In this study, the

[revised manuscript text omitted]

Ozone depletion occurs when the temperature is sufficiently low and reaches a threshold temperature. In addition, aA cold and stable polar vortex is a prerequisite for ensuring that Arctic stratospheric temperatures are sufficiently low. The 2019/2020 winter was unique and the polar vortex was unusually stable, prolonged, and cold (Lawrence et al., 2020;

Wohltmann et al., 2020; Rao and Garfinkel, 2020). A large and strong Arctic vortex lasted from early December anomalously into the final week of April (Kuttippurath et al., 2021). The faint planetary wave activity in the Northern Hemisphere also contributed to the formation of a cold and strong vortex (Feng et al., 2021). Unusually low temperature and strong and prolonged vortex in the 2019/2020 winter provided favourable meteorological conditions for ozone depletion in

300 the Arctic.

305 ~~VCD. Similarly, Arctic spring ozone depletion was closely related to PV. When ozone VCD decreased, the PV value increased. The ozone VCDs fluctuated between 241.2–334.6 DU, with ozone recovering to 388.1 DU on April 19, and then returning to normal values (Fig. 8a). The observed ozone VCD and temperature had similar fluctuation patterns, suggesting that ozone was significantly depleted in the colder Arctic low stratospheric vortex. Thus, the effect of the polar vortex on ozone depletion in the stratosphere was clear.~~

310 **3.3 Impact of halogen species**

To further research the conditions and mechanisms of this ODE, we used a chemical model to characterize chemical components between November 1, 2019, and July 1, 2020. To validate the reliability of the WACCM simulated results, we needed to prove its capability for recreating observations in the atmosphere. Therefore, we compared WACCM simulations with ozonesonde measurements. Figure  8 presents the comparison of temperature and ozone profiles over Ny-Ålesund
315 from the WACCM simulations and the ozonesonde between January 1 and July 1, 2020. Fig. 8a–b shows a gradual depletion of ozone from 16 to 20 km in early March, and the mixing ratio at a similar altitude was unusually low from late March to early April, which corresponded to ground-based observations. A mixing ratio of less than 0.5 ppmv within the altitude range suggested that ozone was nearly completely depleted (Bognar et al., 2021). This low value was uncommon, as the ozone mixing ratio was above 0.5 ppmv over the Arctic during 2011 (Solomon et al., 2014). There was an aberrantly
320 cold spring in 2020, with low temperatures lasting until mid-April (Fig. 8c–d). In January and February 2020, the

temperature in the 15–25 km altitude range was lower than $T_{nat}$, providing favourable conditions for PSC formation. As shown in Fig. 98, the WACCM can depict the evolution of ozone during this Arctic ozone depletion event, but there are some problems with overestimation of ozone concentration. On the one hand, the chlorine activation on PSCs is probably underestimated due to the overestimation of temperature simulated by the model. Similar conclusions have also been reported by previous studies (Brakebusch et al., 2013; Solomon et al., 2015; Pan et al., 2018). On the other hand, Müller et al. (1994) found that the $CH_3O_2+ClO$ reaction is also important but is not included in the current chemical mechanism of the model and could be taken into account in future models to improve simulation performance. accurate simulations of the ozone and temperature profiles strengthened the credibility of the WACCM results. However, there are some discrepancies that exist in the model and observations. Because of the overestimation of temperature, the catalytic cycles that cause ozone depletion in PSCs are underestimated. Therefore, there was an overestimation of ozone by the model compared to the observations.

Between late December 2019 and January 2020, we observed abnormally increasing, high $HNO_3$ values above Ny-Ålesund (Fig. 10b), which favoured the formation and existence of PSC (Hanson and Mauersberger, 1988). suggesting abundant PSC formations. In contrast, in January 2011, analogous but lower values were recorded (Manney et al., 2011; Manney et al., 2020). Between late January and early February 2020, $HNO_3$ changed abruptly from abnormally high values to normal values, which indicated the abundant PSC activities of the period (Bognar et al., 2021). The low value of $HNO_3$ at 20-22 km is probably due to low temperatures (Fig. 8c–d) leading to PSC activity and severe denitrification (Ardra et al., 2022).

[revised manuscript text omitted]

395 predominantly present as HCl and $ClONO_2$ before activation, whereas bromine was predominantly present as HOBr and BrCl before activation. Particularly, bromine existed mainly as HOBr before chlorine activation began. When chlorine was activated, bromine existed mainly as BrCl. In addition, formation of HCl is considered to be the main chlorine deactivation mechanism in Antarctica (Müller et al., 2018). However, in the Arctic, due to HCl increased more slowly than in the Antarctic, chlorine was mainly deactivated as $ClONO_2$.

400 ~~significant increase in reactive chlorine, resulting in an unprecedented ozone loss in the Arctic. An apparent HCl increase occurred in mid April 2020, when $ClONO_2$ stopped increasing and ClO was almost depleted as the ozone concentration started to recover. Before chlorine activation began, bromine mainly existed as HOBr; however, after chlorine activation, bromine mainly existed in the form of BrCl. Furthermore, they rapidly photolyzed to Br in the daytime and had high potential to cause ozone depletion.~~

405

In summary, by ZSL-DOAS observations, we provided another evidence for unprecedented ozone depletion during the Arctic spring of 2020. The ZSL-DOAS ozone VCD observations can also provide calibration for satellite observations and

410 model simulations, and in the future can provide the support for observations at more Chinese research stations or international local stations in the polar area. Additionally, although WACCM can depict the evolution of ozone during this Arctic ozone depletion event, there are some problems such as overestimation of the temperature and the $CH_3O_2$+ClO reaction is not considered in the current chemical mechanism of the model. This could be considered in future models to improve the simulation performance.

415 *Data availability.* Measurements and calculation of ozone VCDs above Ny-Ålesund, Norway, from 2017 to 2021 and the results from the SD-WACCM used in this research are available from https://doi.org/10.17632/jx7nkspkg7.1 . GOME-2 data are download from https://avdc.gsfc.nasa.gov/, Brewer data from

https://woudc.org/, SAOZ data from http://saoz.obs.uvsq.fr/, ECMWF data from https://cds.climate.copernicus.eu/, and ozonesonde data from https://ndaccdemo.org/.

420 *Author contribution.* QL: Methodology, Investigation, Software, Formal analysis, Validation, Visualization, Writing. YL: Funding acquisition, Methodology, Formal analysis, Writing, Reviewing, Editing, Resources. YQ: Methodology, Software, Formal analysis, Visualization. CP: Methodology, Software, Formal analysis, WACCM simulation. KD: Validation, Resources. XH: Methodology, Formal analysis. FS: Validation, Resources. WL: Supervision.

*Competing interests.* The authors declare that they have no conflict of interest.

425 *Acknowledgement.* This research was financially supported by the National Natural Science Foundation of China (Grant Nos.41941011 and 41676184) and the Youth Innovation Promotion Association of CAS (Grant No.2020439). We thank the organizations of the Chinese Arctic and Antarctic Administration (CAAA), Polar Research Institute of China, and teammates of the Chinese Arctic Yellow River Station for their kind help. We gratefully thank the BIRA for providing the QDOAS software. The GOME-2 data can be available from the University of Bremen. The Brewer data can be provided by the World
430 Ozone and Ultraviolet Radiation Data Centre. We appreciate Florence Goutail for providing the SAOZ data. We gratefully thank the Alfred Wegener Institute for providing ozonesonde data. We also gratefully thank ECMWF for providing the ERA5 data.

690 **Table 1. Fitting parameters of spectral retrieval.**

| Parameter | References |
|---|---|
| $O_3$ | 223K, 243K (Bogumil et al., 2003) |
| $O_4$ | 293K (Hermans et al., 2003) |
| $NO_2$ | 298K (VanDaele et al., 1996) |
| Ring | Calculated using QDOAS |
| Fitting Interval | 320–340 nm |
| Polynomial | 5 |

**Table 2. The fitting parameter nodes for spectral retrieval.**

| Parameters | Nodes |
|---|---|
| SZA (°) | 35, 40, 45, 50, 55, 60, 65, 70, 75, 80 |
| Surface albedo | 0.05, 0.1, 0.2, 0.3, 0.4, 0.5, 0.6 |
| Wavelength (nm) | From 320 to 340 in 0.5 intervals |

**Table 3. The number of days below T$_{nat}$ and daily average temperatures (December–February).**

| Date | Days below T$_{nat}$ | Average / interval temperatures (K) |
|---|---|---|
| 2016.12–2017.2 | 0 | 203.5 / 195.2–214.8 |
| 2017.12–2018.2 | 26 | 203.6 / 190.6–236.2 |
| 2018.12–2019.2 | 0 | 211.8 / 198.1–226.5 |
| 2019.12–2020.2 | 32 | 196.9 / 190.2–206.1 |
| 2020.12–2021.2 | 6 | 205.3 / 192.5–225.1 |

[Figure]

**Figure 1. The ground-based ZSL-DOAS instrument and experiment site in Ny-Ålesund.**

[Figure]

695

**Figure 2. Spectrum fits of ozone on June 13, 2021.**

[Figure]

**Figure 3. Linear fit between ozone dSCDs and AMFs for the (a) morning and (b) afternoon on June 13, 2021. The correlation coefficients ($R^2$) are 0.99951 and 0.99909. The ozone VCDs for the morning and afternoon are $8.754 \times 10^{18}$ molec cm$^{-2}$ and $8.844 \times 10^{18}$ molec cm$^{-2}$. The calculated ozone VCD for June 13, 2021 is $8.799 \times 10^{18}$ molec cm$^{-2}$.**

[Figure]

**Figure 4. The ozone VCDs from ZSL-DOAS, GOME-2, Brewer, and SAOZ.**

[Figure]

**Figure 6̶5.** (a) Ozone data for 2020 and the average ozone data (black) of 2017, 2018, 2019, and 2021. (b)  relative ozone  difference for 2020.

[Figure]

**Figure 56. Scatter plots and linear fits of retrieved ozone VCDs with (a) GOME-2, (b) Brewer, and (c) SAOZ.**

[Figure]

710 **Figure 7. Temperatures (at 70 hPa) over Ny-Ålesund from November 2016 to September 2021, where the blue line denotes the threshold temperature for the  existence of PSCs.**

[Figure]

**Figure 98. Between January 1 and July 1, 2020, ozone profiles from (a) ozonesonde measurements and (b) the WACCM simulation, and temperature profiles from (c) ozonesonde measurements and (d) the WACCM simulation.**

[Figure]

715

**Figure 9. Simulated average diurnal profiles of the chlorine and bromine compounds between November 1, 2019, and July 1, 2020, at heights of 10–25 km above Ny-Ålesund: (a) O₃; (b) HNO₃; (c) NO₂; (d) ClO; (e) HCl; (f) ClONO₂; (g) BrO; (h) HBr; (i) BrONO₂.**

[Figure]

Figure 10. Simulated average diurnal mixing ratios of ozone, chlorine, and bromine compounds between November 1, 2019, and July 1, 2020, at a height of 17.5 km above Ny-Ålesund: (a) mixing ratios of ozone and chlorine ($Cl_t$ = ClO + HCl + HOCl + $ClONO_2$); (b) mixing ratios of ozone and bromine ($Br_t$ = BrO + HBr + HOBr + $BrONO_2$ + BrCl).